# A Theory of Feature Learning Dynamics in Transformers: How Depth Reshape Learning Dynamics

## Abstract

Transformers achieve remarkable empirical success, yet we still lack a theory of how their features are learned during training at scale, particularly as depth increases. Using the *Neural Feature Dynamics* (NFD) framework, we develop a training-time theory that characterizes feature and gradient evolution in Transformers trained under depth-$\mu$P scaling in the joint infinite-width and infinite-depth limit. NFD decomposes the training dynamics of each representation into Gaussian, learning, and interaction components, with the distinct depth orders determined by architectural placement. This decomposition reveals a qualitative, depth-induced *shift in feature learning dynamics*: while shallow Transformers actively learn both residual-stream dynamics and internal representations (including attention and MLP features), deep Transformers concentrate learning in the residual stream while *silently* suppressing internal feature updates without loss or gradient spikes, leading to nearly uniform attention scores. NFD further predicts a *structural drift* at large depth under which serial attention–MLP compositions behave effectively as *parallel* pathways. Guided by this analysis, we explore depth-aware interventions that restore internal learning while preserving residual stability, including asymmetric learning rates and decoupled backward weights, and validate these theoretical predictions through numerical experiments.

## 1. Introduction

Transformers (Vaswani et al., 2017) underpin modern large-scale AI systems, and their performance continues to improve as model size, data, and compute scale, a phenomenon

[1] Anonymous Institution, Anonymous City, Anonymous Region, Anonymous Country. Correspondence to: Anonymous Author <anon.email@domain.com>.

Preliminary work. Under review by the International Conference on Machine Learning (ICML). Do not distribute.

*Table 1.* Depth-induced learning phase change in Transformers.

|  | Shallow | Deep |
|---|---|---|
| Residual stream | Active | Active |
| Internal features | Active | Suppressed |
| Attention patterns | Structured | Uniform |
| Block composition | Serial | Parallel |
| Internal learning time | Comparable | Grows with depth |

commonly called *scaling laws* (Kaplan et al., 2020; Hoffmann et al., 2022). While scaling width is now relatively well understood and stable, scaling *depth* remains brittle in practice. Empirically, deeper Transformers often exhibit training instabilities (Wang et al., 2024; Wortsman et al., 2024) and attention pathologies (Dong et al., 2021; Zhai et al., 2023), and they require careful parameterization and warm-up for stable optimization (Liu et al., 2020; Xiong et al., 2020). Despite the impact of depth, we still lack a principled *training-time* theory that predicts *how* Transformer features are learned as depth grows, and *why* some components become harder to train in deep regimes.

Transformers can be viewed as residual networks due to their skip connections (He et al., 2016). Motivated by this connection, prior work analyzes the training dynamics of deep ResNets with single-layer residual branches under depth scaling (Du et al., 2019; Tirer et al., 2022; Gao et al., 2025; Marion et al., 2024). These analyses establish stability and optimization convergence in overparameterized regimes, but they largely focus on kernel or lazy learning (Jacot et al., 2018; Chizat et al., 2019), where features remain close to random initialization (Yang & Hu, 2021). In contrast, the $\mu$P feature-learning framework studies training in the large-width limit (Yang & Hu, 2021). Recent work extends $\mu$P along the depth axis via *depth-$\mu$P* (Yang et al., 2024). For deep ResNets, this line of work shows that $1/\sqrt{L}$ depth scaling can stabilize training (Hayou et al., 2021), enable hyperparameter (HP) transfer across depth (Yang et al., 2024; Bordelon et al., 2024b), and restore gradient independence assumptions during training (Yao et al., 2025). However, it remains unclear whether, and how, these insights extend to Transformers.

Transformers differ from one-layer residual networks because each block contains multiple interacting sublayers (attention and two-layer MLPs) and maintains multiple in-

ternal representations (queries, keys, values, and intermediate MLP features). This structure induces richer forward–backward interactions. Recent empirical and theoretical studies report that depth-$\mu$P fails to support effective learning or HP transfer once residual blocks contain more than one layer (Yang et al., 2024; Dey et al., 2025); gradients to internal representations become increasingly suppressed at large depth (Wortsman et al., 2024; Bordelon et al., 2024a; Yao et al., 2025); and Transformer block composition can change with scale (Chowdhery et al., 2023; Dehghani et al., 2023). These effects are distinct from classical training instabilities: optimization can remain numerically stable, without loss or gradient spikes, while learning of certain representations *silently* collapses. Without a rigorous training-time theory for feature learning in deep Transformers, it is difficult to isolate the mechanisms behind this suppression or to design principled remedies.

This work develops a training-time theory for deep Transformers trained with one-pass stochastic gradient descent (SGD) in the joint infinite-width and infinite-depth limit. Our analysis targets modern depth-scaled residual parameterizations used in large-scale pretraining, including GPT-style models (Radford et al., 2019; Brown et al., 2020) and Megatron-LM (Shoeybi et al., 2019). We derive a coupled forward–backward stochastic differential equation (SDE) system, *Neural Feature Dynamics (NFD)*, that captures the co-evolution of features and gradients during training. In contrast to prior dynamical mean-field approaches (Bordelon et al., 2024a) that primarily characterize *qualitative* limits, NFD *quantifies* how depth scaling and forward–backward weight coupling shape learning at large depth. Building on NFD, our main contributions are as follows:

- **Three-term decomposition revealed by NFD.** In deriving NFD, we find that Transformer feature dynamics naturally decompose into three components: **Gaussian** fluctuations from random initialization, gradient-driven **learning** terms, and forward–backward **interaction** terms induced by weight reuse in backpropagation. We establish explicit depth orders for these components. These depth orders underpin our analysis of learning suppression and serial-to-parallel drift at large depth, and they guide principled remedies that target suppressed channels while controlling channels that can become unstable.

- **Depth-induced learning suppression and regime transition.** Using the NFD decomposition, we identify a depth-induced learning phase transition. In shallow regimes, SGD actively updates both the residual stream and internal representations in attention and MLP sublayers. In sufficiently deep regimes, learning concentrates in the residual stream while updates to internal features become asymptotically suppressed, yielding vanishing attention logits, nearly uniform attention patterns, and increasingly long learning horizons. Beyond suppression, we show

that serial attention–MLP composition drifts toward effectively parallel pathways at large depth. Altogether, this provides a principled explanation for the failure of depth-$\mu$P in complex Transformer architectures. Table 1 summarizes these regimes.

- **Depth-aware remedies via targeted isolation of suppressed channels.** Leveraging NFD's quantitative depth orders, we identify which learning and interaction channels become asymptotically suppressed in deep Transformers, and which can become unstable under naive interventions. We show that naively amplifying the learning rate (LR) of suppressed features can trigger *exploding interaction* on other features, a failure mode that, to our knowledge, has not been previously documented. Guided by this analysis, we develop remedies including asymmetric LR schemes and selective backward weight decoupling that isolate suppressed channels and can restore effective learning while avoiding instability. We also identify a practical tradeoff: while decoupling counteracts learning collapse and improves HP transfer, it can introduce gradient mismatch and slightly degrade performance, suggesting further room for depth-aware design and study.

## 2. Warm-Up: Neural Feature Dynamics via a Two-Layer ResNet

We introduce Neural Feature Dynamics (NFD) (Yao et al., 2025) through a depth-$L$ ResNet with a two-layer MLP residual block. This model isolates the mechanisms behind depth-dependent feature learning while remaining tractable.

Consider a depth-$L$ ResNet $f_\theta(\boldsymbol{x})$ with residual stream:

$$\boldsymbol{h}_\ell = \boldsymbol{h}_{\ell-1} + L^{-1/2}\boldsymbol{W}_\ell\phi(\boldsymbol{x}_\ell), \tag{1}$$

where $\boldsymbol{x}_\ell = \boldsymbol{U}_\ell\boldsymbol{h}_{\ell-1}$ is the *internal representation* after the first layer, and $\boldsymbol{W}_\ell, \boldsymbol{U}_\ell \in \mathbb{R}^{N \times N}$ are trainable weights. We initialize weights under $\mu$P (Yang & Hu, 2021) with fan-in scaling $\mathcal{N}(0, 1/N)$ (He et al., 2015), and use depth scaling $L^{-1/2}$ following depth-$\mu$P (Yang et al., 2024). Similar depth-scaled residual parameterizations appear in large-scale pretraining, including GPT-style models (Brown et al., 2020) and Megatron-LM (Shoeybi et al., 2019)[1].

**Backward dynamics.** Let $\delta\boldsymbol{h}_\ell$ denote the appropriately rescaled gradient $\partial_{\boldsymbol{h}_\ell}f_\theta$, normalized so that its coordinates remain $\Theta(1)$ with respect to both width $N$ and depth $L$. The backward recursion is given by

$$\delta\boldsymbol{h}_{\ell-1} = \delta\boldsymbol{h}_\ell + L^{-1/2}\boldsymbol{U}_\ell^\top\delta\boldsymbol{x}_\ell,$$

where $\delta\boldsymbol{x}_\ell = \boldsymbol{W}_\ell^\top\delta\boldsymbol{h}_\ell \odot \phi'(\boldsymbol{x}_\ell)$. We train the network $f_\theta$ using streaming SGD over *i.i.d.* fresh samples on a loss $\mathcal{L}$.

---

[1]Many technical reports bury or fail to release the specific scaling details; in contrast (Dey et al., 2023, Table 14) offers a detailed summary.

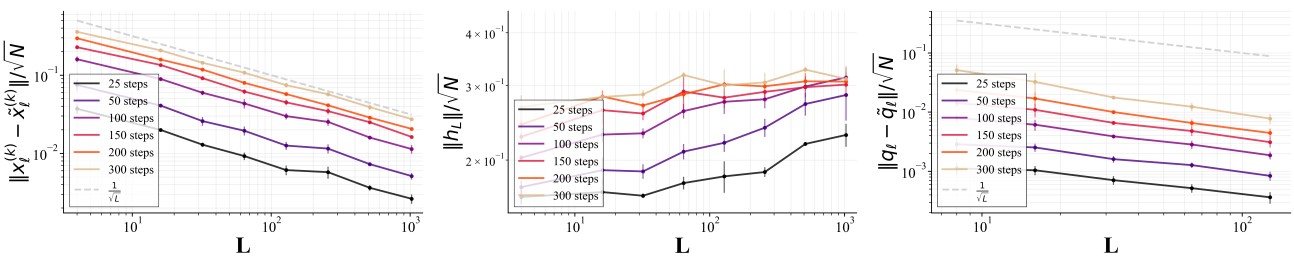

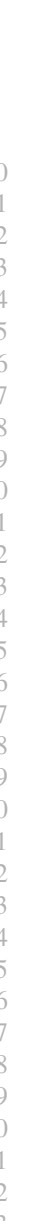

*Figure 1.* **Silent collapse and a pitfall of naive fixes.** (L) Internal learning collapses with depth under depth-$\mu$P Eq. (4). (M) Naively amplifying learning rates can trigger interaction-driven blow-up. (R) Analogous depth-induced collapse occurs in the attention sublayer.

## 2.1. Residual-stream learning dynamics for $\{h_\ell, \delta h_\ell\}$

Under $\mu$P scaling and a finite SGD updates, the forward variables $\{x_\ell, h_\ell\}$ and backward variables $\{\delta x_\ell, \delta h_\ell\}$ are generated by valid Tensor Program (TP) operations (Yang & Hu, 2021). Hence, the training process forms a valid TP.

**Proposition 2.1.** *For any finite SGD steps $k \in \mathbb{N}$, training a two-layer ResNet defined in Eq. (1) is a valid TP.*

The TP Master Theorem (Yang & Hu, 2021, Theorem 7.4) then characterizes the large-width limit: For any finite collection $\{h_m\}_{m=1}^M$ (each $h_m \in \mathbb{R}^N$ may be a feature or a gradient) and any smooth test function $\psi : \mathbb{R}^M \to \mathbb{R}$,

$$\frac{1}{N} \sum_{i=1}^N \psi(h_{1,i}, \ldots, h_{M,i}) \longrightarrow \mathbb{E}\big[\psi(Z^{h_1}, \ldots, Z^{h_M})\big],$$

where $Z^{h_m}$ is the limit of a typical coordinate of $h_m$. Thus, coordinates of features and gradients behave as asymptotically *i.i.d.* samples of $Z^{h_m}$, yielding a mean-field description of forward and backward dynamics.

Applying the Master Theorem, we obtain the one-step SGD dynamics for $\{h_\ell^{(1)}\}$ and $\{\delta h_\ell^{(1)}\}$. This one-step form already exposes the depth orders that drive the regime change; the general $k$-step result is deferred to Appendix B.

**Proposition 2.2.** *Suppose $\phi$, $\phi$, and $\mathcal{L}'$ are pseudo-Lipschitz continuous. As $N \to \infty$, the coordinates of $\{h_\ell^{(1)}\}$ and $\{\delta h_\ell^{(1)}\}$ becomes asymptotically i.i.d. and satisfies:*

$$\underbrace{Z^{h_\ell^{(1)}} - Z^{h_{\ell-1}^{(1)}}}_{Residual} = \underbrace{\frac{Z^{\widetilde{h}_\ell^{(1)}}}{\sqrt{L}}}_{Gaussian} - \underbrace{\frac{\eta_W}{L} \mathring{\mathcal{L}}' K_{x_\ell}^{(0,1)} Z^{\delta h_\ell^{(0)}}}_{Learning}$$
$$- \underbrace{\frac{\eta_U}{L} \mathring{\mathcal{L}}' K_{h_{\ell-1}}^{(0,1)} K_{\phi'_\ell}^{(0,1)} Z^{\delta h_\ell^{(0)}}}_{Interaction}, \quad (2)$$

$$Z^{\delta h_{\ell-1}^{(1)}} - Z^{\delta h_\ell^{(1)}} = \frac{Z^{\delta \widetilde{h}_\ell^{(1)}}}{\sqrt{L}} - \frac{\eta_U}{L} \mathring{\mathcal{L}}' K_{\delta x_\ell}^{(0,1)} Z^{h_{\ell-1}^{(0)}}$$
$$- \frac{\eta_W}{L} \mathring{\mathcal{L}}' K_{\delta h_\ell}^{(0,1)} K_{\phi'_\ell}^{(0,1)} Z^{h_{\ell-1}^{(0)}}, \quad (3)$$

*where $K_{x_\ell}^{(a,b)} := \mathbb{E}[Z^{x_\ell^{(a)}} Z^{x_\ell^{(b)}}]$, and $(Z^{\widetilde{h}_\ell^{(1)}}, Z^{\delta \widetilde{h}_\ell^{(1)}})$ are centered Gaussian innovation terms. Moreover, across*

depth $\ell$, these innovations can be taken independent conditional on the TP limit state.

Equations Eq. (2)–Eq. (3) yield a three-term update for the residual stream: (i) a **Gaussian** term of order $L^{-1/2}$ from initialization, (ii) a gradient-driven **learning** drift of order $L^{-1}$, and (iii) a forward–backward **interaction** drift of order $L^{-1}$ from weight reuse in backpropagation. This three-term decomposition is the basic lens of NFD. It also recurs for Transformer features, with depth orders determined by each representation's position relative to skip connections and other within-block compositions.

We convert the layer index into a continuous depth variable by setting $t = \ell/L \in [0, 1]$ and $\Delta t = 1/L$. Viewing Eq. (2)–Eq. (3) as an Euler–Maruyama discretization with step size $\Delta t$, the $L^{-1/2}$ Gaussian innovations correspond to diffusion terms and the $L^{-1}$ contributions correspond to drift. Under standard regularity conditions (e.g., Lipschitz drift and uniformly non-degenerate diffusion (Yang & Hu, 2021; Yao et al., 2025)), the Euler–Maruyama theorem implies convergence of the depth-discretized dynamics to an SDE in the joint limit $N, L \to \infty$.

**Proposition 2.3.** *Under the assumptions of Theorem B.10, as $N, L \to \infty$ sequentially, the ResNet $f^{(1)}$ Eq. (1) converges to $\mathring{f}^{(1)} = \mathbb{E}[Z^{h_1^{(1)}} Z_1^{\delta h^{(1)}}]$, where the features $Z^{h_t^{(1)}}$ and gradients $Z^{\delta h_t^{(1)}}$ evolve according to the NFD system*

$$dZ^{h_t^{(1)}} = dw_t - \big(\eta_W K_{x_t}^{(0,1)} + \eta_U K_{h_t}^{(0,1)} K_{\phi'_t}^{(0,1)}\big) \mathring{\mathcal{L}}' Z^{\delta h_t^{(0)}} dt,$$
$$dZ^{\delta h_t^{(1)}} = d\widetilde{w}_t - \big(\eta_U K_{\delta x_t}^{(0,1)} + \eta_W K_{\delta h_t}^{(0,1)} K_{\phi'_t}^{(0,1)}\big) \mathring{\mathcal{L}}' Z^{h_t^{(0)}} dt,$$

*where $w_t$ and $\widetilde{w}_t$ are independent Brownian motions induced by the depth-accumulated Gaussian innovations.*

Generally, the Brownian motion's dimension grows with the number of gradient updates, reflecting accumulated stochasticity during training, while the drift terms also accumulate and capture the progressive evolution of learned representations. Importantly, the drift here includes not only gradient-driven learning terms but also forward–backward interaction terms, leading by *different* learning-rate factors. The depth order of the interaction term depends on the residual-block

structure and therefore determines the limiting drift. For example, prior work shows that for *single-layer* ResNets the interaction term appears at order $L^{-2}$ (rather than $L^{-1}$) and vanishes as $L \to \infty$ (Yao et al., 2025). In Transformers, multiple sublayers introduce additional couplings, leading to more intricate learning–interaction decompositions and depth orders; see Section 3 for a detailed discussion.

**2.2. Internal representation learning $\{\boldsymbol{x}_\ell, \delta\boldsymbol{x}_\ell\}$**

The two-layer residual block introduces internal representations $(x_\ell, \delta x_\ell)$ that do *not* have a skip connection. We now state their one-step mean-field dynamics and highlight the depth order that leads to internal learning suppression.

**Proposition 2.4.** *Suppose $\phi$, $\phi'$, and $\mathcal{L}'$ are pseudo-Lipschitz continuous. As $N \to \infty$, the coordinates of $\{\boldsymbol{x}_\ell\}$ and $\{\delta\boldsymbol{x}_\ell\}$ become asymptotically i.i.d. and obey*

$$Z^{\boldsymbol{x}_\ell^{(1)}} = \underbrace{Z^{\widetilde{\boldsymbol{x}}_\ell^{(1)}}}_{Gaussian} - \underbrace{\frac{\eta_U}{\sqrt{L}} \mathring{\mathcal{L}}' K_{h_{\ell-1}}^{(0,1)} Z^{\delta\boldsymbol{x}_\ell^{(0)}}}_{Learning}$$

$$- \underbrace{\frac{\eta_W}{L^{3/2}} \mathring{\mathcal{L}}' K_{\phi_{\ell-1}}^{(0,1)} Z^{\delta\boldsymbol{x}_\ell^{(0)}}}_{Interaction},$$

$$Z^{\delta\boldsymbol{x}_\ell^{(1)}} = Z^{\delta\widetilde{\boldsymbol{x}}_\ell^{(1)}} - \frac{\eta_W}{\sqrt{L}} \mathring{\mathcal{L}}' K_{\delta h_\ell}^{(0,1)} \phi(Z^{\boldsymbol{x}_\ell}) \phi'(Z^{\bar{\boldsymbol{x}}_\ell})$$

$$- \frac{\eta_U}{L^{3/2}} \mathring{\mathcal{L}}' K_{\delta x_{\ell+1}}^{(0,1)} \phi(Z^{\boldsymbol{x}_\ell}) \phi'(Z^{\bar{\boldsymbol{x}}_\ell}),$$

*where $(Z^{\widetilde{\boldsymbol{x}}_\ell^{(1)}}, Z^{\delta\widetilde{\boldsymbol{x}}_\ell^{(1)}})$ are centered Gaussian arising from the TP limit (not an accumulated depth diffusion term).*

Unlike the residual stream, the internal $(\boldsymbol{x}_\ell, \delta\boldsymbol{x}_\ell)$ do not accumulate independent depth innovations through skip connections. More importantly, their gradient-driven learning contributions scale as $\eta_U L^{-1/2}$ (and $\eta_W L^{-1/2}$ for $\delta\boldsymbol{x}_\ell$), which vanish for constant learning rates as $L \to \infty$. Thus,

$$\|\boldsymbol{x}_\ell - \widetilde{\boldsymbol{x}}_\ell\|/\sqrt{N} \sim \eta_U L^{-1/2}, \tag{4}$$

where $\widetilde{\boldsymbol{x}}_\ell$ is the *frozen-weight* baseline generated by the same forward pass but with weights held fixed to the initialization. Consequently, in the large-depth regime, the residual stream can continue to learn while internal representations experience *asymptotic learning suppression*.

**Proposition 2.5.** *Under the same assumptions of Proposition 2.3, as $N, L \to \infty$ sequentially, the internal features obey $Z^{\boldsymbol{x}_t^{(1)}} = Z^{\widetilde{\boldsymbol{x}}_t^{(1)}}$, $Z^{\delta\boldsymbol{x}_t^{(1)}} = Z^{\delta\widetilde{\boldsymbol{x}}_t^{(1)}}$, where centered Gaussian $(Z^{\widetilde{\boldsymbol{x}}_t^{(1)}}, Z^{\delta\widetilde{\boldsymbol{x}}_t^{(1)}})$ are arising from the TP limit.*

Importantly, this collapse is *silent*: it occurs without classical instability signatures (no loss or gradient spikes) because the residual stream learning remains stable and active. At the same time, it does *not* reduce the network to lazy training in a trivial sense: even if the *weights* governing internal channels effectively stop learning, the *distribution* of internal $\boldsymbol{x}_\ell$

still evolve indirectly through the actively learned residual stream $\boldsymbol{h}_\ell$. This mechanism underlies the depth-induced regime shift summarized in Table 1 and will reappear in attention, where $(\boldsymbol{q}, \boldsymbol{k}, \boldsymbol{v})$ play the role of internal channels.

Figure 1 illustrates two warm-up takeaways. (Left) Internal feature learning collapses with depth, consistent with the $\mathcal{O}(L^{-1/2})$ scaling in Eq. (4). (Middle) Naively amplifying depth-suppressed learning rates can destabilize training by amplifying interaction effects, motivating the targeted remedies in Section 4.

## 3. Learning Collapse in Deep Transformer

Section 2 introduced NFD through a two-layer ResNet and showed that, under depth scaling, each representation admits a quantitative three-term decomposition into *Gaussian* (initialization-driven), *learning* (gradient-driven), and *interaction* (forward–backward weight-reuse) contributions. We now apply the same framework to deep Transformers and identify a depth-induced *silent learning collapse* in attention: while optimization remains numerically stable, learning in key internal representations (queries, keys, and values) vanishes as depth increases. This section establishes the mechanism, and Section 4 uses it to design depth-aware remedies. Since the regularity assumptions on the activation $\phi$, loss $\mathcal{L}$, and limiting NFD systems parallel those in the two-layer ResNet analysis, we present informal statements throughout this section to emphasize the core mechanisms and depth-order insights, and defer formal theorems and technical conditions to Appendix D.

We study a ViT-style Transformer (Dosovitskiy et al., 2021) with $S$ tokens with a CLS token for supervision. At layer $\ell \in [L]$ and token index $s \in [S]$, the single-head attention sublayer is

$$\boldsymbol{q}_{\ell,s} = \boldsymbol{W}_{\ell,q} \boldsymbol{h}_{\ell-1,s}, \boldsymbol{k}_{\ell,s} = \boldsymbol{W}_{\ell,k} \boldsymbol{h}_{\ell-1,s}, \boldsymbol{v}_{\ell,s} = \boldsymbol{W}_{\ell,v} \boldsymbol{h}_{\ell-1,s},$$

$$b_{\ell,ss'} = \frac{\langle \boldsymbol{q}_{\ell,s}, \boldsymbol{k}_{\ell,s'} \rangle}{N}, \quad a_{\ell,ss'} = \frac{e^{b_{\ell,ss'}}}{\sum_{s''} e^{b_{\ell,ss''}}},$$

$$\boldsymbol{o}_{\ell,s} = \sum_{s'} a_{\ell,ss'} \boldsymbol{v}_{\ell,s'},$$

$$\boldsymbol{h}_{\ell,s} = \boldsymbol{h}_{\ell-1,s} + L^{-\frac{1}{2}} \boldsymbol{W}_{\ell,o} \boldsymbol{o}_{\ell,s}, \tag{5}$$

All weight matrices are $\mu$P-initialized as in Section 2. Each block also contains a two-layer MLP residual sublayer with the same form and parameterization as in Section 2, and we train all parameters using streaming SGD; see Appendix D for the full ViT block and backward recursion.

We adopt the $\mu$P-consistent $1/N$ logit scaling in Eq. (5), which yields a non-degenerate joint large-width and large-depth limit in the feature-learning regime (Yang & Hu, 2021; Hron et al., 2020; Bordelon et al., 2024a). For scaling anal-

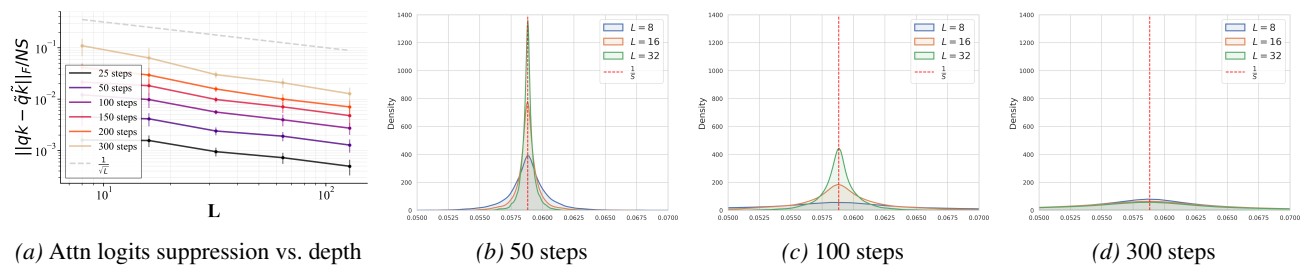

*(a)* Attn logits suppression vs. depth     *(b)* 50 steps     *(c)* 100 steps     *(d)* 300 steps

*Figure 2.* **Depth-induced attention-learning delay.** (a) Attention-logit deviations decay approximately as $\mathcal{O}(L^{-1/2})$ with depth. (b–d) Attention scores at layers $\ell \in \{8, 16, 32\}$ during training: attention stays near-uniform early, while selectivity emerges first in shallow layers and reaches deeper layers after longer training.

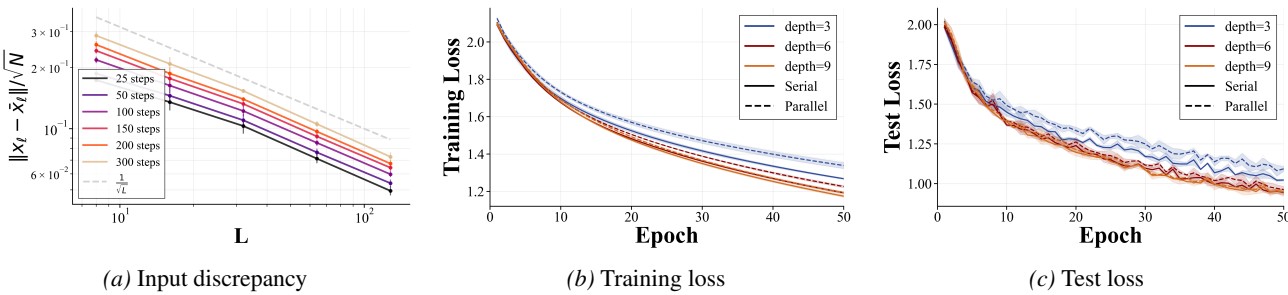

*(a)* Input discrepancy     *(b)* Training loss     *(c)* Test loss

*Figure 3.* **Parallelism between MLP and Attention.** (a) The discrepancy between attention-modulated and residual-only MLP inputs decays as $\mathcal{O}(L^{-1/2})$. (b-c) Serial blocks dominate at shallow regime, while parallelized blocks catch up as depth increases.

ysis, we omit normalization layers, since non-trainable normalizers (LayerNorm (Ba et al., 2016), RMSNorm(Zhang & Sennrich, 2019), QK-norm(Henry et al., 2020)) act as order-one variance rescalings in the mean-field limit and therefore do not change the depth orders of Gaussian, learning, or interaction terms, nor do they prevent learning collapse. Formal discussion appears in Section 3.5.

Analogous to the two-layer ResNet in Section 2, under $\mu$P scaling and any fixed number of SGD steps, all forward features and backward gradients in the Transformer are generated from prior TP variables via valid TP operations. Hence, training a Transformer with attention Eq. (5) and two-layer MLP blocks Eq. (1) is a valid TP, enabling us to characterize the feature-learning dynamics in the Transformer.

### 3.1. Active residual-stream Learning

We first state the NFD form for the residual stream. As in the warm-up, the residual update across depth admits a three-term decomposition with Gaussian innovations of size $L^{-1/2}$ and drift terms of size $L^{-1}$.

**Proposition 3.1** (Informal). *As $N \to \infty$, the coordinates of $(\boldsymbol{h}_{\ell,s}, \delta\boldsymbol{h}_{\ell,s})$ become asymptotically i.i.d. and satisfy,*

$$\Delta Z^{h_{\ell,s}(1)} = \underbrace{\frac{1}{\sqrt{L}} Z^{\widetilde{h}_{\ell,s}^{(1)}}}_{Gaussian} - \underbrace{\frac{\eta_o}{L}\mathcal{G}\left(Z^{\delta h_\ell^{(0)}}\right)}_{Learning} - \underbrace{\frac{\eta_v}{L}\mathcal{I}\left(Z^{\delta h_\ell^{(0)}}\right)}_{Interaction},$$

$$\Delta Z^{\delta h_{\ell,s}^{(1)}} = \frac{1}{\sqrt{L}} Z^{\delta \widetilde{h}_{\ell,s}^{(1)}} - \frac{\eta_{qkv}}{L}\mathcal{G}\left(Z^{h_\ell^{(0)}}\right) - \frac{\eta_{kqo}}{L}\mathcal{I}\left(Z^{h_\ell^{(0)}}\right),$$

*where $\Delta Z^{h_{\ell,s}^{(r)}} := Z^{h_{\ell,s}^{(r)}} - Z^{h_{\ell-1,s}^{(r)}}$, $\eta_{qkv} := \eta_q + \eta_k + \eta_v$, $(Z^{\widetilde{h}_{\ell,s}}, Z^{\delta\widetilde{h}_{\ell,s}^{(1)}})$ are centered Gaussian innovation terms, and $\mathcal{G}(\cdot)$ and $\mathcal{I}(\cdot)$ are linear functionals of the indicated arguments whose coefficients are mean-field kernels (e.g., $\mathbb{E}[Z^{o_{\ell,s'}} Z^{o_{\ell,s}}])$ multiplied by the limiting loss derivative $\mathcal{L}'$. Explicit formulas are given in Appendix D.*

**Residual-stream NFD.** Let $t = \ell/L \in [0, 1]$ and $\Delta t = 1/L$. Interpreting the recursions in Proposition 3.1 as an Euler–Maruyama scheme, the $L^{-1/2}$ Gaussian terms correspond to diffusion increments of size $\sqrt{\Delta t}$, while the $L^{-1}$ learning and interaction terms form the drift. Under regularity conditions analogous to those in the warm-up analysis, the residual-stream dynamics converge, as $N, L \to \infty$, to a forward–backward coupled SDE, which we refer to as the *residual-stream NFD*. Due to page limit, the full theorem and assumptions are deferred to Appendix D (see also Proposition 2.3). The key point is that, due to these depth orders, the residual stream remains in an *active learning* regime at large depth and the limiting drift retains contributions from both learning and interaction terms.

### 3.2. Suppressed internal representation learning

We now contrast the residual stream with the internal attention channels $(\boldsymbol{q}, \boldsymbol{k}, \boldsymbol{v})$. As in Section 2, these internal representations lack skip connections. Under $L^{-1/2}$ residual scaling, their gradient-driven learning terms appear at higher depth order and therefore vanish as $L \to \infty$.

**Proposition 3.2** (Informal). *As $N \to \infty$, the coordinate limits of the internal attention features satisfy*

$$Z^{\boldsymbol{q}_{\ell,s}^{(1)}} = \underbrace{Z^{\widetilde{\boldsymbol{q}}_{\ell,s}^{(1)}}}_{Gaussian} - \underbrace{\frac{\eta_q}{\sqrt{L}} \mathcal{G}\left(Z^{\delta \boldsymbol{q}_{\ell}^{(0)}}\right)}_{Learning} - \underbrace{\frac{\eta_o}{L^{3/2}} \mathcal{I}\left(Z^{\delta \boldsymbol{q}_{\ell}^{(0)}}\right)}_{Interaction},$$

$$Z^{\boldsymbol{k}_{\ell,s}^{(1)}} = Z^{\widetilde{\boldsymbol{k}}_{\ell,s}^{(1)}} - \frac{\eta_k}{\sqrt{L}} \mathcal{G}\left(Z^{\delta \boldsymbol{k}_{\ell}^{(0)}}\right) - \frac{\eta_o}{L^{3/2}} \mathcal{I}\left(Z^{\delta \boldsymbol{k}_{\ell}^{(0)}}\right),$$

$$Z^{\boldsymbol{v}_{\ell,s}^{(1)}} = Z^{\widetilde{\boldsymbol{v}}_{\ell,s}^{(1)}} - \frac{\eta_v}{\sqrt{L}} \mathcal{G}\left(Z^{\delta \boldsymbol{v}_{\ell}^{(0)}}\right) - \frac{\eta_o}{L^{3/2}} \mathcal{I}\left(Z^{\delta \boldsymbol{v}_{\ell}^{(0)}}\right),$$

*with backward dynamics*

$$Z^{\delta \bar{\boldsymbol{o}}_{\ell,s}^{(1)}} = Z^{\delta \widetilde{\boldsymbol{o}}_{\ell,s}^{(1)}} - \frac{\eta_o}{\sqrt{L}} \mathcal{G}\left(Z^{\boldsymbol{o}_{\ell}^{(0)}}\right) - \frac{\eta_{qkv}}{L^{3/2}} \mathcal{I}\left(Z^{\boldsymbol{o}_{\ell}^{(0)}}\right).$$

*where $\mathcal{G}., \mathcal{I}.$ are kernel-weighted linear functionals (explicit in Appendix D).*

In Proposition 3.2, the dominant learning term scales as $L^{-1/2}$, while the interaction term is further suppressed at $L^{-3/2}$. Without an additive skip connection, these terms do not accumulate across depth. Consequently,

$$\frac{\|\boldsymbol{q}_\ell - \widetilde{\boldsymbol{q}}_\ell\|}{\sqrt{N}} = \mathcal{O}\left(\frac{\eta_q}{\sqrt{L}}\right), \tag{6}$$

where $\widetilde{\boldsymbol{q}}_\ell$ denotes the frozen-weight baseline. Thus, internal attention learning is *asymptotically suppressed* as $L \to \infty$, while the residual stream continues to actively learn. Figure 1 empirically illustrates this suppression and its dependence on training steps.

**Proposition 3.3** (Informal). *As $N, L \to \infty$, the internal features converge to their Gaussian parts: $Z^{x_t^{(1)}} = Z^{\widetilde{x}_t^{(1)}}$, $\forall x \in \{q, k, v, \delta o\}$ with no contribution from learning and interaction components.*

Analogous to the internal features in MLPs, this collapse is also *silent*: residual-stream learning remains active, without any spike-like signals, while internal feature learning asymptotically collapses. Consequently, shallow and deep Transformers operate in distinct learning regimes. At shallow depth, both residual and internal representations learn effectively; at large depth, learning concentrates almost entirely in the residual stream. This mechanism explains the breakdown of hyperparameter transfer under depth-$\mu$P; see Table 1 for a summary of the resulting phase transition.

### 3.3. Attention collapse in the large-depth limit

Suppressed learning in $\boldsymbol{q}$ and $\boldsymbol{k}$ propagates directly to the attention logits. Since $\boldsymbol{W}_q$ and $\boldsymbol{W}_k$ are independently initialized and their learning contributions vanish with depth, the attention logits remain at their initialization scale.

**Proposition 3.4** (Informal). *As $N, L \to \infty$, the attention logits and scores satisfy $b_{\ell,ss'} = \mathcal{O}(\eta_{qk}/\sqrt{L})$, and $a_{\ell,ss'} =$*

$1/S + \mathcal{O}(b_{\ell,ss'})$, *respectively. In particular, the mean-field limits converge to uniform attention: $\mathring{b}_{\ell,ss'} = 0$ and $\mathring{a}_{\ell,ss'} = 1/S \ \forall s, s'$. Under a causal mask, $\mathring{a}_{\ell,ss'} = 1/s, \ \forall s' \le s$.*

Thus, in the infinite-depth limit, attention degenerates to uniform pooling and ceases to learn task-dependent token interactions. As illustrated in Figure 2, at finite depth this effect is gradual: selective attention emerges first in shallow layers and propagates to deeper layers over longer training horizons.

### 3.4. Emergent parallelism between MLP and attention

Depth-induced suppression also induces a *structural drift* in deep Transformers. Let $\boldsymbol{h}_\ell^{\text{attn}}$ be the attention output and define the MLP internal feature $\boldsymbol{x}_\ell = \boldsymbol{U}_\ell \boldsymbol{h}_\ell^{\text{attn}}$. Let $\bar{\boldsymbol{x}}_\ell = \boldsymbol{U}_\ell \boldsymbol{h}_{\ell-1}$ denote the residual-only counterpart. As depth grows, attention modulation becomes asymptotically negligible, and these two features coincide in the mean-field limit.

**Proposition 3.5** (Informal). *As $N, L \to \infty$, the corresponding mean-field limits satisfy $Z^{x_t} = Z^{\bar{x}_t}$, so the MLP and attention pathways become asymptotically parallel.*

Figure 3 provides numerical evidence: the discrepancy between $x_\ell^{attn}$ and $\bar{x}_\ell$ decays as $\mathcal{O}(L^{-1/2})$, and the performance gap between serial and parallel blocks narrows with depth. These results align with empirical observations of parallel Transformer blocks at large scale (Chowdhery et al., 2023; Dehghani et al., 2023).

### 3.5. Effect of Normalization on Depth Scaling

Modern Transformers rely on normalization layers (e.g., LayerNorm, RMSNorm, QK-norm) for numerical stability. For scaling analysis, we show that *non-trainable* normalizers act only as order-one variance rescalings, and thus do not alter the depth orders of the Gaussian, learning, or interaction terms in the NFD decomposition. Consequently, non-trainable normalization alone does not prevent the depth-induced suppression of internal feature learning identified in previous sections.

**Proposition 3.6** (Normalization Layers[2]). *Let $\boldsymbol{x}, \boldsymbol{q}, \boldsymbol{k} \in \mathbb{R}^N$ be TP variables with well-defined mean-field limits. As $N \to \infty$, the coordinates satisfy $Z^{\text{LN}(\boldsymbol{x})} = (Z^{\boldsymbol{x}} - \mu)/\sigma$ with $\mu = \mathbb{E}[Z^{\boldsymbol{x}}]$ and $\sigma^2 = \mathbb{E}[(Z^{\boldsymbol{x}} - \mu)^2]$, $Z^{\text{RMS}(\boldsymbol{x})} = Z^{\boldsymbol{x}}/\sqrt{\mathbb{E}[(Z^{\boldsymbol{x}})^2]}$, and $b_{ss'}$ converges in distribution to $\text{corr}(Z^{\boldsymbol{q}}, Z^{\boldsymbol{k}})$.*

Notably, Proposition 3.6 applies only to *non-trainable* nor-

---

[2]LayerNorm is defined by $\text{LN}(\boldsymbol{x}) := (\boldsymbol{x} - \mu_N \mathbf{1})/\sigma_N$, where $\mu_N = \frac{1}{N}\sum_i \boldsymbol{x}_i$ and $\sigma_N^2 = \frac{1}{N}\sum_i(\boldsymbol{x}_i - \mu_N)^2$. RMSNorm is defined by $\text{RMS}(\boldsymbol{x}) := \boldsymbol{x}/\sqrt{\frac{1}{N}\sum_i \boldsymbol{x}_i^2}$. QK-normalized logits are defined by $b_{ss'} := \langle \boldsymbol{q}_s, \boldsymbol{k}_{s'} \rangle/(\|\boldsymbol{q}_s\|_2 \|\boldsymbol{k}_{s'}\|_2)$.

malization. Trainable normalizers introduce additional parameters that follow the same NFD decomposition; consequently, their effect depends on their placement within the Transformer block (e.g., pre-norm vs. post-norm) and the corresponding depth orders, and must therefore be analyzed within the full coupled NFD framework.

# 4. Design Tradeoffs Revealed by Internal Learning Dynamics

The previous sections showed that, under depth-$\mu$P with $1/\sqrt{L}$ residual scaling, internal representations in both attention and MLP layers experience *silent* learning collapse at large depth: internal feature updates decay, while residual-stream learning remains active and stable. Through the NFD lens, we decomposed feature learning dynamics into Gaussian, learning, and interaction components with distinct depth orders, which makes the collapse mechanism explicit and quantitative.

Building on the same decomposition, this section explores depth-aware interventions that can restore internal feature learning *without* destabilizing residual dynamics. Our goal is not to propose a definitive remedy, but to use these quantitative constraints to characterize the tradeoffs among internal learning recovery, interaction growth, and residual-stream stability, and to distill principles for future scaling-aware architecture and optimizer design.

## 4.1. Depth-aware learning rates and interaction growth

A direct response to internal learning collapse is to increase learning rates (LRs) to offset depth-induced decay. In the two-layer ResNet setting, Proposition 2.4 together with Eq. 4 suggests that choosing $\eta_U = \Theta(\sqrt{L})$ restores order-one internal learning in the forward pass.

NFD also predicts, however, that the same scaling amplifies interaction effects in the residual stream. In particular, Proposition 2.2 shows that the interaction drift in $\Delta \boldsymbol{h}_\ell$ scales as $\eta_U/L = \Theta(L^{-1/2})$. Under an Euler–Maruyama view, this corresponds to an overly large per-layer drift that accumulates across depth, leading to

$$\|\boldsymbol{h}_L\|/\sqrt{N} = \mathcal{O}(\sqrt{L}).$$

Figure 1 empirically confirms this interaction-driven growth.

This analysis highlights a fundamental tradeoff: increasing LRs can restore internal feature learning, but it can simultaneously amplify interaction channels that destabilize the learning dynamics of other features via interaction channels.

## 4.2. Asymmetric learning rates (tied backward weights)

The instability induced by naive depth-aware scaling is due to *forward–backward coupling*: interaction terms arise because the same weight matrices are reused in the forward and backward passes. Consequently, increasing the learning rate to counteract internal learning collapse also amplifies coupling-induced interaction drift in other feature dynamics.

A natural way to address this coupling is to decouple backward weights from forward weights, enabling independent backward parameterizations and independent forward/backward learning rates. As a minimal step, this section keeps backward weights tied to forward weights while introducing *asymmetric* learning rates for forward and backward updates.

Concretely, we assign separate LRs $\widetilde{\eta}$ to backward updates while leaving the forward-side updates unchanged. In the Transformer setting, the corresponding mean-field limits in the backward pass become

$$Z^{\delta \bar{\boldsymbol{o}}^{(1)}_{\ell,s}} = Z^{\delta \widetilde{\boldsymbol{o}}^{(1)}_{\ell,s}} - \frac{\widetilde{\eta}_o}{\sqrt{L}} \, \mathcal{G}(Z^{\boldsymbol{o}^{(0)}_\ell}) - \frac{\widetilde{\eta}_{qkv}}{L^{3/2}} \, \mathcal{I}(Z^{\boldsymbol{o}^{(0)}_\ell}),$$

$$\Delta Z^{\delta h^{(1)}_{\ell,s}} = \frac{1}{\sqrt{L}} Z^{\delta \widetilde{h}^{(1)}_{\ell,s}} - \frac{\widetilde{\eta}_{qkv}}{L} \, \mathcal{G}(Z^{h^{(0)}_\ell}) - \frac{\widetilde{\eta}_{qkv}}{L} \, \mathcal{I}(Z^{h^{(0)}_\ell}).$$

Crucially, this asymmetry allows forward LRs $\eta_q$ and $\eta_k$ to scale as $\Theta(\sqrt{L})$ and to restore internal learning, without inducing order-$L^{-1/2}$ interaction drift in the backward residual stream $\Delta \delta \boldsymbol{h}_\ell$. Figure 4 shows that this restores nontrivial QK learning and attention structure across depth while maintaining stable residual magnitudes.

## 4.3. Decoupled backward weights

While asymmetric learning rates restore QK learning without destabilizing residual dynamics, they do not fully recover learning in all internal quantities, most notably the backward features $\delta \boldsymbol{o}_\ell$. NFD identifies the remaining limitation as interaction terms induced by *shared random initialization* between forward and backward weights, which persist even when forward and backward learning rates are decoupled.

To eliminate this interaction source, we decouple backward weight initialization while keeping gradient updates unchanged. Concretely, we replace $\boldsymbol{W}^{(0)\top}_{\ell,\boldsymbol{o}}$ with independently initialized backward weights $\widetilde{\boldsymbol{W}}^{(0)\top}_{\ell,\boldsymbol{o}}$. Because these interactions originate from initialization-induced correlations rather than gradient flow, this modification removes the coupling channel entirely. The resulting mean-field recursion simplifies to

$$\Delta Z^{\delta h^{(1)}_{\ell,s}} = \frac{1}{\sqrt{L}} Z^{\delta \widetilde{h}^{(1)}_{\ell,s}} - \frac{\widetilde{\eta}_{qkv}}{L} \, \mathcal{G}(Z^{h^{(0)}_\ell}),$$

which permits learning rates such as $\widetilde{\eta}_o = \Theta(\sqrt{L})$ to restore internal learning without inducing interaction-driven instability.

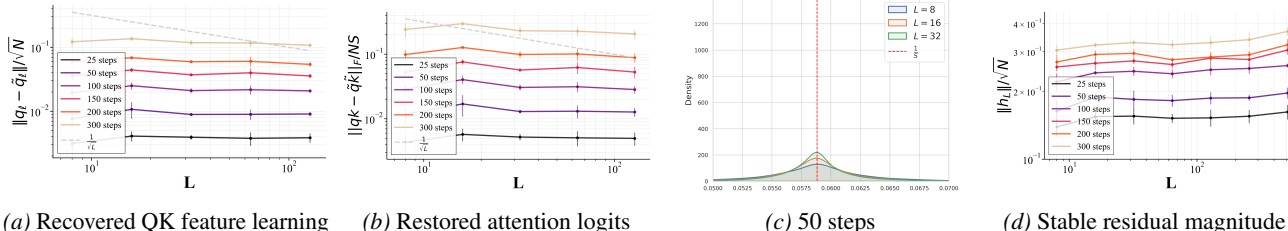

*(a)* Recovered QK feature learning    *(b)* Restored attention logits    *(c)* 50 steps    *(d)* Stable residual magnitude

*Figure 4.* **Asymmetric learning rates restore internal feature learning without inducing interaction-driven instability.** Using asymmetric learning rates guided by NFD analysis: (**a**) learning in query/key features is reactivated; (**b**) attention logits recover nontrivial structure; (**c**) selective attention patterns emerge across both shallow and deep layers; (**d**) the residual stream remains stable, avoiding the $\mathcal{O}(\sqrt{L})$ growth. These results illustrate how asymmetric LR effectively restores internal learning while preserving depth stability.

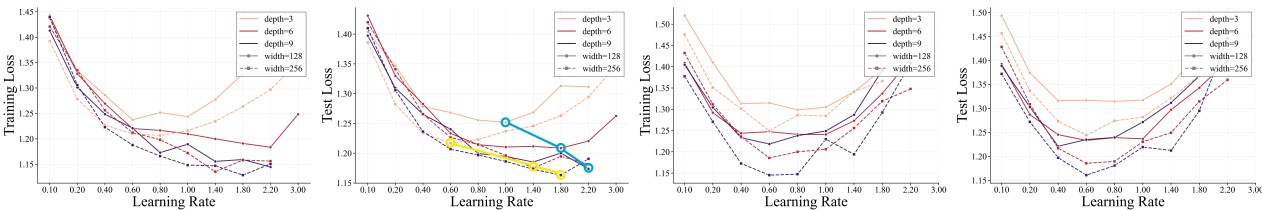

*Figure 5.* **Hyperparameter transfer reveals depth-dependent learning regimes.** Hyperparameter sweeps for ViT on CIFAR-10 trained with SGD. (**a,b**) Under vanilla depth-$\mu$P, the learning rate minimizing validation loss *increases* systematically with depth, and the optimal test performance shifts accordingly, due to depth-induced internal feature learning collapse. (**c,d**) Under the decoupled remedy, optimal learning rates align more consistently across depths, and learning rates selected on training loss remain near-optimal for test loss, reflecting improved hyperparameter transfer. While decoupling introduces mild performance degradation due to gradient misalignment, the resulting behavior is substantially more stable across depth, highlighting the role of NFD-guided design in restoring predictable scaling.

In principle, decoupled initialization yields the cleanest NFD limit with fully restored feature learning. In practice, however, it introduces *gradient misalignment*, which acts as structured optimization noise and can degrade empirical performance. This tradeoff is illustrated in Figure 5.

### 4.4. Hyperparameter transfer

We conclude by examining HP transfer across depth. Figure 5 compares depth-$\mu$P with decoupled variants on ViT models[3] trained on CIFAR-10. Under depth-$\mu$P, the learning rate minimizing loss shifts systematically toward larger values as depth increases, due to the depth-induced internal learning collapse. In contrast, decoupled methods largely align optimal learning rates across depth. Despite gradient misalignment, their performance remains competitive with depth-$\mu$P. We provide additional HP transfer for two-layer ResNets, including depth-aware LRs, in Appendix H. Although vanilla depth-aware LR theoretically amplifies interaction terms, it performs well at practical model scales, suggesting that interaction growth may be moderate and tolerable in finite-depth. Together, these results reinforce the central message of this section: quantitatively separating learning and interaction channels clarifies design tradeoffs and informs scaling-aware architecture and optimizer de-

sign.

## 5. Conclusion

We developed a training-time theory of feature learning dynamics for Transformers, termed *Neural Feature Dynamics* (NFD), which decomposes the evolution of each representation into Gaussian, learning, and interaction components with depth orders determined by architectural placement. This framework makes explicit how depth scaling reshapes training dynamics and reveals a depth-induced *learning shift* and *structural shift*: in deep regimes, internal representations undergo a *silent learning collapse* while residual-stream learning remains active without loss or gradient spikes, and serial attention–MLP compositions become effectively parallel. Using NFD, we analyzed interventions aimed at restoring internal learning, showing that asymmetric learning rates recover QK learning without preserving residual stability, whereas decoupled backward weights restore full internal learning at the cost of gradient misalignment. Together, these results provide a principled perspective on learning–interaction tradeoffs in deep Transformers and inform scaling-aware model and optimizer design, with extensions to trainable normalization layers, advanced optimizers, and long-horizon training dynamics left for future work.

---

[3]In the experiments, the ViT employs affine-free Pre-LayerNorm with a single attention head, trained for a maximum of 20 epochs per run.

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

## A. You *can* have an appendix here.

You can have as much text here as you want. The main body must be at most 8 pages long. For the final version, one more page can be added. If you want, you can use an appendix like this one.

The \onecolumn command above can be kept in place if you prefer a one-column appendix, or can be removed if you prefer a two-column appendix. Apart from this possible change, the style (font size, spacing, margins, page numbering, etc.) should be kept the same as the main body.

## B. Two-layer coupled case

In this paper, we consider a standard transformer consisting of a two-layer FNN and an attention block. We start our analysis with a two-layer FNN block:

$$\boldsymbol{h}_0 = \frac{1}{\sqrt{D}} \boldsymbol{E} \boldsymbol{x}$$

$$\boldsymbol{h}_\ell = \boldsymbol{h}_{\ell-1} + \frac{1}{\sqrt{LN}} \boldsymbol{W}_\ell \phi \left( \underbrace{\frac{1}{\sqrt{N}} \boldsymbol{U}_\ell \boldsymbol{h}_{\ell-1}}_{\boldsymbol{x}_\ell} \right), \qquad \forall \ell \in [L]$$

$$f(\boldsymbol{x}) = \frac{1}{N} \boldsymbol{v}^\top \boldsymbol{h}_L$$

where $\boldsymbol{x} \in \mathbb{R}^D$ is the embedding vector after the embedding layer, and its dimension does not change, and $\boldsymbol{x}_\ell$ is the **internal representation** at each layer. The weights are randomly initialized:

$$\boldsymbol{E}_{ij}, \ \boldsymbol{W}_{\ell,ij}, \ \boldsymbol{U}_{\ell,ij}, \ \boldsymbol{v}_i, \ \overset{i.i.d.}{\sim} \mathcal{N}(0,1).$$

Then, by using width-wise $\mu$P, the gradient propagation can be characterized as follows:

$$d\boldsymbol{h}_L = N \frac{\partial f}{\partial \boldsymbol{h}_L} = \boldsymbol{v}$$

$$d\boldsymbol{h}_{\ell-1} = N \frac{\partial f}{\partial \boldsymbol{h}_{\ell-1}} = d\boldsymbol{h}_\ell + \frac{1}{\sqrt{LN}} \boldsymbol{U}_\ell^\top d\boldsymbol{x}_\ell, \quad \ell \in [L],$$

where we denote

$$d\boldsymbol{x}_\ell = \frac{1}{\sqrt{N}} \boldsymbol{W}_\ell^\top d\boldsymbol{h}_\ell \odot \phi'(\boldsymbol{x}_\ell)$$

which is not $N \cdot \partial f / \partial \boldsymbol{x}_\ell$ but additionally scaled by $\sqrt{L}$.

Then, using the backward gradients, we can compute the

$$d\boldsymbol{v} = N \frac{\partial f}{\partial \boldsymbol{v}} = \boldsymbol{h}_L$$

$$d\boldsymbol{W}_\ell = N \frac{\partial f}{\partial \boldsymbol{W}_\ell} = \frac{1}{\sqrt{LN}} d\boldsymbol{h}_\ell \phi(\boldsymbol{x}_\ell)^\top$$

$$d\boldsymbol{U}_\ell = N \frac{\partial L}{\partial \boldsymbol{U}_\ell} = \frac{1}{\sqrt{LN}} d\boldsymbol{x}_\ell \boldsymbol{h}_{\ell-1}^\top = \frac{1}{\sqrt{LN}} \left[ \frac{1}{\sqrt{N}} \boldsymbol{W}_\ell^\top d\boldsymbol{h}_\ell \odot \phi'(\boldsymbol{x}_\ell) \right] \boldsymbol{h}_{\ell-1}^\top$$

$$d\boldsymbol{E} = \frac{1}{\sqrt{D}} d\boldsymbol{h}_0 \boldsymbol{x}^\top$$

As a result, the gradient descents are given by

$$\boldsymbol{v}^+ = \boldsymbol{v} - \eta_v \mathcal{L}' \cdot \frac{1}{N} d\boldsymbol{v} = \boldsymbol{v} - \eta_v' \mathcal{L}' \cdot \boldsymbol{h}_L$$

$$\boldsymbol{W}_\ell^+ = \boldsymbol{W}_\ell - \eta_W \mathcal{L}' \cdot \frac{1}{N} d\boldsymbol{W}_\ell = \boldsymbol{W}_\ell - \eta_W' \mathcal{L}' \frac{1}{\sqrt{LN}} d\boldsymbol{h}_\ell \phi(\boldsymbol{x}_\ell)^\top$$

$$\boldsymbol{U}_\ell^+ = \boldsymbol{U}_\ell - \eta_U \mathcal{L}' \cdot \frac{1}{N} d\boldsymbol{U}_\ell = \boldsymbol{U}_\ell - \eta_U' \mathcal{L}' \frac{1}{\sqrt{LN}} d\boldsymbol{x}_\ell \boldsymbol{h}_{\ell-1}^\top$$

$$\boldsymbol{E}_\ell^+ = \boldsymbol{E}_\ell - \eta_E \mathcal{L}' \cdot \frac{1}{N} d\boldsymbol{E}_\ell = \boldsymbol{E}_\ell - \eta_E' \mathcal{L}' \frac{1}{\sqrt{D}} d\boldsymbol{h}_0 \boldsymbol{x}^\top,$$

where $\eta_v = \eta_v' N$ and the same cases are applied to $\eta_U, \eta_W, \eta_E$.

### B.1. First-Forward Pass

As the entries in matrix $\boldsymbol{E}$ are all iid Gaussian, the coordinates of $\boldsymbol{h}_0$ are governed by the mean-field random variable $Z^{\boldsymbol{h}_0} \sim \mathcal{N}(0, \|\boldsymbol{x}\|^2/D)$. For the hidden layers, we can inductively hypothesis that the coordinates of $\boldsymbol{h}_{\ell-1}$ are governed by an iid (Gaussian) random variable $Z^{\boldsymbol{h}_{\ell-1}}$, then the iid Gaussian entries in $\boldsymbol{U}_\ell$ imply the coordinates of $\boldsymbol{x}_\ell$ are also roughly iid Gaussian $Z^{\boldsymbol{x}_\ell} \sim \mathcal{N}(0, \mathbb{E}[(Z^{\boldsymbol{h}_{\ell-1}})^2])$. Furthermore, the iid entries in $\boldsymbol{W}_\ell$ with scaling $1/\sqrt{N}$ implies the vector $\frac{1}{\sqrt{N}} \boldsymbol{W}_\ell \phi(\boldsymbol{x}_\ell)$ converges to iid $Z^{\boldsymbol{W}_\ell \phi(\boldsymbol{x}_\ell)} \sim \mathcal{N}(0, \mathbb{E}[\phi(Z^{\boldsymbol{x}_\ell})^2])$. As $\boldsymbol{h}_{\ell-1}$ and $\frac{1}{\sqrt{N}} \boldsymbol{W}_\ell \phi(\boldsymbol{x}_\ell)$ all have asymptotically iid coordinates, so does $\boldsymbol{h}_\ell$. Hence, in the large-width limit, the forward propagation is governed by the following SDE:

$$Z^{\boldsymbol{h}_0} \sim \mathcal{N}(0, \|\boldsymbol{x}\|^2/D)$$

$$Z^{\boldsymbol{h}_\ell} = Z^{\boldsymbol{h}_{\ell-1}} + \frac{1}{\sqrt{L}} Z^{\boldsymbol{W}_\ell \phi(\boldsymbol{x}_\ell)}, \quad Z^{\boldsymbol{W}_\ell \phi(\boldsymbol{x}_\ell)} = \mathcal{N}(0, \mathbb{E}[\phi(Z^{\boldsymbol{x}_\ell})^2]), \quad Z^{\boldsymbol{x}_\ell} \sim \mathcal{N}(0, \mathbb{E}[(Z^{\boldsymbol{h}_{\ell-1}})^2])$$

$$\mathring{f}(\boldsymbol{x}) = \mathbb{E}[Z^{\boldsymbol{v}} Z^{\boldsymbol{h}_L}], \quad Z^{\boldsymbol{v}} \sim \mathcal{N}(0, 1),$$

where $\{Z^{\boldsymbol{W}_\ell \phi(\boldsymbol{x}_\ell)}, Z^{\boldsymbol{x}_\ell}\}_\ell$ are mutually independent Gaussians.

### B.2. First-Backward Pass

By the random initialization, the coordinates of $d\boldsymbol{h}_L$ follow $Z^{d\boldsymbol{h}_L} \sim \mathcal{N}(0, 1)$. For the hidden layers, we can inductively show that as the vector $d\boldsymbol{h}_\ell$ has roughly iid coordinates following $Z^{d\boldsymbol{h}_\ell}$, then the vector $\frac{1}{\sqrt{N}} \boldsymbol{W}_\ell^\top d\boldsymbol{h}_\ell$ has roughly iid Gaussian $Z^{\boldsymbol{W}_\ell^\top d\boldsymbol{h}_\ell} \sim \mathcal{N}(0, \mathbb{E}[(Z^{d\boldsymbol{h}_\ell})^2])$. Then we denote

$$\frac{1}{\sqrt{N}} \boldsymbol{W}_\ell^\top d\boldsymbol{h}_\ell \odot \phi'(\boldsymbol{x}_\ell) = \sqrt{L} d\boldsymbol{x}_\ell := d\hat{\boldsymbol{x}}_\ell \sim Z^{d\hat{\boldsymbol{x}}_\ell} = Z^{\boldsymbol{W}_\ell^\top d\boldsymbol{h}_\ell} \phi(Z^{\boldsymbol{x}_\ell}).$$

Here, we can see the $d\hat{\boldsymbol{x}}_\ell$ is different from $d\boldsymbol{x}_\ell$ since the coordinates of $d\hat{\boldsymbol{x}}_\ell$ are all $\Theta(1)$ in depth $L$, while $d\boldsymbol{x}_\ell$'s coordinates have $\Theta(L^{-\frac{1}{2}})$. Then in the large-width limit, the coordinates of $d\hat{\boldsymbol{x}}_\ell$ are roughly iid Gaussians $Z^{d\hat{\boldsymbol{x}}_\ell} = Z^{\boldsymbol{W}_\ell^\top d\boldsymbol{h}_\ell} \phi'(Z^{\boldsymbol{x}_\ell})$. Then we have the vector $\frac{1}{\sqrt{N}} \boldsymbol{U}_\ell^\top d\hat{\boldsymbol{x}}_\ell$ has iid Gaussian coordinates $Z^{\boldsymbol{U}_\ell d\boldsymbol{x}_\ell} \sim \mathcal{N}(0, \mathbb{E}[Z^{\boldsymbol{W}_\ell^\top d\boldsymbol{h}_\ell} \phi'(Z^{\boldsymbol{x}_\ell})]^2)$. Therefore, the first backward pass can be characterized as follows:

$$Z^{d\boldsymbol{h}_L} \sim \mathcal{N}(0, 1)$$

$$Z^{d\boldsymbol{h}_{\ell-1}} = Z^{d\boldsymbol{h}_\ell} + \frac{1}{\sqrt{L}} Z^{\boldsymbol{U}^\top d\boldsymbol{x}_\ell},$$

where

$$Z^{\boldsymbol{U}^\top d\boldsymbol{x}_\ell} \sim \mathcal{N}(0, \mathbb{E}[Z^{d\boldsymbol{x}_\ell}]^2), \quad Z^{d\boldsymbol{x}_\ell} = Z^{\boldsymbol{W}_\ell^\top d\boldsymbol{h}_\ell} \phi'(Z^{\boldsymbol{x}_\ell}) \quad Z^{\boldsymbol{W}_\ell^\top d\boldsymbol{h}_\ell} \sim \mathcal{N}(0, \mathbb{E}[(Z^{d\boldsymbol{h}_\ell})^2])$$

*Remark* B.1. The forward and backward passes are driven naturally by independent Brownian motions, *i.e.*, $\{\boldsymbol{W}_\ell\}_\ell$ and $\{\boldsymbol{U}_\ell\}_\ell$. Hence, in the following studies of their multiple passes, the key is to describe their interaction terms and whether they vanish in the large-depth limit.

### B.3. Second-Forward Pass

After one-step gradient descent, the embedding layer is updated as

$$\bar{\boldsymbol{h}}_0 = \frac{1}{\sqrt{D}}(\boldsymbol{E}(0) + \Delta\boldsymbol{E}(0))\bar{\boldsymbol{x}}$$

$$= \frac{1}{\sqrt{D}}\boldsymbol{E}(0)\bar{\boldsymbol{x}} - \eta'_E \mathcal{L}' \frac{\langle \boldsymbol{x}, \bar{\boldsymbol{x}}\rangle}{D} \cdot d\boldsymbol{h}_0$$

Hence, in the large-width limit, the coordinates of $\bar{\boldsymbol{h}}_0$ is governed by

$$Z^{\bar{\boldsymbol{h}}_0} \sim Z^{\boldsymbol{E}\bar{\boldsymbol{x}}} - \eta'_E \mathring{\mathcal{L}}' \frac{\langle \boldsymbol{x}, \bar{\boldsymbol{x}}\rangle}{D} Z^{d\boldsymbol{h}_0}, \quad Z^{\boldsymbol{E}\bar{\boldsymbol{x}}} \sim \mathcal{N}(0, \|\bar{x}\|^2/D)$$

where $Z^{\boldsymbol{E}\bar{\boldsymbol{x}}}$ is correlated with $Z^{\boldsymbol{h}_0} = Z^{\boldsymbol{E}\boldsymbol{x}}$ with covariance

$$\mathbb{E}[Z^{\boldsymbol{E}\bar{\boldsymbol{x}}} Z^{\boldsymbol{E}\boldsymbol{x}}] = \langle \boldsymbol{x}, \bar{\boldsymbol{x}}\rangle / D.$$

For the hidden state, we first characterize the update for the internal representation $\bar{\boldsymbol{x}}_\ell$:

$$\bar{\boldsymbol{x}}_\ell = \frac{1}{\sqrt{N}}\left(\boldsymbol{U}_\ell(0) + \Delta\boldsymbol{U}_\ell(0)\right)\bar{\boldsymbol{h}}_{\ell-1}$$

$$= \frac{1}{\sqrt{N}}\boldsymbol{U}_\ell(0)\bar{\boldsymbol{h}}_{\ell-1} - \frac{1}{\sqrt{L}}\eta'_U \mathcal{L}' \frac{\langle \boldsymbol{h}_{\ell-1}, \bar{\boldsymbol{h}}_{\ell-1}\rangle}{N} d\boldsymbol{x}_\ell.$$

The interaction term occurs in

$$\frac{1}{\sqrt{N}}\boldsymbol{U}_\ell\bar{\boldsymbol{h}}_{\ell-1} = \frac{1}{\sqrt{N}}\boldsymbol{U}_\ell\left(\bar{\boldsymbol{h}}_{\ell-2} + \frac{1}{\sqrt{LN}}\left(\boldsymbol{W}_{\ell-1} + \Delta\boldsymbol{W}_{\ell-1}\right)\phi(\bar{\boldsymbol{x}}_{\ell-1})\right)$$

$$\sim \frac{1}{\sqrt{N}}\boldsymbol{U}_\ell \cdot \frac{1}{\sqrt{LN}}\left[-\eta'_W \mathcal{L}' \frac{1}{\sqrt{LN}} d\boldsymbol{h}_{\ell-1}\phi(\boldsymbol{x}_{\ell-1})^\top\right]\phi(\bar{\boldsymbol{x}}_{\ell-1})$$

$$= -\frac{1}{L}\eta'_W \mathcal{L}' \frac{\langle \phi(\boldsymbol{x}_{\ell-1}), \phi(\bar{\boldsymbol{x}}_{\ell-1})\rangle}{N}\frac{1}{\sqrt{N}}\boldsymbol{U}_\ell d\boldsymbol{h}_{\ell-1}$$

$$= -\frac{1}{L}\eta'_W \mathcal{L}' \frac{\langle \phi(\boldsymbol{x}_{\ell-1}), \phi(\bar{\boldsymbol{x}}_{\ell-1})\rangle}{N}\frac{1}{\sqrt{N}}\boldsymbol{U}_\ell\left[d\boldsymbol{h}_\ell + \frac{1}{\sqrt{LN}}\boldsymbol{U}_\ell^\top d\hat{\boldsymbol{x}}_\ell\right]$$

$$\sim -L^{-\frac{3}{2}}\eta'_W \mathcal{L}' \frac{\langle \phi(\boldsymbol{x}_{\ell-1}), \phi(\bar{\boldsymbol{x}}_{\ell-1})\rangle}{N}\left[\frac{1}{N}\boldsymbol{U}_\ell\boldsymbol{U}_\ell^\top d\hat{\boldsymbol{x}}_\ell\right]$$

$$\sim -L^{-\frac{3}{2}}\eta'_W \mathring{\mathcal{L}}' \mathbb{E}[\phi(Z^{\boldsymbol{x}_{\ell-1}})\phi(Z^{\bar{\boldsymbol{x}}_{\ell-1}})]Z^{d\hat{\boldsymbol{x}}_\ell}$$

Hence, as the width tends to infinity, the master theorem from TP yields

$$\boxed{Z^{\bar{\boldsymbol{x}}_\ell} = \underbrace{Z^{\boldsymbol{U}\bar{\boldsymbol{h}}_{\ell-1}}}_{\text{Gaussian}} - \underbrace{\frac{1}{\sqrt{L}}\eta'_U \mathring{\mathcal{L}}' \mathbb{E}[Z^{\boldsymbol{h}_{\ell-1}} Z^{\bar{\boldsymbol{h}}_{\ell-1}}] Z^{\boldsymbol{W}_\ell^\top d\boldsymbol{h}_\ell}\phi'(Z^{\boldsymbol{x}_\ell})}_{\text{GD update}} - \underbrace{L^{-\frac{3}{2}}\eta'_W \mathring{\mathcal{L}}' \mathbb{E}[\phi(Z^{\boldsymbol{x}_{\ell-1}})\phi(Z^{\bar{\boldsymbol{x}}_{\ell-1}})]Z^{d\hat{\boldsymbol{x}}_\ell}}_{\text{interaction}}.} \quad (7)$$

Remarkably, GD update term on the RHS is $\Theta(L^{-\frac{1}{2}})$ in depth $L$; hence, in the large-depth limit, the gradient update term becomes negligible if $\eta'_U = \Theta(1)$ in depth $L$ too, indicating the internal layer does not refine representation but just passes the features. Hence, to recover effective feature learning, it is important to have

$$\eta'_U = \eta_c\sqrt{L} \implies \eta_U = \eta_c\sqrt{L}N.$$

Similarly, the expression of $\bar{\boldsymbol{h}}_\ell$ can be expanded by

$$\bar{\boldsymbol{h}}_\ell = \bar{\boldsymbol{h}}_{\ell-1} + \frac{1}{\sqrt{LN}}\left(\boldsymbol{W}_\ell(0) + \Delta\boldsymbol{W}_\ell(0)\right)\phi(\bar{\boldsymbol{x}}_\ell)$$

$$= \bar{\boldsymbol{h}}_{\ell-1} + \frac{1}{\sqrt{LN}}\boldsymbol{W}_\ell(0)\phi(\bar{\boldsymbol{x}}_\ell) - \frac{1}{L}\eta'_W \mathcal{L}' \frac{\langle \phi(\boldsymbol{x}_\ell), \phi(\bar{\boldsymbol{x}}_\ell)\rangle}{N} d\boldsymbol{h}_\ell.$$

The interaction term also occurs in the update of $\bar{\boldsymbol{h}}_{\ell-1}$. For simplicity, assume $\phi = \mathrm{id}$, then

$$
\begin{aligned}
\frac{1}{\sqrt{LN}}\boldsymbol{W}_\ell \bar{\boldsymbol{x}}_\ell &= \frac{1}{\sqrt{LN}}\boldsymbol{W}_\ell \cdot \frac{1}{\sqrt{N}}(\boldsymbol{U}_\ell + \Delta \boldsymbol{U}_\ell)\bar{\boldsymbol{h}}_{\ell-1} \\
&\sim \frac{1}{\sqrt{LN}}\boldsymbol{W}_\ell \cdot \frac{1}{\sqrt{N}}\left[ -\eta'_U \mathcal{L}' \frac{1}{\sqrt{LN}} \cdot \frac{1}{\sqrt{N}}\boldsymbol{W}_\ell^\top d\boldsymbol{h}_\ell \boldsymbol{h}_{\ell-1}^\top \right]\bar{\boldsymbol{h}}_{\ell-1} \\
&= -\frac{1}{L}\eta'_U \mathcal{L}' \frac{\boldsymbol{h}_{\ell-1}^\top \bar{\boldsymbol{h}}_{\ell-1}}{N}\left[ \frac{1}{N}\boldsymbol{W}_\ell \boldsymbol{W}_\ell^\top d\boldsymbol{h}_\ell \right] \\
&\sim -\frac{1}{L}\eta'_U \mathring{\mathcal{L}}' \mathbb{E}[Z^{\boldsymbol{h}_{\ell-1}} Z^{\bar{\boldsymbol{h}}_{\ell-1}}] Z^{d\boldsymbol{h}_\ell}
\end{aligned}
$$

where we consider the $\alpha$-th coordinate

$$
\begin{aligned}
\frac{1}{N}\sum_\beta \boldsymbol{W}_{\ell,\alpha\beta}(\boldsymbol{W}_\ell^\top d\boldsymbol{h}_\ell)_\beta &= \frac{1}{N}\sum_\beta \boldsymbol{W}_{\ell,\alpha\beta}\sum_\gamma \boldsymbol{W}_{\ell,\gamma\beta}d\boldsymbol{h}_{\ell,\gamma} \\
&= \left( \frac{1}{N}\sum_\beta \boldsymbol{W}_{\ell,\alpha\beta}^2 \right)d\boldsymbol{h}_{\ell,\alpha} + \frac{1}{N}\sum_\beta \boldsymbol{W}_{\ell,\alpha\beta}\sum_{\gamma\neq\alpha}\boldsymbol{W}_{\ell,\gamma\beta}d\boldsymbol{h}_{\ell,\gamma}
\end{aligned}
$$

and then

$$
Z^{\bar{\boldsymbol{h}}_\ell} = Z^{\bar{\boldsymbol{h}}_{\ell-1}} + \underbrace{\frac{1}{\sqrt{L}}Z^{\boldsymbol{W}_\ell \bar{\boldsymbol{x}}_\ell}}_{\text{Gaussian}} - \underbrace{\frac{1}{L}\eta'_W \mathring{\mathcal{L}}' \mathbb{E}[Z^{\boldsymbol{x}_\ell} Z^{\bar{\boldsymbol{x}}_\ell}]Z^{d\boldsymbol{h}_\ell}}_{\text{GD}} - \underbrace{\frac{1}{L}\eta'_U \mathring{\mathcal{L}}' \mathbb{E}[Z^{\boldsymbol{h}_{\ell-1}} Z^{\bar{\boldsymbol{h}}_{\ell-1}}]Z^{d\boldsymbol{h}_\ell}}_{\text{interaction}}
$$

Hence, if we set $\eta'_U = \eta_c \sqrt{L}$, then the interaction term will blow up, thus, in general case, we can only use $\eta'_U = \eta_c$, which cause the feature learning collapse in $\boldsymbol{x}_\ell$. However, if we ensure the backward weight $\boldsymbol{W}_\ell^\top$ are independent from the forward weight, that *manually* for the interaction to be vanished. Hence, we recover the effective feature learning in the first layer and also solve the blowing up problem.

Similarly, we can obtain the form for a general $\phi$:

$$
\begin{aligned}
\frac{1}{\sqrt{LN}}W_\ell \phi(\bar{x}_\ell) &= \frac{1}{\sqrt{LN}}W_\ell \phi\left( \frac{1}{\sqrt{N}}U_\ell \bar{h}_{\ell-1} + \frac{1}{\sqrt{N}}\Delta U_\ell \bar{h}_{\ell-1} \right) \\
&\sim \frac{1}{\sqrt{LN}}W_\ell \left[ \phi(\hat{x}_\ell) + \phi'(\hat{x}_\ell)\odot \frac{1}{\sqrt{N}}\Delta U_\ell \bar{h}_{\ell-1} \right] \\
&\sim \frac{1}{\sqrt{LN}}W_\ell \left[ \phi'(\hat{x}_\ell)\odot -\frac{1}{\sqrt{LN}}\eta'_U \mathcal{L}' \frac{\langle h_{\ell-1}, \bar{h}_{\ell-1}\rangle}{N}W_\ell^\top dh_\ell \odot \phi'(x_\ell) \right]
\end{aligned}
$$

where $\hat{x}_\ell = \frac{1}{\sqrt{N}}U_\ell \bar{h}_{\ell-1}$. By considering the $\alpha$-coordinate, we can identify the interaction term:

$$
-\frac{1}{L}\eta'_U \mathcal{L}' \frac{\langle h_{\ell-1}, \bar{h}_{\ell-1}\rangle}{N}\left[ \frac{1}{N}\sum_\beta W_{\ell,\alpha\beta}^2 \phi'(\hat{x}_{\ell,\beta})\phi'(x_{\ell,\beta}) \right]dh_{\ell,\alpha}
$$

Hence, the characterization of $\bar{\boldsymbol{h}}_\ell$ becomes

$$
\boxed{Z^{\bar{\boldsymbol{h}}_\ell} = Z^{\bar{\boldsymbol{h}}_{\ell-1}} + \underbrace{\frac{1}{\sqrt{L}}Z^{W_\ell(0)\phi(\bar{x}_\ell)}}_{\text{Gaussian}} - \underbrace{\frac{1}{L}\eta'_W \mathring{\mathcal{L}}' \mathbb{E}[\phi(Z^{x_\ell})\phi(Z^{\bar{x}_\ell})]Z^{dh_\ell}}_{\text{GD}} - \underbrace{\frac{1}{L}\eta'_U \mathring{\mathcal{L}}' \mathbb{E}[Z^{h_{\ell-1}} Z^{\bar{h}_{\ell-1}}]\mathbb{E}[\phi'(Z^{\bar{x}_\ell})\phi'(Z^{x_\ell})]Z^{dh_\ell}}_{\text{interaction}}.}
$$

(8)

where $\eta_U = \eta'_U N$ and $\eta'_U$ depends on us.

### B.4. Second-Backward Pass

We have

$$d\bar{h}_L = (v + \Delta v) = v - \eta_v \mathcal{L}' h_L \implies Z^{d\bar{h}_L} = Z^v - \eta_v \mathring{\mathcal{L}}' Z^{h_L}.$$

Let us first consider $d\bar{\boldsymbol{x}}_\ell$'s update:

$$d\bar{\boldsymbol{x}}_\ell = \frac{1}{\sqrt{N}} (\boldsymbol{W}_\ell + \Delta \boldsymbol{W}_\ell)^\top d\bar{\boldsymbol{h}}_\ell \odot \phi'(\bar{\boldsymbol{x}}_\ell)$$

$$= \frac{1}{\sqrt{N}} (\boldsymbol{W}_\ell - \eta'_W \mathcal{L}' \frac{1}{\sqrt{LN}} d\boldsymbol{h}_\ell \phi(\boldsymbol{x}_\ell)^\top)^\top d\bar{\boldsymbol{h}}_\ell \odot \phi'(\bar{\boldsymbol{x}}_\ell)$$

$$= \left[ \frac{1}{\sqrt{N}} \boldsymbol{W}_\ell^\top d\bar{\boldsymbol{h}}_\ell - \frac{1}{\sqrt{L}} \eta'_W \mathcal{L}' \frac{\langle d\bar{\boldsymbol{h}}_\ell, d\boldsymbol{h}_\ell \rangle}{N} \phi(\boldsymbol{x}_\ell) \right] \odot \phi'(\bar{\boldsymbol{x}}_\ell)$$

Before writing the mean-field limit, we need to first figure out the interaction term raised in

$$\frac{1}{\sqrt{N}} \boldsymbol{W}_\ell^\top(0) d\bar{\boldsymbol{h}}_\ell = \frac{1}{\sqrt{N}} \boldsymbol{W}_\ell(0)^\top \left( d\bar{\boldsymbol{h}}_{\ell+1} + \frac{1}{\sqrt{LN}} (\boldsymbol{U}_{\ell+1} + \Delta \boldsymbol{U}_{\ell+1})^\top d\bar{\boldsymbol{x}}_{\ell+1} \right)$$

$$\sim \frac{1}{\sqrt{N}} \boldsymbol{W}_\ell(0)^\top \cdot \frac{1}{\sqrt{LN}} \Delta \boldsymbol{U}_{\ell+1}^\top d\bar{\boldsymbol{x}}_{\ell+1}$$

$$= \frac{1}{\sqrt{N}} \boldsymbol{W}_\ell(0)^\top \cdot \frac{1}{\sqrt{LN}} \left( -\eta'_U \mathcal{L}' \frac{1}{\sqrt{LN}} d\boldsymbol{x}_{\ell+1} \boldsymbol{h}_\ell^\top \right)^\top d\bar{\boldsymbol{x}}_{\ell+1}$$

$$= -\frac{1}{L} \eta'_U \mathcal{L}' \frac{\langle d\boldsymbol{x}_{\ell+1}, d\bar{\boldsymbol{x}}_{\ell+1} \rangle}{N} \frac{1}{\sqrt{N}} \boldsymbol{W}_\ell(0)^\top \boldsymbol{h}_\ell$$

$$= -\frac{1}{L} \eta'_U \mathcal{L}' \frac{\langle d\boldsymbol{x}_{\ell+1}, d\bar{\boldsymbol{x}}_{\ell+1} \rangle}{N} \frac{1}{\sqrt{N}} \boldsymbol{W}_\ell(0)^\top \left( \boldsymbol{h}_{\ell-1} + \frac{1}{\sqrt{LN}} \boldsymbol{W}_\ell(0) \phi(\boldsymbol{x}_\ell) \right)$$

$$\sim -L^{-\frac{3}{2}} \eta'_U \mathcal{L}' \frac{\langle d\boldsymbol{x}_{\ell+1}, d\bar{\boldsymbol{x}}_{\ell+1} \rangle}{N} \frac{1}{N} \boldsymbol{W}_\ell(0)^\top \boldsymbol{W}_\ell(0) \phi(\boldsymbol{x}_\ell)$$

$$\sim -L^{-\frac{3}{2}} \eta'_U \mathring{\mathcal{L}}' \mathbb{E}[Z^{d\boldsymbol{x}_{\ell+1}} Z^{d\bar{\boldsymbol{x}}_{\ell+1}}] \phi(Z^{\boldsymbol{x}_\ell})$$

where we focus only on the terms that raise the interaction term, while dropping the terms that do not. Hence, the mean-field characterization is given by

$$Z^{d\bar{\boldsymbol{x}}_\ell} = \left[ \underbrace{Z^{\boldsymbol{W}_\ell^\top d\bar{\boldsymbol{h}}_\ell}}_{\text{Gaussian}} - \underbrace{\frac{1}{\sqrt{L}} \eta'_W \mathring{\mathcal{L}}' \mathbb{E}[Z^{d\bar{\boldsymbol{h}}_\ell} Z^{d\boldsymbol{h}_\ell}] \phi(Z^{\boldsymbol{x}_\ell})}_{\text{Learning}} - \underbrace{L^{-\frac{3}{2}} \eta'_U \mathring{\mathcal{L}}' \mathbb{E}[Z^{d\boldsymbol{x}_{\ell+1}} Z^{d\bar{\boldsymbol{x}}_{\ell+1}}] \phi(Z^{\boldsymbol{x}_\ell})}_{\text{interaction}} \right] \phi'(Z^{\bar{\boldsymbol{x}}_\ell}) \quad (9)$$

Critically, as depth $L$ tends to infinity, the learning term vanishes at a rate of $L^{-\frac{1}{2}}$. Hence, to resolve this problem, we need to set

$$\eta'_W = \eta_c \sqrt{L} \implies \eta_W = \eta_c \sqrt{L} N$$

Now consider

$$d\bar{\boldsymbol{h}}_{\ell-1} = d\bar{\boldsymbol{h}}_\ell + \frac{1}{\sqrt{LN}} (\boldsymbol{U}_\ell(0) + \Delta \boldsymbol{U}_\ell(0))^\top d\bar{\boldsymbol{x}}_\ell$$

$$= d\bar{\boldsymbol{h}}_\ell + \frac{1}{\sqrt{LN}} \boldsymbol{U}_\ell(0)^\top d\bar{\boldsymbol{x}}_\ell + \frac{1}{\sqrt{LN}} \left( -\eta'_U \mathcal{L}' \frac{1}{\sqrt{LN}} d\boldsymbol{x}_\ell \boldsymbol{h}_{\ell-1}^\top \right)^\top d\bar{\boldsymbol{x}}_\ell$$

$$= d\bar{\boldsymbol{h}}_\ell + \frac{1}{\sqrt{LN}} \boldsymbol{U}_\ell(0)^\top d\bar{\boldsymbol{x}}_\ell - \frac{1}{L} \eta'_U \mathcal{L}' \frac{\langle d\boldsymbol{x}_\ell, d\bar{\boldsymbol{x}}_\ell \rangle}{N} \boldsymbol{h}_{\ell-1}$$

The interaction term stems from

$$\frac{1}{\sqrt{LN}}\boldsymbol{U}_\ell^\top d\bar{\boldsymbol{x}}_\ell = \frac{1}{\sqrt{LN}}\boldsymbol{U}_\ell^\top \left[\frac{1}{\sqrt{N}}(\boldsymbol{W}_\ell + \Delta\boldsymbol{W}_\ell)^\top d\bar{\boldsymbol{h}}_\ell \odot \phi'(\bar{\boldsymbol{x}}_\ell)\right]$$

$$\sim \frac{1}{\sqrt{LN}}\boldsymbol{U}_\ell^\top \left[\frac{1}{\sqrt{N}}\left(-\eta'_W\mathcal{L}'\frac{1}{\sqrt{LN}}d\boldsymbol{h}_\ell\phi(\boldsymbol{x}_\ell)^\top\right)d\bar{\boldsymbol{h}}_\ell \odot \phi'(\bar{\boldsymbol{x}}_\ell)\right]$$

$$= -\frac{1}{L}\eta'_W\mathcal{L}'\frac{\langle d\boldsymbol{h}_\ell, d\bar{\boldsymbol{h}}_\ell\rangle}{N}\frac{1}{\sqrt{N}}\boldsymbol{U}_\ell^\top\left[\phi(\boldsymbol{x}_\ell)\odot\phi'(\bar{\boldsymbol{x}}_\ell)\right]$$

$$= -\frac{1}{L}\eta'_W\mathcal{L}'\frac{\langle d\boldsymbol{h}_\ell, d\bar{\boldsymbol{h}}_\ell\rangle}{N}\frac{1}{\sqrt{N}}\boldsymbol{U}_\ell^\top\left[\phi\left(\frac{1}{\sqrt{N}}\boldsymbol{U}_\ell\boldsymbol{h}_{\ell-1}\right)\odot\phi'(\bar{\boldsymbol{x}}_\ell)\right].$$

By focusing on the $\alpha$-th coordinate, we have

$$-\frac{1}{L}\eta'_W\mathcal{L}'\frac{\langle d\boldsymbol{h}_\ell, d\bar{\boldsymbol{h}}_\ell\rangle}{N}\frac{1}{\sqrt{N}}\sum_\beta\boldsymbol{U}_{\ell,\beta\alpha}\sum_\gamma\phi\left(\frac{1}{\sqrt{N}}\boldsymbol{U}_{\ell,\beta\gamma}\boldsymbol{h}_{\ell-1,\gamma}\right)\phi'(\bar{\boldsymbol{x}}_{\ell,\beta})$$

$$\sim -\frac{1}{L}\eta'_W\mathcal{L}'\frac{\langle d\boldsymbol{h}_\ell, d\bar{\boldsymbol{h}}_\ell\rangle}{N}\frac{1}{\sqrt{N}}\sum_\beta\boldsymbol{U}_{\ell,\beta\alpha}\phi\left(\frac{1}{\sqrt{N}}\boldsymbol{U}_{\ell,\beta\alpha}\boldsymbol{h}_{\ell-1,\alpha}\right)\phi'(\bar{\boldsymbol{x}}_{\ell,\beta})$$

$$\sim -\frac{1}{L}\eta'_W\mathcal{L}'\frac{\langle d\boldsymbol{h}_\ell, d\bar{\boldsymbol{h}}_\ell\rangle}{N}\left[\frac{1}{N}\sum_\beta\boldsymbol{U}_{\ell,\beta\alpha}^2\phi'(\boldsymbol{x}_{\ell,\beta})\phi'(\bar{\boldsymbol{x}}_{\ell,\beta})\right]\boldsymbol{h}_{\ell-1,\alpha}$$

$$\sim -\frac{1}{L}\eta'_W\mathring{\mathcal{L}}'\mathbb{E}[Z^{d\boldsymbol{h}_\ell}Z^{d\bar{\boldsymbol{h}}_\ell}]\mathbb{E}[\phi(Z^{\boldsymbol{x}_\ell})\phi(Z^{\bar{\boldsymbol{x}}_\ell})]Z^{\boldsymbol{h}_{\ell-1}}$$

Therefore, the mean-field characterization is given by

$$\boxed{Z^{d\bar{\boldsymbol{h}}_{\ell-1}} = Z^{d\bar{\boldsymbol{h}}_\ell} + \underbrace{\frac{1}{\sqrt{L}}Z^{\boldsymbol{U}_\ell^\top d\bar{\boldsymbol{x}}_\ell}}_{\text{Gaussian}} - \underbrace{\frac{1}{L}\eta'_U\mathring{\mathcal{L}}'\mathbb{E}[Z^{d\boldsymbol{x}_\ell}Z^{d\bar{\boldsymbol{x}}_\ell}]Z^{\boldsymbol{h}_{\ell-1}}}_{\text{learning}} - \underbrace{\frac{1}{L}\eta'_W\mathring{\mathcal{L}}'\mathbb{E}[Z^{d\boldsymbol{h}_\ell}Z^{d\bar{\boldsymbol{h}}_\ell}]\mathbb{E}[\phi'(Z^{\boldsymbol{x}_\ell})\phi'(Z^{\bar{\boldsymbol{x}}_\ell})]Z^{\boldsymbol{h}_{\ell-1}}}_{\text{interaction}}} \quad (10)$$

Note that if either $\eta'_U \sim \sqrt{L}$ or $\eta'_W \sim \sqrt{L}$, then the learning or interaction term would blow up, respectively.

In summary, we have forward dynamics:

$$Z^{\boldsymbol{x}_\ell(1)} = Z^{\tilde{\boldsymbol{x}}_\ell(1)} - \frac{\eta'_U}{\sqrt{L}}\mathcal{G}(Z^{\delta\boldsymbol{x}_\ell(0)}) - \frac{\eta'_W}{L^{\frac{3}{2}}}\mathcal{I}(Z^{\delta\boldsymbol{x}_\ell(0)})$$

$$Z^{\boldsymbol{h}_\ell(1)} - Z^{\boldsymbol{h}_{\ell-1}(1)} = \frac{1}{\sqrt{L}}Z^{\tilde{\boldsymbol{h}}_\ell(1)} - \frac{\eta_W}{L}\mathcal{G}(Z^{\delta\boldsymbol{h}_\ell}) - \frac{\eta'_U}{L}\mathcal{I}(Z^{\delta\boldsymbol{h}_\ell}).$$

and

$$Z^{\delta\boldsymbol{x}_\ell(1)} = Z^{\delta\tilde{\boldsymbol{x}}_\ell(1)} - \frac{\eta'_W}{\sqrt{L}}\mathcal{G}(Z^{\boldsymbol{x}_\ell(0)}) - \frac{\eta'_U}{L^{\frac{3}{2}}}\mathcal{I}(Z^{\boldsymbol{x}_\ell(0)})$$

$$Z^{\delta\boldsymbol{h}_{\ell-1}(1)} - Z^{\delta\boldsymbol{h}_\ell(1)} = \frac{1}{\sqrt{L}}Z^{\delta\tilde{\boldsymbol{h}}_{\ell-1}(1)} - \frac{\eta'_U}{L}\mathcal{G}(Z^{\boldsymbol{h}_{\ell-1}(0)}) - \frac{\eta'_W}{L}\mathcal{I}(Z^{\boldsymbol{h}_{\ell-1}(0)})$$

Hence, if we have $\eta'_U \sim \sqrt{L}$, we counteract the learning collapses in $\boldsymbol{x}_\ell$ but make the *interaction* term in the forward residual stream $\partial_t\boldsymbol{h}$ and the *learning* term in the backward residual stream $\partial_t\delta\boldsymbol{h}$ explore. If we use an asymmetric learning rate $\tilde{\eta}'_U$ in the backward, increasing the forward learning rate $\eta'_U$ does to results in an exploding learning term in the backward residual stream. While it still theoretically results in an exploding term in the forward residual stream, we empirically verify that this change only makes the meld increase in the network output.

If we use decoupled backward weights, the interaction terms disappear, and the resulting dynamics are

$$Z^{\boldsymbol{x}_\ell(1)} = Z^{\tilde{\boldsymbol{x}}_\ell(1)} - \frac{\eta'_U}{\sqrt{L}}\mathcal{G}(Z^{\delta\boldsymbol{x}_\ell(0)})$$

$$Z^{\boldsymbol{h}_\ell(1)} - Z^{\boldsymbol{h}_{\ell-1}(1)} = \frac{1}{\sqrt{L}}Z^{\tilde{\boldsymbol{h}}_\ell(1)} - \frac{\eta'_W}{L}\mathcal{G}(Z^{\delta\boldsymbol{h}_\ell}),$$

and

$$Z^{\delta \boldsymbol{x}_\ell(1)} = Z^{\delta \widetilde{\boldsymbol{x}}_\ell(1)} - \frac{\widetilde{\eta}'_W}{\sqrt{L}} \mathcal{G}(Z^{\boldsymbol{x}_\ell(0)})$$

$$Z^{\delta \boldsymbol{h}_{\ell-1}(1)} - Z^{\delta \boldsymbol{h}_\ell(1)} = \frac{1}{\sqrt{L}} Z^{\delta \widetilde{\boldsymbol{h}}_{\ell-1}(1)} - \frac{\widetilde{\eta}'_U}{L} \mathcal{G}(Z^{\boldsymbol{h}_{\ell-1}(0)})$$

Accordingly, increasing learning rates does not lead to exploration of interactions. Hence, we can use asymmetric learning rates as follows:

$$\eta'_U = \widetilde{\eta}'_W = \eta_c \sqrt{L}, \qquad \eta'_W = \widetilde{\eta}'_U = \eta_c.$$

### B.5. Convergence rate of first forward and first backward

The analysis of first forward and first backward is analogous to that in Appendix C of (Yao et al., 2025), albeit more involved. So we provide statements of main lemmas and propositions, while proofs are omitted.

The first forward can be rewritten as

$$\boldsymbol{h}_0 = \frac{1}{\sqrt{D}} \boldsymbol{E} \boldsymbol{x}$$

$$\boldsymbol{x}_\ell = \frac{1}{\sqrt{N}} \|\boldsymbol{h}_{\ell-1}\| \boldsymbol{z}_\ell^{\boldsymbol{U}}, \quad \boldsymbol{z}_\ell^{\boldsymbol{U}} \sim \mathcal{N}(0, \boldsymbol{I}_N), \qquad \forall \ell \in [L]$$

$$\boldsymbol{h}_\ell = \boldsymbol{h}_{\ell-1} + \frac{1}{\sqrt{LN}} \|\phi(\boldsymbol{x}_\ell)\| \boldsymbol{z}_\ell^{\boldsymbol{W}}, \quad \boldsymbol{z}_\ell^{\boldsymbol{W}} \sim \mathcal{N}(0, \boldsymbol{I}_N), \qquad \forall \ell \in [L]$$

$$f(\boldsymbol{x}) = \frac{1}{N} \boldsymbol{v}^\top \boldsymbol{h}_L$$

To distinguish from quantities after taking limits of $n \to \infty$ and $L \to \infty$, we add superscripts and write each coordinate as

$$x_{\ell,i}^{N,L} = \frac{1}{\sqrt{N}} \|\boldsymbol{h}_{\ell-1}^{N,L}\| z_{\ell,i}^{\boldsymbol{U}}, \qquad h_{\ell,i}^{N,L} = h_{\ell-1,i}^{N,L} + \frac{1}{\sqrt{LN}} \|\phi(\boldsymbol{x}_{\ell-1}^{N,L})\| z_{\ell,i}^{\boldsymbol{W}}.$$

We want to show that each coordinate converges to

$$x_{\ell,i}^L = \sqrt{\mathbb{E}(h_{\ell-1,i}^L)^2} z_{\ell,i}^{\boldsymbol{U}}, \qquad h_{\ell,i}^L = h_{\ell-1,i}^L + \frac{1}{\sqrt{L}} \sqrt{\mathbb{E}\phi^2(x_{\ell-1,i}^L)} z_{\ell,i}^{\boldsymbol{W}}$$

as $n \to \infty$. For this, we have the following preparatory result.

**Lemma B.2.** *For each* $i \in \mathbb{N}$, $\sup_{L \geq 1} \max_{\ell=0,1,\dots,L} \mathbb{E}[h_{\ell,i}^L]^4 < \infty$, $\inf_{L \geq 1} \min_{\ell=0,1,\dots,L} \mathbb{E}(h_{\ell,i}^L)^2 > 0$, *and* $\inf_{L \geq 1} \min_{\ell=0,1,\dots,L} \mathbb{E}\phi^2(x_{\ell,i}^L) > 0$.

Using this, we have the following convergence.

**Proposition B.3.** *For each* $i \in \mathbb{N}$,

$$\sup_{L \geq 1} \max_{\ell=0,1,\dots,L} \mathbb{E}(h_{\ell,i}^{N,L} - h_{\ell,i}^L)^2 \leq C/N, \quad \sup_{L \geq 1} \max_{\ell=0,1,\dots,L} \mathbb{E}(x_{\ell,i}^{N,L} - x_{\ell,i}^L)^2 \leq C/N.$$

Next, to analyze the limit of $x_{\ell,i}^L$ and $h_{\ell,i}^L$ as $L \to \infty$, we omit the subscript $i$ and view

$$\frac{1}{\sqrt{L}} z_\ell^{\boldsymbol{W}} = w\left(\frac{\ell}{L}\right) - w\left(\frac{\ell-1}{L}\right), \quad z_\ell^{\boldsymbol{U}} = u\left(\frac{\ell}{L}\right)$$

for a standard Brownian motion $w$ and a standard white noise $u$, that is, $\{u(t) : t \in [0,1]\}$ are independent standard Gaussians. Then we can write $h_\ell^L = h_{\ell/L}^{(L)}$ and $x_\ell^L = x_{\ell/L}^{(L)}$, where

$$dh_t^{(L)} = \sqrt{\mathbb{E}\phi^2(x_{t_L}^{(L)})}\, dw_t, \quad x_t^{(L)} = \sqrt{\mathbb{E}(h_t^{(L)})^2}\, u_t$$

and $t_L := \frac{\lfloor tL \rfloor}{L}$ for $t \in [0, 1]$. Consider the McKean–Vlasov process

$$dh_t = \sqrt{\mathbb{E}\phi^2(x_t)}\, dw_t, \quad x_t = \sqrt{\mathbb{E}h_t^2}\, u_t.$$

Then $\{h_t^{(L)}\}$ is just the Euler–Maruyama discretization for $\{h_t\}$ with step size $\Delta t = 1/L$.

**Proposition B.4.** *There exists a unique $\{h_t\}$ and*

$$\sup_{0 \le t \le 1} \mathbb{E}h_t^2 < \infty, \quad \mathbb{E}[h_t - h_{t_L}]^2 \le C(t - t_L) \le C/L. \tag{11}$$

Using this, we have the following result quantifying the error as $L \to \infty$.

**Proposition B.5.** *For all $L \ge 1$,*

$$\max_{\ell=0,1,\ldots,L} \mathbb{E}[h_\ell^L - h_{\ell/L}]^2 = \max_{\ell=0,1,\ldots,L} \mathbb{E}[h_{\ell/L}^{(L)} - h_{\ell/L}]^2 \le C/L,$$

$$\max_{\ell=0,1,\ldots,L} \mathbb{E}[x_\ell^L - x_{\ell/L}]^2 = \max_{\ell=0,1,\ldots,L} \mathbb{E}[x_{\ell/L}^{(L)} - x_{\ell/L}]^2 \le C/L.$$

Combining Propositions B.3 and B.5, we get

$$\max_{\ell=0,1,\ldots,L} \max_{i=1,2,\ldots,N} \mathbb{E}(\boldsymbol{h}_{\ell,i} - h_{\ell/L,i})^2 \le \frac{C}{N} + \frac{C}{L}, \quad \max_{\ell=0,1,\ldots,L} \max_{i=1,2,\ldots,N} \mathbb{E}(\boldsymbol{x}_{\ell,i} - x_{\ell/L,i})^2 \le \frac{C}{N} + \frac{C}{L}. \tag{12}$$

Next we analyze the first backward under the gradient independence assumption. In the next subsection we will prove the convergence (as $L \to \infty$) without imposing this assumption. The first backward path can be rewritten as

$$\delta\boldsymbol{h}_L = \boldsymbol{v},$$

$$\delta\boldsymbol{x}_\ell = \phi'(\boldsymbol{x}_\ell) \odot \frac{1}{\sqrt{N}} \|\delta\boldsymbol{h}_\ell\| \boldsymbol{z}_\ell^{\widetilde{\boldsymbol{W}}}, \quad \boldsymbol{z}_\ell^{\widetilde{\boldsymbol{W}}} \sim \mathcal{N}(0, \boldsymbol{I}_N), \quad \forall \ell \in [L],$$

$$\delta\boldsymbol{h}_{\ell-1} = \delta\boldsymbol{h}_\ell + \frac{1}{\sqrt{LN}} \|\delta\boldsymbol{x}_\ell\| \boldsymbol{z}_\ell^{\widetilde{\boldsymbol{U}}}, \quad \boldsymbol{z}_\ell^{\widetilde{\boldsymbol{U}}} \sim \mathcal{N}(0, \boldsymbol{I}_N), \quad \forall \ell \in [L],$$

where $\boldsymbol{z}^{\widetilde{\boldsymbol{W}}}$ and $\boldsymbol{z}^{\widetilde{\boldsymbol{U}}}$ are independent of $\boldsymbol{z}^{\boldsymbol{W}}$ and $\boldsymbol{z}^{\boldsymbol{U}}$. Similar to the first forward notation, the $i$-th coordinate is

$$\delta h_{L,i}^{N,L} = v_i,$$

$$\delta x_{\ell,i}^{N,L} = \frac{1}{\sqrt{N}} \phi'(x_{\ell,i}^{N,L}) \|\delta\boldsymbol{h}_\ell^{N,L}\| z_{\ell,i}^{\widetilde{\boldsymbol{W}}}, \quad \forall \ell \in [L],$$

$$\delta h_{\ell-1,i}^{N,L} = \delta h_{\ell,i}^{N,L} + \frac{1}{\sqrt{LN}} \|\delta\boldsymbol{x}_\ell^{N,L}\| z_{\ell,i}^{\widetilde{\boldsymbol{U}}}, \quad \forall \ell \in [L].$$

We want to show as $n \to \infty$, $\{\delta x_{\ell,i}^{n,L}, \delta h_{\ell,i}^{n,L}\}$ converges to

$$\delta x_{\ell,i}^L = \phi'(x_{\ell,i}^L)\sqrt{\mathbb{E}(\delta h_{\ell,i}^L)^2} z_{\ell,i}^{\widetilde{\boldsymbol{W}}}, \quad \delta h_{\ell-1,i}^L = \delta h_{\ell,i}^L + \frac{1}{\sqrt{L}}\sqrt{\mathbb{E}(\delta x_{\ell,i}^L)^2} z_{\ell,i}^{\widetilde{\boldsymbol{U}}}.$$

For this, we have the following preparatory result.

**Lemma B.6.** *For each $i \in \mathbb{N}$, $\sup_{L \ge 1} \max_{\ell=0,1,\ldots,L} \mathbb{E}[\delta h_{\ell,i}^L]^4 < \infty$, $\inf_{L \ge 1} \min_{\ell=0,1,\ldots,L} \mathbb{E}(\delta h_{\ell,i}^L)^2 > 0$ and $\inf_{L \ge 1} \min_{\ell=0,1,\ldots,L} \mathbb{E}(\delta x_{\ell,i}^L)^2 > 0$.*

Using this, we have the following convergence.

**Proposition B.7.** *For each $i \in \mathbb{N}$,*

$$\sup_{L \ge 1} \max_{\ell=0,1,\ldots,L} \mathbb{E}(\delta h_{\ell,i}^{N,L} - \delta h_{\ell,i}^L)^2 \le C/N, \quad \sup_{L \ge 1} \max_{\ell=0,1,\ldots,L} \mathbb{E}(\delta x_{\ell,i}^{N,L} - \delta x_{\ell,i}^L)^2 \le C/N.$$

Next, to analyze the limit of $\delta x^L_{\ell,i}$ and $\delta h^L_{\ell,i}$ as $L \to \infty$, we omit the subscript $i$ and view

$$\frac{1}{\sqrt{L}} z^{\widetilde{U}}_\ell = \tilde{u}\left(\frac{\ell}{L}\right) - \tilde{u}\left(\frac{\ell-1}{L}\right), \quad z^{\widetilde{W}}_\ell = \tilde{w}\left(\frac{\ell}{L}\right)$$

for a standard Brownian motion $\tilde{u}$ and a standard white noise $\tilde{w}$, that is, $\{\tilde{w}(t) : t \in [0,1]\}$ are independent standard Gaussians. Then we can write $\delta h^L_\ell = \delta h^{(L)}_{\ell/L}$ and $\delta x^L_\ell = \delta x^{(L)}_{\ell/L}$, where

$$d(\delta h^{(L)}_t) = \sqrt{\mathbb{E}(\delta x^{(L)}_{\tilde{t}_L})^2}\, d\tilde{u}_t, \quad \delta x^{(L)}_t = \phi'(x^{(L)}_t)\sqrt{\mathbb{E}(\delta h^{(L)}_t)^2}\tilde{w}_t,$$

and $\tilde{t}_L := \frac{[tL]}{L}$ for $t \in [0,1]$. Consider the McKean–Vlasov process (that goes backward in time)

$$d(\delta h_t) = \sqrt{\mathbb{E}(\delta x_t)^2}\, d\tilde{u}_t, \quad \delta x_t = \phi'(x_t)\sqrt{\mathbb{E}(\delta h_t)^2}\tilde{w}_t.$$

Then $\{\delta h^{(L)}_t\}$ is just the Euler–Maruyama discretization for $\{\delta h_t\}$ with step size $\Delta t = 1/L$.

**Proposition B.8.** *There exists a unique $\{\delta h_t\}$ and*

$$\sup_{0 \le t \le 1} \mathbb{E}(\delta h_t)^2 < \infty, \quad \mathbb{E}[\delta h_t - \delta h_{\tilde{t}_L}]^2 \le C(\tilde{t}_L - t) \le C/L. \tag{13}$$

Using this, we have the following result quantifying the error as $L \to \infty$.

**Proposition B.9.** *For all $L \ge 1$,*

$$\max_{\ell=0,1,\ldots,L} \mathbb{E}[\delta h^L_\ell - \delta h_{\ell/L}]^2 = \max_{\ell=0,1,\ldots,L} \mathbb{E}[\delta h^{(L)}_{\ell/L} - \delta h_{\ell/L}]^2 \le C/L,$$

$$\max_{\ell=0,1,\ldots,L} \mathbb{E}[\delta x^L_\ell - \delta x_{\ell/L}]^2 = \max_{\ell=0,1,\ldots,L} \mathbb{E}[\delta x^{(L)}_{\ell/L} - \delta x_{\ell/L}]^2 \le C/L.$$

Combining Propositions B.7 and B.9, we get

$$\max_{\ell=0,1,\ldots,L} \max_{i=1,2,\ldots,N} \mathbb{E}(\delta \boldsymbol{h}_{\ell,i} - \delta h_{\ell/L,i})^2 \le \frac{C}{N} + \frac{C}{L}, \quad \max_{\ell=0,1,\ldots,L} \max_{i=1,2,\ldots,N} \mathbb{E}(\delta \boldsymbol{x}_{\ell,i} - \delta x_{\ell/L,i})^2 \le \frac{C}{N} + \frac{C}{L}. \tag{14}$$

### B.6. Convergence of second forward and second backward

We start with the infinite-width limit given in Sections B.1, B.2, B.3 (in particular Eq. (7) and Eq. (8)) and B.4 (in particular Eq. (9), and Eq. (10)). The limit as $N \to \infty$ for the first two forwards and backwards can be rewritten as

$$x_\ell^{(0),L} = z_\ell^{(0),\boldsymbol{U}},$$

$$h_\ell^{(0),L} = h_{\ell-1}^{(0),L} + \frac{1}{\sqrt{L}} z_\ell^{(0),\boldsymbol{W}},$$

$$\delta x_\ell^{(0),L} = \phi'(x_\ell^{(0),L}) z_\ell^{(0),\widetilde{\boldsymbol{W}}},$$

$$\delta h_{\ell-1}^{(0),L} = \delta h_\ell^{(0),L} + \frac{1}{\sqrt{L}} z_\ell^{(0),\widetilde{\boldsymbol{U}}},$$

$$x_\ell^{(1),L} = z_\ell^{(1),\boldsymbol{U}} - \frac{1}{\sqrt{L}} \eta_U' \mathring{\mathcal{L}}' \mathbb{E}[h_{\ell-1}^{(0),L} h_{\ell-1}^{(1),L}] \phi'(x_\ell^{(0),L}) z_\ell^{(0),\widetilde{\boldsymbol{W}}}$$

$$- L^{-\frac{3}{2}} \eta_W' \mathring{\mathcal{L}}' \mathbb{E}[\phi(x_{\ell-1}^{(0),L}) \phi(x_{\ell-1}^{(1),L})] \phi(x_\ell^{(0),L}) z_\ell^{(0),\widetilde{\boldsymbol{W}}},$$

$$h_\ell^{(1),L} = h_{\ell-1}^{(1),L} + \frac{1}{\sqrt{L}} z_\ell^{(1),\boldsymbol{W}} - \frac{1}{L} \eta_W' \mathring{\mathcal{L}}' \mathbb{E}[\phi(x_\ell^{(0),L}) \phi(x_\ell^{(1),L})] \delta h_\ell^{(0),L}$$

$$- \frac{1}{L} \eta_U' \mathring{\mathcal{L}}' \mathbb{E}[h_{\ell-1}^{(0),L} h_{\ell-1}^{(1),L}] \mathbb{E}[\phi'(x_\ell^{(0),L}) \phi'(x_\ell^{(1),L})] \delta h_\ell^{(0),L},$$

$$\delta x_\ell^{(1),L} = \phi'(x_\ell^{(1),L}) z_\ell^{(1),\widetilde{\boldsymbol{W}}} - \frac{1}{\sqrt{L}} \eta_W' \mathring{\mathcal{L}}' \mathbb{E}[\delta h_\ell^{(0),L} \delta h_\ell^{(1),L}] \phi'(x_\ell^{(1),L}) \phi(x_\ell^{(0),L})$$

$$- L^{-\frac{3}{2}} \eta_U' \mathring{\mathcal{L}}' \mathbb{E}[\delta x_{\ell+1}^{(0),L} \delta x_{\ell+1}^{(1),L}] \phi'(x_\ell^{(1),L}) \phi(x_\ell^{(0),L}),$$

$$\delta h_{\ell-1}^{(1),L} = \delta h_\ell^{(1),L} + \frac{1}{\sqrt{L}} z_\ell^{(1),\widetilde{\boldsymbol{U}}} - \frac{1}{L} \eta_U' \mathring{\mathcal{L}}' \mathbb{E}[\delta x_\ell^{(0),L} \delta x_\ell^{(1),L}] h_{\ell-1}^{(0),L}$$

$$- \frac{1}{L} \eta_W' \mathring{\mathcal{L}}' \mathbb{E}[\delta h_\ell^{(0),L} \delta h_\ell^{(1),L}] \mathbb{E}[\phi'(x_\ell^{(0),L}) \phi'(x_\ell^{(1),L})] h_{\ell-1}^{(0),L},$$

where $(z_\ell^{(0),\boldsymbol{W}}, z_\ell^{(1),\boldsymbol{W}}), (z_\ell^{(0),\widetilde{\boldsymbol{W}}}, z_\ell^{(1),\widetilde{\boldsymbol{W}}}), (z_\ell^{(0),\boldsymbol{U}}, z_\ell^{(1),\boldsymbol{U}}), (z_\ell^{(0),\widetilde{\boldsymbol{U}}}, z_\ell^{(1),\widetilde{\boldsymbol{U}}})$ are independent Gaussian random vectors with mean 0 and variance-covariance matrix

$$\mathrm{Cov}(z_\ell^{(k),\boldsymbol{W}}, z_\ell^{(k'),\boldsymbol{W}}) = \mathbb{E}[\phi(x_\ell^{(k),L}) \phi(x_\ell^{(k'),L})],$$

$$\mathrm{Cov}(z_\ell^{(k),\widetilde{\boldsymbol{W}}}, z_\ell^{(k'),\widetilde{\boldsymbol{W}}}) = \mathbb{E}[\delta h_\ell^{(k),L} \delta h_\ell^{(k'),L}],$$

$$\mathrm{Cov}(z_\ell^{(k),\boldsymbol{U}}, z_\ell^{(k'),\boldsymbol{U}}) = \mathbb{E}[h_{\ell-1}^{(k),L} h_{\ell-1}^{(k'),L}],$$

$$\mathrm{Cov}(z_\ell^{(k),\widetilde{\boldsymbol{U}}}, z_\ell^{(k'),\widetilde{\boldsymbol{U}}}) = \mathbb{E}[\delta x_\ell^{(k),L} \delta x_\ell^{(k'),L}].$$

Letting $L \to \infty$, we expect to have

$$x_t^{(0)} = u_t^{(0)}, \tag{15}$$

$$dh_t^{(0)} = dw_t^{(0)}, \tag{16}$$

$$\delta x_t^{(0)} = \phi'(x_t^{(0)}) \tilde{w}_t^{(0)}, \tag{17}$$

$$d(\delta h_t^{(0)}) = d\tilde{u}_t^{(0)}, \tag{18}$$

$$x_t^{(1)} = u_t^{(1)}, \tag{19}$$

$$dh_t^{(1)} = dw_t^{(1)} - \eta_W' \mathring{\mathcal{L}}' \mathbb{E}[\phi(x_t^{(0)}) \phi(x_t^{(1)})] \delta h_t^{(0)} \, dt - \eta_U' \mathring{\mathcal{L}}' \mathbb{E}[h_t^{(0)} h_t^{(1)}] \mathbb{E}[\phi'(x_t^{(0)}) \phi'(x_t^{(1)})] \delta h_t^{(0)} \, dt, \tag{20}$$

$$\delta x_t^{(1)} = \phi'(x_t^{(1)}) \tilde{w}_t^{(1)}, \tag{21}$$

$$d(\delta h_t^{(1)}) = d\tilde{u}_t^{(1)} - \eta_U' \mathring{\mathcal{L}}' \mathbb{E}[\delta x_t^{(0)} \delta x_t^{(1)}] h_t^{(0)} \, dt - \eta_W' \mathring{\mathcal{L}}' \mathbb{E}[\delta h_t^{(0)} \delta h_t^{(1)}] \mathbb{E}[\phi'(x_t^{(0)}) \phi'(x_t^{(1)})] h_t^{(0)} \, dt, \tag{22}$$

where $\{(w_t^{(0)}, w_t^{(1)}), (\tilde{u}_t^{(0)}, \tilde{u}_t^{(1)})\}$ are independent Brownian motions with mean 0 and cross-variations

$$d\langle w^{(k)}, w^{(k')}\rangle_t = \mathbb{E}[\phi(x_t^{(k)})\phi(x_t^{(k')})]\,dt,$$

$$d\langle \tilde{u}^{(k)}, \tilde{u}^{(k')}\rangle_t = \mathbb{E}[\delta x_t^{(k)}\delta x_t^{(k')}]\,dt,$$

and $\{(u_t^{(0)}, u_t^{(1)}), (\tilde{w}_t^{(0)}, \tilde{w}_t^{(1)})\}$ are independent Gaussian random vectors with mean 0 and covariances

$$\mathrm{Cov}(u^{(k)}, u^{(k')}) = \mathbb{E}[h_t^{(k)} h_t^{(k')}],$$

$$\mathrm{Cov}(\tilde{w}^{(k)}, \tilde{w}^{(k')}) = \mathbb{E}[\delta h_t^{(k)} \delta h_t^{(k')}].$$

We note that here the evolution of $\delta h_t^{(0)}$ and $\delta h_t^{(1)}$ is interpreted backward from $t = 1$ to $t = 0$.

**Theorem B.10.** *Assume that*

- *$\mathring{\mathcal{L}}'$, $\phi$, $\phi'$ are Lipschitz continuous;*

- *there exists a solution to the system Eq. (15)–Eq. (22) such that the $2 \times 2$ matrices $(\mathbb{E}[\phi(x_t^{(k)})\phi(x_t^{(k')})])$ and $(\mathbb{E}[\delta x_t^{(k)}\delta x_t^{(k')}])$ are uniformly strictly positive definite, that is, the smallest eigenvalue is always great than some $\varepsilon > 0$.*

*Then for all $L \geq 1$, for each of the variable $\theta = x, h, \delta x, \delta h$, we have*

$$\sup_{\ell=0,1,\ldots,L} \mathbb{E}\left[|\theta_\ell^{(0),L} - \theta_{\ell/L}^{(0)}|^2 + |\theta_\ell^{(1),L} - \theta_{\ell/L}^{(1)}|^2\right] \leq C/L.$$

For the proof of this theorem, a key observation is that the following higher order terms in the evolution of $x^{(1),L}$ and $\delta x^{(1),L}$ are vanishing as $L \to \infty$:

$$\mathbb{E}\left|\frac{1}{\sqrt{L}}\eta_U'\mathring{\mathcal{L}}'\mathbb{E}[h_{\ell-1}^{(0),L} h_{\ell-1}^{(1),L}]\phi'(x_\ell^{(0),L})z_\ell^{(0),\widetilde{W}} + L^{-\frac{3}{2}}\eta_W'\mathring{\mathcal{L}}'\mathbb{E}[\phi(x_{\ell-1}^{(0),L})\phi(x_{\ell-1}^{(1),L})]\phi(x_\ell^{(0),L})z_\ell^{(0),\widetilde{W}}\right| \leq \frac{1}{\sqrt{L}} \to 0,$$

$$\mathbb{E}\left|\frac{1}{\sqrt{L}}\eta_W'\mathring{\mathcal{L}}'\mathbb{E}[\delta h_\ell^{(0),L} \delta h_\ell^{(1),L}]\phi'(x_\ell^{(1),L})\phi(x_\ell^{(0),L}) + L^{-\frac{3}{2}}\eta_U'\mathring{\mathcal{L}}'\mathbb{E}[\delta x_{\ell+1}^{(0),L}\delta x_{\ell+1}^{(1),L}]\phi'(x_\ell^{(1),L})\phi(x_\ell^{(0),L})\right| \leq \frac{1}{\sqrt{L}} \to 0,$$

which follows from the moment bound that can be obtained via a standard Gronwall's inequality argument:

$$\sup_{L\geq 1}\max_{\ell=0,1,\ldots,L} \mathbb{E}\left[|\theta_\ell^{(0),L}|^2 + |\theta_\ell^{(1),L}|^2\right] < \infty, \quad \text{for } \theta = x, h, \delta x, \delta h.$$

The rest proof is analogous to that of (Yao et al., 2025, Proposition G.6), albeit more involved, and hence is omitted.

## C. Fully decoupled Two-layer case

We consider the forward:

$$\boldsymbol{h}_0 = \frac{1}{\sqrt{D}}\boldsymbol{E}\boldsymbol{x}$$

$$\boldsymbol{x}_\ell = \frac{1}{\sqrt{N}}\boldsymbol{U}_\ell\boldsymbol{h}_{\ell-1}, \qquad \forall \ell \in [L]$$

$$\boldsymbol{h}_\ell = \boldsymbol{h}_{\ell-1} + \frac{1}{\sqrt{LN}}\boldsymbol{W}_\ell\phi(\boldsymbol{x}_\ell), \qquad \forall \ell \in [L]$$

$$f(\boldsymbol{x}) = \frac{1}{N}\boldsymbol{v}^\top\boldsymbol{h}_L$$

Then, by using width-wise $\mu$P, the gradient propagation can be characterized as follows:

$$d\boldsymbol{h}_L = N\frac{\partial f}{\partial \boldsymbol{h}_L} = \boldsymbol{v}$$

$$d\boldsymbol{h}_{\ell-1} = N\frac{\partial f}{\partial \boldsymbol{h}_{\ell-1}} = d\boldsymbol{h}_\ell + \frac{1}{\sqrt{LN}}\widetilde{\boldsymbol{U}}_\ell^\top d\boldsymbol{x}_\ell, \quad \ell \in [L],$$

where we abuse the term

$$dx_\ell = \frac{1}{\sqrt{N}} \widetilde{W}_\ell^\top dh_\ell \odot \phi'(x_\ell),$$

since it is **note** derivative of $\partial f / \partial x_\ell$ but a scaled one by the factor $\sqrt{L}$. **Note** that we essentially **swap** the order of $W_\ell$ and $U_\ell$ in the backpropagation, so that $W_\ell$ governs both forward and backward driven-stochastic.

Then, using the backward gradients, we can compute the

$$dv = N \frac{\partial f}{\partial v} = h_L$$

$$dW_\ell = N \frac{\partial f}{\partial W_\ell} = \frac{1}{\sqrt{LN}} dh_\ell \phi(x_\ell)^\top$$

$$dU_\ell = N \frac{\partial L}{\partial U_\ell} = \frac{1}{\sqrt{LN}} dx_\ell h_{\ell-1}^\top$$

$$dE = \frac{1}{\sqrt{D}} dh_0 x^\top$$

As a result, the gradient descents are given by

$$v^+ = v - \eta_v \mathcal{L}' \cdot \frac{1}{N} dv = v - \eta_v' \mathcal{L}' \cdot h_L$$

$$W_\ell^+ = W_\ell - \eta_W \mathcal{L}' \cdot \frac{1}{N} dW_\ell = W_\ell - \eta_W' \mathcal{L}' \frac{1}{\sqrt{LN}} dh_\ell \phi(x_\ell)^\top$$

$$\widetilde{W}_\ell^+ = \widetilde{W}_\ell - \eta_W \mathcal{L}' \cdot \frac{1}{N} dW_\ell = \widetilde{W}_\ell - \tilde{\eta}_W' \mathcal{L}' \frac{1}{\sqrt{LN}} dh_\ell \phi(x_\ell)^\top$$

$$U_\ell^+ = U_\ell - \eta_U \mathcal{L}' \cdot \frac{1}{N} dU_\ell = U_\ell - \eta_U' \mathcal{L}' \frac{1}{\sqrt{LN}} dx_\ell h_{\ell-1}^\top$$

$$\widetilde{U}_\ell^+ = \widetilde{U}_\ell - \tilde{\eta}_U \mathcal{L}' \cdot \frac{1}{N} dU_\ell = \widetilde{U}_\ell - \tilde{\eta}_U' \mathcal{L}' \frac{1}{\sqrt{LN}} dx_\ell h_{\ell-1}^\top$$

$$E_\ell^+ = E_\ell - \eta_E \mathcal{L}' \cdot \frac{1}{N} dE_\ell = E_\ell - \eta_E' \mathcal{L}' \frac{1}{\sqrt{D}} dh_0 x^\top,$$

where $\eta_v = \eta_v' N$ and the same cases are applied to $\eta_U, \eta_W, \eta_E$.

### C.1. First-Forward Pass

As the forward is the same as before, we still have

$$Z^{h_0} \sim \mathcal{N}(0, \|x\|^2 / D)$$

$$Z^{x_\ell} \sim \mathcal{N}(0, \mathbb{E}[Z^{U_\ell h_{\ell-1}}]^2)$$

$$Z^{h_\ell} = Z^{h_{\ell-1}} + \frac{1}{\sqrt{L}} Z^{W_\ell \phi(x_\ell)}, \quad Z^{W_\ell \phi(x_\ell)} = \mathcal{N}(0, \mathbb{E}[\phi(Z^{x_\ell})^2]),$$

$$\mathring{f}(x) = \mathbb{E}[Z^v Z^{h_L}], \quad Z^v \sim \mathcal{N}(0, 1),$$

where $\{Z^{W_\ell \phi(x_\ell)}, Z^{x_\ell}\}_\ell$ are mutually independent Gaussians.

### C.2. First-Backward Pass

We have

$$Z^{dh_L} = Z^v \sim \mathcal{N}(0, 1)$$

$$Z^{dx_\ell} = Z^{\widetilde{W}_\ell^\top dh_\ell} \phi'(Z^{x_\ell}), \quad Z^{\widetilde{W}_\ell^\top dh_\ell} \sim \mathcal{N}(0, \mathbb{E}[Z^{dh_\ell}]^2)$$

$$Z^{dh_{\ell-1}} = Z^{dh_\ell} + \frac{1}{\sqrt{L}} Z^{\widetilde{U}_\ell^\top dx_\ell}$$

### C.3. Second-Forward Pass

We have

$$
\bar{x}_\ell = \frac{1}{\sqrt{N}}(U_\ell + \Delta U_\ell)\bar{h}_{\ell-1}
$$

$$
= \frac{1}{\sqrt{N}}U_\ell\bar{h}_{\ell-1} + \frac{1}{\sqrt{N}}\left[-\eta'_U\mathcal{L}'\frac{1}{\sqrt{LN}}dx_\ell h_{\ell-1}^\top\right]\bar{h}_{\ell-1}
$$

$$
= \frac{1}{\sqrt{N}}U_\ell\bar{h}_{\ell-1} - \frac{1}{\sqrt{L}}\eta'_U\mathcal{L}'\frac{\langle h_{\ell-1}, \bar{h}_{\ell-1}\rangle}{N}dx_\ell
$$

where the interaction term stems from

$$
\frac{1}{\sqrt{N}}U_\ell\bar{h}_{\ell-1} = \frac{1}{\sqrt{N}}U_\ell\left[\bar{h}_{\ell-2} + \frac{1}{\sqrt{LN}}(W_{\ell-1} + \Delta W_{\ell-1})\phi(\bar{x}_{\ell-1})\right]
$$

$$
\sim \frac{1}{\sqrt{N}}U_\ell \cdot \frac{1}{\sqrt{LN}}\left(-\eta'_W\mathcal{L}'\frac{1}{\sqrt{LN}}dh_{\ell-1}\phi(x_{\ell-1})^\top\right)\phi(\bar{x}_{\ell-1})
$$

$$
= -\frac{1}{L}\eta'_W\mathcal{L}'\frac{\langle\phi(x_{\ell-1}), \phi(\bar{x}_{\ell-1})\rangle}{N}\frac{1}{\sqrt{N}}U_\ell dh_{\ell-1}
$$

$$
= -\frac{1}{L}\eta'_W\mathcal{L}'\frac{\langle\phi(x_{\ell-1}), \phi(\bar{x}_{\ell-1})\rangle}{N}\frac{1}{\sqrt{N}}U_\ell\left(dh_\ell + \frac{1}{\sqrt{LN}}\widetilde{U}_\ell^\top dx_\ell\right)
$$

Here, because we replace $\widetilde{U}_\ell$ with $U_\ell$, hence, **no** interaction term anymore. Therefore, we obtain

$$
\boxed{Z^{\bar{x}_\ell} = Z^{U_\ell\bar{h}_{\ell-1}} - \frac{1}{\sqrt{L}}\eta'_U\mathring{\mathcal{L}}'\mathbb{E}[Z^{h_{\ell-1}}Z^{\bar{h}_{\ell-1}}]Z^{dx_\ell}.} \tag{23}
$$

**Note** that if $\tilde{U}_\ell$ has shared initialization, then it is not fully decoupled but still has the interaction term

$$
Z^{\bar{x}_\ell} = Z^{U_\ell\bar{h}_{\ell-1}} - \frac{1}{\sqrt{L}}\eta'_U\mathring{\mathcal{L}}'\mathbb{E}[Z^{h_{\ell-1}}Z^{\bar{h}_{\ell-1}}]Z^{dx_\ell} - L^{-\frac{3}{2}}\eta'_W\mathring{\mathcal{L}}'\mathbb{E}[\phi(Z^{x_{\ell-1}})\phi(Z^{\bar{x}_{\ell-1}})]Z^{d\hat{x}_\ell}
$$

Hence, we need to have $\eta'_U = \eta_c\sqrt{L}$ and so $\eta_U = \eta_c\sqrt{L}N$. Moreover, we also have

$$
\bar{h}_\ell = \bar{h}_{\ell-1} + \frac{1}{\sqrt{LN}}(W_\ell + \Delta W_\ell)\phi(\bar{x}_\ell)
$$

$$
= \bar{h}_{\ell-1} + \frac{1}{\sqrt{LN}}W_\ell\phi(\bar{x}_\ell) + \frac{1}{\sqrt{LN}}\Delta W_\ell\phi(\bar{x}_\ell)
$$

$$
= \bar{h}_{\ell-1} + \frac{1}{\sqrt{LN}}W_\ell\phi(\bar{x}_\ell) + \frac{1}{\sqrt{LN}}\left[-\eta'_W\mathcal{L}'\frac{1}{\sqrt{LN}}dh_\ell\phi(x_\ell)^\top\right]\phi(\bar{x}_\ell)
$$

$$
= \bar{h}_{\ell-1} + \frac{1}{\sqrt{LN}}W_\ell\phi(\bar{x}_\ell) - \frac{1}{L}\eta'_W\mathcal{L}'\frac{\langle\phi(x_\ell), \phi(\bar{x}_\ell)\rangle}{N}dh_\ell
$$

where the interaction term stems from

$$
\frac{1}{\sqrt{LN}}W_\ell\phi(\bar{x}_\ell) = \frac{1}{\sqrt{LN}}W_\ell\phi\left(\frac{1}{\sqrt{N}}U_\ell\bar{h}_{\ell-1} - \frac{1}{\sqrt{L}}\eta'_U\mathcal{L}'\frac{\langle h_{\ell-1}, \bar{h}_{\ell-1}\rangle}{N}dx_\ell\right)
$$

$$
\sim \frac{1}{\sqrt{LN}}W_\ell\left[\phi'(\bar{x}_\ell)\odot - \frac{1}{\sqrt{L}}\eta'_U\mathcal{L}'\frac{\langle h_{\ell-1}, \bar{h}_{\ell-1}\rangle}{N}dx_\ell\right]
$$

$$
\sim -\frac{1}{\sqrt{L}}\eta'_U\mathcal{L}'\frac{\langle h_{\ell-1}, \bar{h}_{\ell-1}\rangle}{N}\cdot\frac{1}{\sqrt{LN}}W_\ell\left[\phi'(\bar{x}_\ell)\odot\left(\frac{1}{\sqrt{N}}\widetilde{W}_\ell^\top dh_\ell\odot\phi'(x_\ell)\right)\right]
$$

The independence of $\widetilde{\boldsymbol{W}}_\ell$ and $\boldsymbol{W}_\ell$ implies **no** interaction term anymore. This is direct result from swapping $\boldsymbol{W}_\ell$ and $\boldsymbol{U}_\ell$. Therefore, we obtain

$$\boxed{Z^{\bar{\boldsymbol{h}}_\ell} = Z^{\bar{\boldsymbol{h}}_{\ell-1}} + \frac{1}{\sqrt{L}}Z^{\boldsymbol{W}_\ell\phi(\bar{\boldsymbol{x}}_\ell)} - \frac{1}{L}\eta'_W\mathring{\mathcal{L}}'\mathbb{E}[\phi(Z^{\boldsymbol{x}_\ell})\phi(Z^{\bar{\boldsymbol{x}}_\ell})]Z^{d\boldsymbol{h}_\ell}} \tag{24}$$

Thus, it is fine just using $\eta'_W = \eta_c$. On the other hand, in the shared initialization setup, we have

$$Z^{\bar{\boldsymbol{h}}_\ell} = Z^{\bar{\boldsymbol{h}}_{\ell-1}} + \frac{1}{\sqrt{L}}Z^{\boldsymbol{W}_\ell\phi(\bar{\boldsymbol{x}}_\ell)} - \frac{1}{L}\eta'_W\mathbb{E}[\phi(Z^{\boldsymbol{x}_\ell})\phi(Z^{\bar{\boldsymbol{x}}_\ell})]Z^{d\boldsymbol{h}_\ell} - \frac{1}{L}\eta'_U\mathring{\mathcal{L}}'\mathbb{E}[Z^{\boldsymbol{h}_{\ell-1}}Z^{\bar{\boldsymbol{h}}_{\ell-1}}]\mathbb{E}[\phi'(Z^{\bar{\boldsymbol{x}}_\ell})\phi'(Z^{\boldsymbol{x}_\ell})]Z^{d\boldsymbol{h}_\ell}$$

Clearly, $\eta'_U = \eta_c\sqrt{L}$ results in exploding interaction term in the large-depth limit.

### C.4. Second-Backward Pass

First, observe that

$$\begin{aligned}
d\bar{\boldsymbol{x}}_\ell &= \frac{1}{\sqrt{N}}(\widetilde{\boldsymbol{W}}_\ell + \Delta\widetilde{\boldsymbol{W}}_\ell)^\top d\bar{\boldsymbol{h}}_\ell \odot \phi'(\bar{\boldsymbol{x}}_\ell) \\
&= \left[\frac{1}{\sqrt{N}}\widetilde{\boldsymbol{W}}_\ell^\top d\bar{\boldsymbol{h}}_\ell + \frac{1}{\sqrt{N}}\Delta\widetilde{\boldsymbol{W}}_\ell^\top d\bar{\boldsymbol{h}}_\ell\right] \odot \phi'(\bar{\boldsymbol{x}}_\ell) \\
&= \left[\frac{1}{\sqrt{N}}\widetilde{\boldsymbol{W}}_\ell^\top d\bar{\boldsymbol{h}}_\ell + \frac{1}{\sqrt{N}}\left(-\tilde{\eta}'_W\mathcal{L}'\frac{1}{\sqrt{LN}}d\boldsymbol{h}_\ell\phi(\boldsymbol{x}_\ell)^\top\right)^\top d\bar{\boldsymbol{h}}_\ell\right] \odot \phi'(\bar{\boldsymbol{x}}_\ell) \\
&= \left[\frac{1}{\sqrt{N}}\widetilde{\boldsymbol{W}}_\ell^\top d\bar{\boldsymbol{h}}_\ell - \frac{1}{\sqrt{L}}\tilde{\eta}'_W\mathcal{L}'\frac{\langle d\boldsymbol{h}_\ell, d\bar{\boldsymbol{h}}_\ell\rangle}{N}\phi(\boldsymbol{x}_\ell)\right] \odot \phi'(\bar{\boldsymbol{x}}_\ell)
\end{aligned}$$

where notice that

$$\begin{aligned}
\frac{1}{\sqrt{N}}\widetilde{\boldsymbol{W}}_\ell^\top d\bar{\boldsymbol{h}}_\ell &= \frac{1}{\sqrt{N}}\widetilde{\boldsymbol{W}}_\ell^\top\left[d\bar{\boldsymbol{h}}_{\ell+1} + \frac{1}{\sqrt{LN}}\left(\tilde{\boldsymbol{U}}_{\ell+1} + \Delta\tilde{\boldsymbol{U}}_{\ell+1}\right)^\top d\bar{\boldsymbol{x}}_{\ell+1}\right] \\
&\sim \frac{1}{\sqrt{N}}\widetilde{\boldsymbol{W}}_\ell^\top\frac{1}{\sqrt{LN}}\left(-\tilde{\eta}'_U\mathcal{L}'\frac{1}{\sqrt{LN}}d\boldsymbol{x}_{\ell+1}\boldsymbol{h}_\ell^\top\right)^\top d\bar{\boldsymbol{x}}_{\ell+1} \\
&= -\frac{1}{L}\tilde{\eta}'_U\mathcal{L}'\frac{\langle d\boldsymbol{x}_{\ell+1}, d\bar{\boldsymbol{x}}_{\ell+1}\rangle}{N}\frac{1}{\sqrt{N}}\widetilde{\boldsymbol{W}}_\ell^\top\boldsymbol{h}_\ell \\
&= -\frac{1}{L}\tilde{\eta}'_U\mathcal{L}'\frac{\langle d\boldsymbol{x}_{\ell+1}, d\bar{\boldsymbol{x}}_{\ell+1}\rangle}{N}\frac{1}{\sqrt{N}}\widetilde{\boldsymbol{W}}_\ell^\top\left(\boldsymbol{h}_{\ell-1} + \frac{1}{\sqrt{LN}}\boldsymbol{W}_\ell\phi\left(\boldsymbol{x}_\ell\right)\right) \\
&\sim -L^{-\frac{3}{2}}\tilde{\eta}'_U\mathcal{L}'\frac{\langle d\boldsymbol{x}_{\ell+1}, d\bar{\boldsymbol{x}}_{\ell+1}\rangle}{N}\frac{1}{N}\widetilde{\boldsymbol{W}}_\ell^\top\boldsymbol{W}_\ell\phi(\boldsymbol{x}_\ell)
\end{aligned}$$

implies no interaction term. Hence, the characterization is

$$\boxed{Z^{d\bar{\boldsymbol{x}}_\ell} = \left[Z^{\widetilde{\boldsymbol{W}}_\ell^\top d\bar{\boldsymbol{h}}_\ell} - \frac{1}{\sqrt{L}}\tilde{\eta}'_W\mathcal{L}'\mathbb{E}[Z^{d\boldsymbol{h}_\ell}Z^{d\bar{\boldsymbol{h}}_\ell}]\phi(Z^{\boldsymbol{x}_\ell})\right]\phi'(Z^{\bar{\boldsymbol{x}}_\ell})} \tag{25}$$

Hence, we need to have $\tilde{\eta}'_W = \eta_c\sqrt{L}$ and so $\tilde{\eta}_W = \eta_c\sqrt{L}N$. In the case of shared initialization, we have

$$Z^{d\bar{\boldsymbol{x}}_\ell} = \left[Z^{\widetilde{\boldsymbol{W}}_\ell^\top d\bar{\boldsymbol{h}}_\ell} - \frac{1}{\sqrt{L}}\tilde{\eta}'_W\mathcal{L}'\mathbb{E}[Z^{d\boldsymbol{h}_\ell}Z^{d\bar{\boldsymbol{h}}_\ell}\phi(Z^{\boldsymbol{x}_\ell})] - L^{-\frac{3}{2}}\tilde{\eta}'_U\mathring{\mathcal{L}}'\mathbb{E}[Z^{d\boldsymbol{x}_{\ell+1}}Z^{d\bar{\boldsymbol{x}}_{\ell+1}}]\phi(Z^{\boldsymbol{x}_\ell})\right]\phi'(Z^{\bar{\boldsymbol{x}}_\ell})$$

Then we can consider

$$d\bar{\boldsymbol{h}}_{\ell-1} = d\bar{\boldsymbol{h}}_\ell + \frac{1}{\sqrt{LN}}\left(\widetilde{\boldsymbol{U}}_\ell + \Delta\widetilde{\boldsymbol{U}}\right)^\top d\bar{\boldsymbol{x}}_\ell$$

$$= d\bar{\boldsymbol{h}}_\ell + \frac{1}{\sqrt{LN}}\widetilde{\boldsymbol{U}}_\ell^\top d\bar{\boldsymbol{x}}_\ell + \frac{1}{\sqrt{LN}}\Delta\widetilde{\boldsymbol{U}}^\top d\bar{\boldsymbol{x}}_\ell$$

$$= d\bar{\boldsymbol{h}}_\ell + \frac{1}{\sqrt{LN}}\widetilde{\boldsymbol{U}}_\ell^\top d\bar{\boldsymbol{x}}_\ell + \frac{1}{\sqrt{LN}}\left(-\widetilde{\eta}_U'\mathcal{L}'\frac{1}{\sqrt{LN}}d\boldsymbol{x}_\ell\boldsymbol{h}_{\ell-1}^\top\right)^\top d\bar{\boldsymbol{x}}_\ell$$

$$= d\bar{\boldsymbol{h}}_\ell + \frac{1}{\sqrt{LN}}\widetilde{\boldsymbol{U}}_\ell^\top d\bar{\boldsymbol{x}}_\ell - \frac{1}{L}\widetilde{\eta}_U'\mathcal{L}'\frac{\langle d\boldsymbol{x}_\ell, d\bar{\boldsymbol{x}}_\ell\rangle}{N}\boldsymbol{h}_{\ell-1}$$

Observe that

$$\frac{1}{\sqrt{LN}}\widetilde{\boldsymbol{U}}_\ell^\top d\bar{\boldsymbol{x}}_\ell = \frac{1}{\sqrt{LN}}\widetilde{\boldsymbol{U}}_\ell^\top\left[\frac{1}{\sqrt{N}}(\widetilde{\boldsymbol{W}}_\ell + \Delta\widetilde{\boldsymbol{W}}_\ell)^\top d\bar{\boldsymbol{h}}_\ell \odot \phi'(\bar{\boldsymbol{x}}_\ell)\right]$$

$$\sim \frac{1}{\sqrt{LN}}\widetilde{\boldsymbol{U}}_\ell^\top\left[\frac{1}{\sqrt{N}}\left(-\widetilde{\eta}_W'\mathcal{L}'\frac{1}{\sqrt{LN}}d\boldsymbol{h}_\ell\phi(\boldsymbol{x}_\ell)^\top\right)d\bar{\boldsymbol{h}}_\ell \odot \phi'(\bar{\boldsymbol{x}}_\ell)\right]$$

$$= -\frac{1}{L}\widetilde{\eta}_W'\mathcal{L}'\frac{\langle d\boldsymbol{h}_\ell, d\bar{\boldsymbol{h}}_\ell\rangle}{N}\frac{1}{\sqrt{N}}\widetilde{\boldsymbol{U}}_\ell^\top[\phi(\boldsymbol{x}_\ell) \odot \phi'(\bar{\boldsymbol{x}}_\ell)]$$

$$= -\frac{1}{L}\widetilde{\eta}_W'\mathcal{L}'\frac{\langle d\boldsymbol{h}_\ell, d\bar{\boldsymbol{h}}_\ell\rangle}{N}\frac{1}{\sqrt{N}}\widetilde{\boldsymbol{U}}_\ell^\top\left[\phi\left(\frac{1}{\sqrt{N}}\boldsymbol{U}_\ell\boldsymbol{h}_{\ell-1}\right) \odot \phi'(\bar{\boldsymbol{x}}_\ell)\right].$$

Hence, no interaction terms are introduced, and we have

$$\boxed{Z^{d\bar{\boldsymbol{h}}_{\ell-1}} = Z^{d\bar{\boldsymbol{h}}_\ell} + \frac{1}{\sqrt{L}}Z^{\widetilde{\boldsymbol{U}}_\ell^\top d\bar{\boldsymbol{x}}_\ell} - \frac{1}{L}\widetilde{\eta}_U'\mathring{\mathcal{L}}'\mathbb{E}[Z^{d\boldsymbol{x}_\ell}Z_\ell^{d\bar{\boldsymbol{x}}}]Z^{\boldsymbol{h}_{\ell-1}}} \tag{26}$$

If share initialization is used, we have

$$Z^{d\bar{\boldsymbol{h}}_{\ell-1}} = Z^{d\bar{\boldsymbol{h}}_\ell} + \frac{1}{\sqrt{L}}Z^{\widetilde{\boldsymbol{U}}_\ell^\top d\bar{\boldsymbol{x}}_\ell} - \frac{1}{L}\widetilde{\eta}_U'\mathring{\mathcal{L}}'\mathbb{E}[Z^{d\boldsymbol{x}_\ell}Z_\ell^{d\bar{\boldsymbol{x}}}]Z^{\boldsymbol{h}_{\ell-1}} \underbrace{- \frac{1}{L}\widetilde{\eta}_W'\mathring{\mathcal{L}}'\mathbb{E}[Z^{d\boldsymbol{h}_\ell}Z^{d\bar{\boldsymbol{h}}_\ell}]\mathbb{E}[\phi'(Z^{\boldsymbol{x}_\ell})\phi'(Z^{\bar{\boldsymbol{x}}_\ell})]Z^{\boldsymbol{h}_{\ell-1}}}_{\text{interaction}}$$

Hence, it is fine just using $\tilde{\eta}_U' = \eta_c$. In summary, we have learning rates

$$\boxed{\eta_U = \eta_c\sqrt{L}N, \quad \eta_W = \eta_c N, \quad \tilde{\eta}_U = \eta_c N, \quad \tilde{\eta}_W = \eta_c\sqrt{L}N.} \tag{27}$$

Additionally, it is worth noting that only decoupling $\widetilde{\boldsymbol{W}}_\ell$ or $\widetilde{\boldsymbol{U}}_\ell$ does not work as the learning $\eta_U'$ governs the interaction term in forward $\boldsymbol{h}_t$ and the learning term in the backward $d\boldsymbol{h}_t$. Hence, only decoupling one case only solve partial problems.

### C.5. Convergence of first two forwards and backwards

We note that compared with derivations in Sections B.1 and B.2, the evolution of first forward and first backward is exactly the same. Therefore the analysis and results in Section B.5 are still valid, such as the rate of convergence obtained in Eq. (12) and Eq. (14).

We also note that compared with derivations in Sections B.3 and B.4, the evolution of second forward and second backward is much simpler, with no interaction terms involved. The only additional feature is that the learning rates $\eta_U$ and $\tilde{\eta}_W$ are scaled up by $\sqrt{L}$, which will lead to some non-vanishing terms in the limit as $L \to \infty$. Therefore, we will follow the analysis in Section B.6, highlight key differences, and state analogous results as follows.

We start with the infinite-width limit given in Sections C.1, C.2, C.3 (in particular Eq. (23) and Eq. (24)) and C.4 (in particular Eq. (25), and Eq. (26)) with the learning rate given in Eq. (27). The limit as $N \to \infty$ for the first two forwards

and backwards can be rewritten as

$$x_\ell^{(0),L} = z_\ell^{(0),\boldsymbol{U}},$$

$$h_\ell^{(0),L} = h_{\ell-1}^{(0),L} + \frac{1}{\sqrt{L}} z_\ell^{(0),\boldsymbol{W}},$$

$$\delta x_\ell^{(0),L} = \phi'(x_\ell^{(0),L}) z_\ell^{(0),\widetilde{\boldsymbol{W}}},$$

$$\delta h_{\ell-1}^{(0),L} = \delta h_\ell^{(0),L} + \frac{1}{\sqrt{L}} z_\ell^{(0),\widetilde{\boldsymbol{U}}},$$

$$x_\ell^{(1),L} = z_\ell^{(1),\boldsymbol{U}} - \eta_c \mathring{\mathcal{L}}' \mathbb{E}[h_{\ell-1}^{(0),L} h_{\ell-1}^{(1),L}] \phi'(x_\ell^{(0),L}) \delta x_\ell^{(0),L},$$

$$h_\ell^{(1),L} = h_{\ell-1}^{(1),L} + \frac{1}{\sqrt{L}} z_\ell^{(1),\boldsymbol{W}} - \frac{1}{L} \eta_c \mathring{\mathcal{L}}' \mathbb{E}[\phi(x_\ell^{(0),L}) \phi(x_\ell^{(1),L})] \delta h_\ell^{(0),L},$$

$$\delta x_\ell^{(1),L} = \phi'(x_\ell^{(1),L}) z_\ell^{(1),\widetilde{\boldsymbol{W}}} - \eta_c \mathring{\mathcal{L}}' \mathbb{E}[\delta h_\ell^{(0),L} \delta h_\ell^{(1),L}] \phi'(x_\ell^{(1),L}) \phi(x_\ell^{(0),L}),$$

$$\delta h_{\ell-1}^{(1),L} = \delta h_\ell^{(1),L} + \frac{1}{\sqrt{L}} z_\ell^{(1),\widetilde{\boldsymbol{U}}} - \frac{1}{L} \eta_c \mathring{\mathcal{L}}' \mathbb{E}[\delta x_\ell^{(0),L} \delta x_\ell^{(1),L}] h_{\ell-1}^{(0),L},$$

where $(z_\ell^{(0),\boldsymbol{W}}, z_\ell^{(1),\boldsymbol{W}}), (z_\ell^{(0),\widetilde{\boldsymbol{W}}}, z_\ell^{(1),\widetilde{\boldsymbol{W}}}), (z_\ell^{(0),\boldsymbol{U}}, z_\ell^{(1),\boldsymbol{U}}), (z_\ell^{(0),\widetilde{\boldsymbol{U}}}, z_\ell^{(1),\widetilde{\boldsymbol{U}}})$ are independent Gaussian random vectors with mean 0 and variance-covariance matrix

$$\mathrm{Cov}(z_\ell^{(k),\boldsymbol{W}}, z_\ell^{(k'),\boldsymbol{W}}) = \mathbb{E}[\phi(x_\ell^{(k),L}) \phi(x_\ell^{(k'),L})],$$

$$\mathrm{Cov}(z_\ell^{(k),\widetilde{\boldsymbol{W}}}, z_\ell^{(k'),\widetilde{\boldsymbol{W}}}) = \mathbb{E}[\delta h_\ell^{(k),L} \delta h_\ell^{(k'),L}],$$

$$\mathrm{Cov}(z_\ell^{(k),\boldsymbol{U}}, z_\ell^{(k'),\boldsymbol{U}}) = \mathbb{E}[h_{\ell-1}^{(k),L} h_{\ell-1}^{(k'),L}],$$

$$\mathrm{Cov}(z_\ell^{(k),\widetilde{\boldsymbol{U}}}, z_\ell^{(k'),\widetilde{\boldsymbol{U}}}) = \mathbb{E}[\delta x_\ell^{(k),L} \delta x_\ell^{(k'),L}].$$

Letting $L \to \infty$, we expect to have

$$x_t^{(0)} = u_t^{(0)}, \tag{28}$$

$$dh_t^{(0)} = dw_t^{(0)}, \tag{29}$$

$$\delta x_t^{(0)} = \phi'(x_t^{(0)}) \tilde{w}_t^{(0)}, \tag{30}$$

$$d(\delta h_t^{(0)}) = d\tilde{u}_t^{(0)}, \tag{31}$$

$$x_t^{(1)} = u_t^{(1)} - \eta_c \mathring{\mathcal{L}}' \mathbb{E}[h_t^{(0)} h_t^{(1)}] \phi'(x_t^{(0)}) \delta x_t^{(0)}, \tag{32}$$

$$dh_t^{(1)} = dw_t^{(1)} - \eta_c \mathring{\mathcal{L}}' \mathbb{E}[\phi(x_t^{(0)}) \phi(x_t^{(1)})] \delta h_t^{(0)} \, dt, \tag{33}$$

$$\delta x_t^{(1)} = \phi'(x_t^{(1)}) \tilde{w}_t^{(1)} - \eta_c \mathring{\mathcal{L}}' \mathbb{E}[\delta h_t^{(0)} \delta h_t^{(1)}] \phi'(x_t^{(1)}) \phi(x_t^{(0)}), \tag{34}$$

$$d(\delta h_t^{(1)}) = d\tilde{u}_t^{(1)} - \eta_U' \mathring{\mathcal{L}}' \mathbb{E}[\delta x_t^{(0)} \delta x_t^{(1)}] h_t^{(0)} \, dt, \tag{35}$$

where $\{(w_t^{(0)}, w_t^{(1)}), (\tilde{u}_t^{(0)}, \tilde{u}_t^{(1)})\}$ are independent Brownian motions with mean 0 and cross-variations

$$d\langle w^{(k)}, w^{(k')} \rangle_t = \mathbb{E}[\phi(x_t^{(k)}) \phi(x_t^{(k')})] \, dt,$$

$$d\langle \tilde{u}^{(k)}, \tilde{u}^{(k')} \rangle_t = \mathbb{E}[\delta x_t^{(k)} \delta x_t^{(k')}] \, dt,$$

and $\{(u_t^{(0)}, u_t^{(1)}), (\tilde{w}_t^{(0)}, \tilde{w}_t^{(1)})\}$ are independent Gaussian random vectors with mean 0 and covariances

$$\mathrm{Cov}(u^{(k)}, u^{(k')}) = \mathbb{E}[h_t^{(k)} h_t^{(k')}],$$

$$\mathrm{Cov}(\tilde{w}^{(k)}, \tilde{w}^{(k')}) = \mathbb{E}[\delta h_t^{(k)} \delta h_t^{(k')}].$$

We note that here the evolution of $\delta h_t^{(0)}$ and $\delta h_t^{(1)}$ is interpreted backward from $t = 1$ to $t = 0$.

We have the following theorem quantifying the rate of convergence as $L \to \infty$. The proof is similar to that of Theorem B.10 and (Yao et al., 2025, Proposition G.6), and hence is omitted.

**Theorem C.1.** *Assume that*

- *$\mathring{\mathcal{L}}'$, $\phi$, $\phi'$ are Lipschitz continuous;*

- *there exists a solution to the system Eq. (28)–Eq. (35) such that the $2 \times 2$ matrices $(\mathbb{E}[\phi(x_t^{(k)})\phi(x_t^{(k')})])$ and $(\mathbb{E}[\delta x_t^{(k)} \delta x_t^{(k')}])$ are uniformly strictly positive definite, that is, the smallest eigenvalue is always great than some $\varepsilon > 0$.*

*Then for all $L \geq 1$, for each of the variable $\theta = x, h, \delta x, \delta h$, we have*

$$\sup_{\ell=0,1,\ldots,L} \mathbb{E}\left[|\theta_\ell^{(0),L} - \theta_{\ell/L}^{(0)}|^2 + |\theta_\ell^{(1),L} - \theta_{\ell/L}^{(1)}|^2\right] \leq C/L.$$

## D. Attention Layer: ViT-Type

Consider $\{h_s^0\}_s$ the embeddings added the position embeddings. Additionally, the first token will be the classification token. Then

$$q_{\ell,s} = \frac{1}{\sqrt{N}} W_{\ell,q} h_{\ell-1,s}$$

$$k_{\ell,s} = \frac{1}{\sqrt{N}} W_{\ell,k} h_{\ell-1,s}$$

$$v_{\ell,s} = \frac{1}{\sqrt{N}} W_{\ell,v} h_{\ell-1,s}$$

$$b_{\ell,ss'} = \langle q_s, k_{s'} \rangle / N$$

$$a_{\ell,ss'} = \frac{e^{b_{ss'}}}{\sum_{s''} e^{b_{ss''}}}$$

$$o_{\ell,s} = \sum_{s'} a_{ss'} v_{\ell,s'}$$

$$h_{\ell,s} = h_{\ell-1,s} + \frac{1}{\sqrt{LN}} W_{\ell,o} o_{\ell,s}.$$

Note that only CLS is directly supervised; other tokens receive gradients indirectly via attention.

$$f = \frac{1}{N} v^\top h_{L,1}.$$

Then we have

$$\frac{\partial f}{\partial h_{L,s}} = \delta_{s,1} \frac{1}{N} v$$

**Note**, only the first (CLS) token has the tunnel to back propagate the error via the final representation $f$, since only the CLS token is used in $f$. For any variable $\theta_{\ell,s}$, the forward propagation implies that

$$\frac{\partial f}{\partial \theta_{\ell,s}} = \sum_{s'} \frac{\partial f}{\partial h_{\ell,s'}} \cdot \frac{\partial h_{\ell,s'}}{\partial \theta_{\ell,s}}$$

Then we have

$$\partial_{o_{\ell,s}} f = \sum_{s'} \delta_{ss'} \frac{1}{\sqrt{LN}} W_{\ell,o}^\top \frac{\partial f}{\partial h_{\ell,s'}} = \frac{1}{\sqrt{LN}} W_{\ell,o}^\top \frac{\partial f}{\partial h_{\ell,s}}$$

where we use the fact that $o_{\ell,s}$ is only used to compute $h_{\ell,s}$. As $v_{\ell,s}$ is used to calcuate all $z_{\ell,s'}$, we have

$$\frac{\partial f}{\partial v_{\ell,s}} = \sum_{s'} \frac{\partial f}{\partial z_{\ell,s'}} a_{s',s} = \sum_{s'} \frac{1}{\sqrt{LN}} W_{\ell,o}^\top \frac{\partial f}{\partial h_{\ell,s'}} a_{s',s}$$

As $\boldsymbol{q}_{\ell,s}$ is only used to compute $\boldsymbol{h}_{\ell,s}$ and $\boldsymbol{o}_{\ell,s}$, we have

$$
\begin{aligned}
\frac{\partial f}{\partial \boldsymbol{q}_{\ell,s}} &= \partial_{\boldsymbol{o}_{\ell,s}} f \cdot \frac{\partial \boldsymbol{o}_{\ell,s}}{\partial \boldsymbol{q}_{\ell,s}} \\
&= \partial_{\boldsymbol{o}_{\ell,s}} f \cdot \frac{\partial}{\partial \boldsymbol{q}_{\ell,s}} \left( \sum_{s'} a_{ss'} \boldsymbol{v}_{\ell,s'} \right) \\
&= \partial_{\boldsymbol{o}_{\ell,s}} f \cdot \left( \sum_{s'} \frac{\partial a_{ss'}}{\partial \boldsymbol{q}_{\ell,s}} \boldsymbol{v}_{\ell,s'} \right) \\
&= \sum_{s'} \left\langle \partial_{\boldsymbol{o}_{\ell,s}} f, \boldsymbol{v}_{\ell,s'} \right\rangle \sum_{s''} \frac{\partial a_{ss'}}{\partial b_{\ell,ss''}} \frac{\partial b_{\ell,ss''}}{\partial \boldsymbol{q}_{\ell,s}} \\
&= \sum_{s'} \frac{\left\langle \partial_{\boldsymbol{o}_{\ell,s}} f, \boldsymbol{v}_{\ell,s'} \right\rangle}{N} \sum_{s''} a_{ss'}(\delta_{s's''} - a_{\ell,ss''}) \boldsymbol{k}_{\ell,s''} \\
&= \sum_{s'} \frac{\left\langle \partial_{\boldsymbol{o}_{\ell,s}} f, \boldsymbol{v}_{\ell,s'} - \boldsymbol{o}_{\ell,s} \right\rangle}{N} a_{\ell,ss'} \boldsymbol{k}_{\ell,s'},
\end{aligned}
$$

where we use the fact $\boldsymbol{o}_{\ell,s} = \sum_{s'} a_{ss'} \boldsymbol{v}_{\ell,s'}$. As $\boldsymbol{k}_{\ell,s}$ is used to calculate all $\boldsymbol{z}_{\ell,s'}$, we have

$$
\begin{aligned}
\frac{\partial f}{\partial \boldsymbol{k}_{\ell,s}} &= \sum_{s'} \partial_{\boldsymbol{z}_{\ell,s'}} f \cdot \frac{\partial \boldsymbol{z}_{\ell,s'}}{\partial \boldsymbol{k}_{\ell,s}} \\
&= \sum_{s'} \partial_{\boldsymbol{z}_{\ell,s'}} f \cdot \frac{\partial}{\partial \boldsymbol{k}_{\ell,s}} \left( \sum_{s''} a_{s',s''} \boldsymbol{v}_{\ell,s''} \right) \\
&= \sum_{s'} \sum_{s''} \left\langle \partial_{\boldsymbol{z}_{\ell,s'}} f, \boldsymbol{v}_{\ell,s''} \right\rangle \frac{\partial a_{s',s''}}{\partial \boldsymbol{k}_{\ell,s}} \\
&= \sum_{s'} \sum_{s''} \left\langle \partial_{\boldsymbol{z}_{\ell,s'}} f, \boldsymbol{v}_{\ell,s''} \right\rangle \sum_{s'''} \frac{\partial a_{s',s''}}{\partial b_{s',s'''}} \frac{\partial b_{s',s'''}}{\partial \boldsymbol{k}_{\ell,s}} \\
&= \sum_{s'} \sum_{s''} \frac{\left\langle \partial_{\boldsymbol{z}_{\ell,s'}} f, \boldsymbol{v}_{\ell,s''} \right\rangle}{N} \sum_{s'''} \frac{\partial a_{s',s''}}{\partial b_{s',s'''}} \cdot \delta_{s''',s} \cdot \boldsymbol{q}_{\ell,s'} \\
&= \sum_{s'} \sum_{s''} \frac{\left\langle \partial_{\boldsymbol{z}_{\ell,s'}} f, \boldsymbol{v}_{\ell,s''} \right\rangle}{N} \sum_{s'''} a_{s',s''}(\delta_{s'',s'''} - a_{s',s'''}) \cdot \delta_{s''',s} \cdot \boldsymbol{q}_{\ell,s'} \\
&= \sum_{s'} \sum_{s''} \frac{\left\langle \partial_{\boldsymbol{z}_{\ell,s'}} f, \boldsymbol{v}_{\ell,s''} \right\rangle}{N} a_{s',s''}(\delta_{s'',s} - a_{s',s}) \cdot \boldsymbol{q}_{\ell,s'} \\
&= \sum_{s'} \frac{\left\langle \partial_{\boldsymbol{z}_{\ell,s'}} f, \boldsymbol{v}_{\ell,s} - \boldsymbol{z}_{\ell,s'} \right\rangle}{N} a_{s's} \boldsymbol{q}_{\ell,s'}
\end{aligned}
$$

Finally, we obtain

$$
\begin{aligned}
\frac{\partial f}{\partial \boldsymbol{h}_{\ell-1,s}} &= \frac{\partial f}{\partial \boldsymbol{h}_{\ell,s}} \frac{\partial \boldsymbol{h}_{\ell,s}}{\partial \boldsymbol{h}_{\ell-1,s}} + \frac{\partial f}{\partial \boldsymbol{q}_{\ell,s}} \cdot \frac{\partial \boldsymbol{q}_{\ell,s}}{\partial \boldsymbol{h}_{\ell-1,s}} + \frac{\partial f}{\partial \boldsymbol{k}_{\ell,s}} \frac{\partial \boldsymbol{k}_{\ell,s}}{\partial \boldsymbol{h}_{\ell-1,s}} + \frac{\partial f}{\partial \boldsymbol{v}_{\ell,s}} \frac{\partial \boldsymbol{v}_{\ell,s}}{\partial \boldsymbol{h}_{\ell-1,s}} \\
&= \frac{\partial f}{\partial \boldsymbol{h}_{\ell,s}} + \frac{1}{\sqrt{N}} \boldsymbol{W}_{\ell,q}^{\top} \frac{\partial f}{\partial \boldsymbol{q}_{\ell,s}} + \frac{1}{\sqrt{N}} \boldsymbol{W}_{\ell,k}^{\top} \frac{\partial f}{\partial \boldsymbol{k}_{\ell,s}} + \frac{1}{\sqrt{N}} \boldsymbol{W}_{\ell,v}^{\top} \frac{\partial f}{\partial \boldsymbol{v}_{\ell,s}}
\end{aligned}
$$

To characterize the backward pass in the mean-field limit, we consider the following backward flow by scaling with gradients with network width $N$ and making the $1/\sqrt{L}$ scaling explicitly:

$$\delta \boldsymbol{h}_{L,s} = \delta_{s,1} \boldsymbol{v}$$

$$\delta \boldsymbol{o}_{\ell,s} = \frac{1}{\sqrt{N}} \boldsymbol{W}_{\ell,o}^\top \delta \boldsymbol{h}_{\ell,s}$$

$$\delta \boldsymbol{v}_{\ell,s} = \sum_{s'} a_{s',s} \delta \boldsymbol{o}_{\ell,s'}$$

$$\delta \boldsymbol{q}_{\ell,s} = \sum_{s'} \frac{\langle \delta \boldsymbol{o}_{\ell,s}, \boldsymbol{v}_{\ell,s'} - \boldsymbol{o}_{\ell,s} \rangle}{N} a_{\ell,ss'} \boldsymbol{k}_{\ell,s'},$$

$$\delta \boldsymbol{k}_{\ell,s} = \sum_{s'} \frac{\langle \delta \boldsymbol{o}_{\ell,s'}, \boldsymbol{v}_{\ell,s} - \boldsymbol{o}_{\ell,s'} \rangle}{N} a_{\ell,s's} \boldsymbol{q}_{\ell,s'}$$

$$\delta \boldsymbol{h}_{\ell-1,s} = \delta \boldsymbol{h}_{\ell,s} + \frac{1}{\sqrt{LN}} \boldsymbol{W}_{\ell,q}^\top \delta \boldsymbol{q}_{\ell,s} + \frac{1}{\sqrt{LN}} \boldsymbol{W}_{\ell,k}^\top \delta \boldsymbol{k}_{\ell,s} + \frac{1}{\sqrt{LN}} \boldsymbol{W}_{\ell,v}^\top \delta \boldsymbol{v}_{\ell,s}$$

Using the backward flow, we can further define

$$\delta \boldsymbol{W}_{\ell,o} = \sum_s \frac{1}{\sqrt{LN}} \delta \boldsymbol{h}_{\ell,s} \boldsymbol{o}_{\ell,s}^\top$$

$$\delta \boldsymbol{W}_{\ell,q} = \sum_s \frac{1}{\sqrt{LN}} \delta \boldsymbol{q}_{\ell,s} \boldsymbol{h}_{\ell-1,s}^\top$$

$$\delta \boldsymbol{W}_{\ell,k} = \sum_s \frac{1}{\sqrt{LN}} \delta \boldsymbol{k}_{\ell,s} \boldsymbol{h}_{\ell-1,s}^\top$$

$$\delta \boldsymbol{W}_{\ell,v} = \sum_s \frac{1}{\sqrt{LN}} \delta \boldsymbol{v}_{\ell,s} \boldsymbol{h}_{\ell-1,s}^\top$$

Using these quantities, the gradient descent update rules can be written as follows:

$$\boldsymbol{W}_{\ell,o}^+ = \boldsymbol{W}_{\ell,o} - \frac{\eta_o}{N} \mathcal{L}' \cdot \sum_s \frac{1}{\sqrt{LN}} \delta \boldsymbol{h}_{\ell,s} \boldsymbol{o}_{\ell,s}^\top$$

$$\boldsymbol{W}_{\ell,q}^+ = \boldsymbol{W}_{\ell,q} - \frac{\eta_q}{N} \mathcal{L}' \cdot \sum_s \frac{1}{\sqrt{LN}} \delta \boldsymbol{q}_{\ell,s} \boldsymbol{h}_{\ell-1,s}^\top$$

$$\boldsymbol{W}_{\ell,k}^+ = \boldsymbol{W}_{\ell,k} - \frac{\eta_k}{N} \mathcal{L}' \cdot \sum_s \frac{1}{\sqrt{LN}} \delta \boldsymbol{k}_{\ell,s} \boldsymbol{h}_{\ell-1,s}^\top$$

$$\boldsymbol{W}_{\ell,v}^+ = \boldsymbol{W}_{\ell,v} - \frac{\eta_v}{N} \mathcal{L}' \cdot \sum_s \frac{1}{\sqrt{LN}} \delta \boldsymbol{v}_{\ell,s} \boldsymbol{h}_{\ell-1,s}^\top$$

Here by using $\mu$P parameterization, the learning rate has $\eta_x = \eta_x' N$ for $x \in \{o, q, k, v\}$.

### D.1. First-Forward Pass

We have

$$Z^{\boldsymbol{q}_{\ell,s}} = Z^{\widetilde{\boldsymbol{q}}_{\ell,s}} \sim \mathcal{N}(0, \mathbb{E}[(Z^{\boldsymbol{h}_{\ell-1,s}})^2])$$

$$Z^{\boldsymbol{k}_{\ell,s}} = Z^{\widetilde{\boldsymbol{k}}_{\ell,s}} \sim \mathcal{N}(0, \mathbb{E}[(Z^{\boldsymbol{h}_{\ell-1,s}})^2])$$

$$Z^{\boldsymbol{v}_{\ell,s}} = Z^{\widetilde{\boldsymbol{v}}_{\ell,s}} \sim \mathcal{N}(0, \mathbb{E}[(Z^{\boldsymbol{h}_{\ell-1,s}})^2])$$

$$\mathring{b}_{\ell,ss'} = \mathbb{E}[Z^{\boldsymbol{q}_{\ell,s}} Z^{\boldsymbol{k}_{\ell,s'}}] = 0$$

$$\mathring{a}_{\ell,ss'} = \frac{e^{\mathring{b}_{ss'}}}{\sum_{s''} e^{b_{\ell,ss''}}} = \frac{1}{S}$$

$$Z^{\boldsymbol{o}_{\ell,s}} = \sum_{s'} \mathring{a}_{ss'} Z^{\boldsymbol{v}_{\ell,s'}}$$

$$Z^{\boldsymbol{h}_{\ell,s}} = Z^{\boldsymbol{h}_{\ell-1,s}} + \frac{1}{\sqrt{L}} Z^{\widetilde{\boldsymbol{h}}_{\ell,s}}, \quad Z^{\widetilde{\boldsymbol{h}}_{\ell,s}} \sim \mathcal{N}(0, \mathbb{E}[(Z^{\boldsymbol{o}_{\ell,s}})^2])$$

### D.2. First-Backward Pass

Then we can obtain

$$Z^{\delta\boldsymbol{h}_{L,s}} = \delta_{s,1} Z^{\boldsymbol{v}}, \quad Z^{\boldsymbol{v}} \sim \mathcal{N}(0,1)$$

$$Z^{\delta\boldsymbol{o}_{\ell,s}} = Z^{\delta\widetilde{\boldsymbol{o}}_{\ell,s}} \sim \mathcal{N}(0, \mathbb{E}[(Z^{\delta\boldsymbol{h}_{\ell,s}})^2])$$

$$Z^{\delta\boldsymbol{v}_{\ell,s}} = \sum_{s'} \mathring{a}_{\ell,s's} Z^{\delta\boldsymbol{z}_{\ell,s'}}$$

$$Z^{\delta\boldsymbol{q}_{\ell,s}} = \sum_{s'} \mathbb{E}[Z^{\delta\boldsymbol{o}_{\ell,s}}(Z^{\boldsymbol{v}_{\ell,s'}} - Z^{\boldsymbol{o}_{\ell,s}})]\mathring{a}_{\ell,ss'} Z^{\boldsymbol{k}_{\ell,s'}}$$

$$Z^{\delta\boldsymbol{k}_{\ell,s}} = \sum_{s'} \mathbb{E}[Z^{\delta\boldsymbol{o}_{\ell,s'}}(Z^{\boldsymbol{v}_{\ell,s}} - Z^{\boldsymbol{o}_{\ell,s'}})]\mathring{a}_{\ell,s's} Z^{\boldsymbol{q}_{\ell,s'}}$$

$$Z^{\delta\boldsymbol{h}_{\ell-1,s}} = Z^{\delta\boldsymbol{h}_{\ell,s}} + \frac{1}{\sqrt{L}} Z^{\delta\widetilde{\boldsymbol{h}}_{\ell,s}}, \quad Z^{\delta\widetilde{\boldsymbol{h}}_{\ell,s}} \sim \mathcal{N}(0, \mathbb{E}[(Z^{\delta\boldsymbol{q}_{\ell,s}})^2] + \mathbb{E}[(Z^{\delta\boldsymbol{k}_{\ell,s}})^2] + \mathbb{E}[(Z^{\delta\boldsymbol{v}_{\ell,s}})^2])$$

### D.3. Second-Forward Pass

As we apply $\mu$P scaling, we set $\eta'_x = \eta_x/N$ for all $x \in \{q, k, v, o\}$ in the analysis.

**Dynamics of internal representation**   We have

$$\bar{\boldsymbol{q}}_{\ell,s} = \frac{1}{\sqrt{N}}(\boldsymbol{W}_{\ell,q} + \Delta\boldsymbol{W}_{\ell,q})\bar{\boldsymbol{h}}_{\ell-1,s}$$

$$= \frac{1}{\sqrt{N}}\boldsymbol{W}_{\ell,q}\bar{\boldsymbol{h}}_{\ell-1,s} + \frac{1}{\sqrt{N}}\left(-\eta'_q \mathcal{L}' \cdot \sum_{s'} \frac{1}{\sqrt{LN}}\delta\boldsymbol{q}_{\ell,s'}\boldsymbol{h}^\top_{\ell-1,s'}\right)\bar{\boldsymbol{h}}_{\ell-1,s}$$

$$= \frac{1}{\sqrt{N}}\boldsymbol{W}_{\ell,q}\bar{\boldsymbol{h}}_{\ell-1,s} - \frac{1}{\sqrt{L}}\eta'_q \mathcal{L}' \cdot \sum_{s'} \frac{\langle \boldsymbol{h}_{\ell-1,s'}, \bar{\boldsymbol{h}}_{\ell-1,s}\rangle}{N}\delta\boldsymbol{q}_{\ell,s'}$$

The interaction term occurs from

$$\frac{1}{\sqrt{N}}\boldsymbol{W}_{\ell,q}\bar{\boldsymbol{h}}_{\ell-1,s} = \frac{1}{\sqrt{N}}\boldsymbol{W}_{\ell,q}\left[\bar{\boldsymbol{h}}_{\ell-2,s} + \frac{1}{\sqrt{LN}}(\boldsymbol{W}_{\ell,o}+\Delta\boldsymbol{W}_{\ell,o})\bar{\boldsymbol{o}}_{\ell-1,s}\right]$$

$$\sim \frac{1}{\sqrt{N}}\boldsymbol{W}_{\ell,q}\cdot\frac{1}{\sqrt{LN}}\left(-\eta'_o\sum_{s'}\frac{1}{\sqrt{LN}}\delta\boldsymbol{h}_{\ell-1,s'}\boldsymbol{o}_{\ell-1,s'}^{\top}\right)\bar{\boldsymbol{o}}_{\ell-1,s}$$

$$= -\frac{1}{L}\eta'_o\sum_{s'}\frac{\langle\boldsymbol{o}_{\ell-1,s'},\bar{\boldsymbol{o}}_{\ell-1,s}\rangle}{N}\cdot\frac{1}{\sqrt{N}}\boldsymbol{W}_{\ell,q}\delta\boldsymbol{h}_{\ell-1,s'}$$

$$\sim -\frac{1}{L}\eta'_o\sum_{s'}\frac{\langle\boldsymbol{o}_{\ell-1,s'},\bar{\boldsymbol{o}}_{\ell-1,s}\rangle}{N}\cdot\frac{1}{\sqrt{N}}\boldsymbol{W}_{\ell,q}\left(\delta\boldsymbol{h}_{\ell,s'}+\frac{1}{\sqrt{LN}}\boldsymbol{W}_{\ell,q}^{\top}\delta\boldsymbol{q}_{\ell,s'}\right)$$

$$\sim -L^{-\frac{3}{2}}\eta'_o\sum_{s'}\frac{\langle\boldsymbol{o}_{\ell-1,s'},\bar{\boldsymbol{o}}_{\ell-1,s}\rangle}{N}\cdot\frac{1}{N}\boldsymbol{W}_{\ell,q}\boldsymbol{W}_{\ell,q}^{\top}\delta\boldsymbol{q}_{\ell,s'}$$

$$\sim -L^{-\frac{3}{2}}\eta'_o\sum_{s'}\mathbb{E}[Z^{\boldsymbol{o}_{\ell-1,s'}}Z^{\bar{\boldsymbol{o}}_{\ell-1,s}}]\cdot Z^{\delta\boldsymbol{q}_{\ell,s'}}$$

where we use $\mu$P learning rate of $\eta'_o = \eta_o/N$. Therefore, combining the interaction term yields

$$Z^{\boldsymbol{q}_{\ell,s}(1)} = \underbrace{Z^{\widetilde{\boldsymbol{q}}_{\ell,s}(1)}}_{\text{Gaussian}} - \underbrace{\frac{\eta'_q}{\sqrt{L}}\mathring{\mathcal{L}}'\sum_{s'}\mathbb{E}[Z^{\boldsymbol{h}_{\ell-1,s'}}Z^{\bar{\boldsymbol{h}}_{\ell-1,s}}]Z^{\delta\boldsymbol{q}_{\ell,s'}}}_{\text{Learning}} - \underbrace{\frac{\eta'_o}{L^{\frac{3}{2}}}\mathring{\mathcal{L}}'\sum_{s'}\mathbb{E}[Z^{\boldsymbol{o}_{\ell-1,s'}}Z^{\bar{\boldsymbol{o}}_{\ell-1,s}}]Z^{\delta\boldsymbol{q}_{\ell,s'}}}_{\text{Interaction}}$$

where $Z^{\widetilde{\boldsymbol{q}}_{\ell-1,s}(1)} \sim \mathcal{N}(0,\mathbb{E}[(Z^{\boldsymbol{h}_{\ell-1,s}(1)})^2])$.

Using a similar analysis, we also obtain

$$Z^{\boldsymbol{k}_{\ell,s}(1)} = \underbrace{Z^{\widetilde{\boldsymbol{k}}_{\ell,s}(1)}}_{\text{Gaussian}} - \underbrace{\frac{\eta'_k}{\sqrt{L}}\mathring{\mathcal{L}}'\sum_{s'}\mathbb{E}[Z^{\boldsymbol{h}_{\ell-1,s'}}Z^{\bar{\boldsymbol{h}}_{\ell-1,s}}]Z^{\delta\boldsymbol{k}_{\ell,s'}}}_{\text{Learning}} - \underbrace{\frac{\eta'_o}{L^{\frac{3}{2}}}\mathring{\mathcal{L}}'\sum_{s'}\mathbb{E}[Z^{\boldsymbol{o}_{\ell-1,s'}}Z^{\bar{\boldsymbol{o}}_{\ell-1,s}}]Z^{\delta\boldsymbol{k}_{\ell,s'}}}_{\text{Interaction}}$$

$$Z^{\boldsymbol{v}_{\ell,s}(1)} = \underbrace{Z^{\widetilde{\boldsymbol{v}}_{\ell,s}(1)}}_{\text{Gaussian}} - \underbrace{\frac{\eta'_v}{\sqrt{L}}\mathring{\mathcal{L}}'\sum_{s'}\mathbb{E}[Z^{\boldsymbol{h}_{\ell-1,s'}}Z^{\bar{\boldsymbol{h}}_{\ell-1,s}}]Z^{\delta\boldsymbol{v}_{\ell,s'}}}_{\text{Learning}} - \underbrace{\frac{\eta'_o}{L^{\frac{3}{2}}}\mathring{\mathcal{L}}'\sum_{s'}\mathbb{E}[Z^{\boldsymbol{o}_{\ell-1,s'}}Z^{\bar{\boldsymbol{o}}_{\ell-1,s}}]Z^{\delta\boldsymbol{v}_{\ell,s'}}}_{\text{Interaction}}$$

Hence, we can see that the interaction term is $\mathcal{O}(L^{-\frac{3}{2}})$, so in the large-depth limit, those interaction terms all vanish.

**Dynamics of attention logits and scores.** We have

$$b_{\ell,ss'} = \frac{\langle\bar{\boldsymbol{q}}_{\ell,s},\bar{\boldsymbol{k}}_{\ell,s'}\rangle}{N} \sim \mathring{b}_{\ell,ss'} = \mathbb{E}[Z^{\bar{\boldsymbol{q}}_{\ell,s}}Z^{\bar{\boldsymbol{k}}_{\ell,s'}}].$$

Observe that

$$\mathbb{E}[Z^{\bar{\boldsymbol{q}}_{\ell,s}}Z^{\bar{\boldsymbol{k}}_{\ell,s'}}] \sim \mathbb{E}\left[\left(Z^{\widetilde{\boldsymbol{q}}_{\ell,s}^{(1)}} + \frac{\eta'_q}{\sqrt{L}}\sum_{s'}K_{\ell-1,s'}^{(0,1)}Z^{\delta\boldsymbol{q}_{\ell,s'}^{(0)}}\right)\left(Z^{\widetilde{\boldsymbol{k}}_{\ell,s}(1)} + \frac{\eta'_k}{\sqrt{L}}\sum_{s'}K_{\ell-1,s'}^{(0,1)}Z^{\delta\boldsymbol{k}_{\ell,s'}^{(0)}}\right)\right]$$

where $\mathring{K}_{\ell-1,ss'}^{(k,k')} := \mathbb{E}[Z^{\boldsymbol{h}_{\ell-1,s}^{(k)}}Z^{\boldsymbol{h}_{\ell-1,s'}^{(k')}}]$

$$\mathbb{E}[Z^{\widetilde{\boldsymbol{q}}_{\ell,s}(1)}Z^{\widetilde{\boldsymbol{k}}_{\ell,s}(1)}] = 0$$

$$\mathbb{E}[Z^{\delta\boldsymbol{q}_{\ell,s'}(0)}Z^{\widetilde{\boldsymbol{k}}_{\ell,s}(1)}] \sim \sum_{s''}\mathbb{E}[Z^{\boldsymbol{k}_{\ell,s''}(0)}Z^{\widetilde{\boldsymbol{k}}_{\ell,s}(1)}] = \sum_{s''}\mathbb{E}[Z^{\boldsymbol{h}_{\ell-1,s''}}Z^{\bar{\boldsymbol{h}}_{\ell-1,s}}]$$

$$\mathbb{E}[Z^{\delta\boldsymbol{k}_{\ell,s'}(0)}Z^{\widetilde{\boldsymbol{q}}_{\ell,s}(1)}] \sim \sum_{s''}\mathbb{E}[Z^{\boldsymbol{q}_{\ell,s''}(0)}Z^{\widetilde{\boldsymbol{q}}_{\ell,s}(1)}] = \sum_{s''}\mathbb{E}[Z^{\boldsymbol{h}_{\ell-1,s''}}Z^{\bar{\boldsymbol{h}}_{\ell-1,s}}]$$

$$\mathbb{E}[Z^{\delta\boldsymbol{q}_{\ell,s}(0)}Z^{\delta\boldsymbol{k}_{\ell,s'}(0)}] \sim \sum_{t,t'}\mathbb{E}[Z^{\boldsymbol{k}_{\ell,t}(0)}Z^{\boldsymbol{q}_{\ell,t'}(0)}] = 0.$$

where we use the independence of $\boldsymbol{W}_{\ell,q}(0)$ and $\boldsymbol{W}_{\ell,k}(0)$. These asymptotic relation imply that

$$\mathring{b}_{\ell,ss'} = \mathbb{E}[Z^{\bar{\boldsymbol{q}}_{\ell,s}} Z^{\bar{\boldsymbol{k}}_{\ell,s'}}] \sim \mathcal{O}\left(\frac{\eta'_{qk}}{\sqrt{L}}\right)$$

where we set $\eta'_{qk} = \eta'_q + \eta'_k$. Then using Taylor expansion of exponential function yields

$$e^{\mathring{b}_{\ell,ss'}} = 1 + \mathcal{O}\left(\frac{\eta'_{qk}}{\sqrt{L}}\right)$$

which further implies

$$\mathring{a}_{\ell,ss'} = \frac{1}{S} + \mathcal{O}\left(\frac{\eta'_q}{\sqrt{L}}\right).$$

Furthermore, we obtain

$$Z^{\bar{\boldsymbol{o}}_{\ell,s}} = \sum_{s'} \mathring{a}_{\ell,ss'} Z^{\bar{\boldsymbol{v}}_{\ell,s'}}$$

**Dynamics of residual stream** The residual stream gives

$$\bar{\boldsymbol{h}}_{\ell,s} = \bar{\boldsymbol{h}}_{\ell-1,s} + \frac{1}{\sqrt{LN}} \left(\boldsymbol{W}_{\ell,o} + \Delta\boldsymbol{W}_{\ell,o}\right) \bar{\boldsymbol{o}}_{\ell,s}$$

$$= \bar{\boldsymbol{h}}_{\ell-1,s} + \frac{1}{\sqrt{LN}}\boldsymbol{W}_{\ell,o}\bar{\boldsymbol{o}}_{\ell,s} + \frac{1}{\sqrt{LN}}\left[-\eta'_o\mathcal{L}'\sum_{s'}\frac{1}{\sqrt{LN}}\delta\boldsymbol{h}_{\ell,s'}\boldsymbol{o}_{\ell,s'}^\top\right]\bar{\boldsymbol{o}}_{\ell,s}$$

$$= \bar{\boldsymbol{h}}_{\ell-1,s} + \frac{1}{\sqrt{LN}}\boldsymbol{W}_{\ell,o}\bar{\boldsymbol{o}}_{\ell,s} - \frac{1}{L}\eta'_o\mathcal{L}'\sum_{s'}\frac{\langle\boldsymbol{o}_{\ell,s'},\bar{\boldsymbol{o}}_{\ell,s}\rangle}{N}\delta\boldsymbol{h}_{\ell,s'}$$

The interaction term raises from

$$\frac{1}{\sqrt{LN}}\boldsymbol{W}_{\ell,o}\bar{\boldsymbol{o}}_{\ell,s} = \frac{1}{\sqrt{LN}}\boldsymbol{W}_{\ell,o}\sum_{s'}\bar{a}_{\ell,ss'}\bar{\boldsymbol{v}}_{\ell,s'}$$

$$= \frac{1}{\sqrt{LN}}\boldsymbol{W}_{\ell,o}\sum_{s'}\bar{a}_{\ell,ss'}\frac{1}{\sqrt{N}}\left(\boldsymbol{W}_{\ell,v} + \Delta\boldsymbol{W}_{\ell,v}\right)\bar{\boldsymbol{h}}_{\ell-1,s'}$$

$$\sim \frac{1}{\sqrt{LN}}\boldsymbol{W}_{\ell,o}\sum_{s'}\bar{a}_{\ell,ss'}\frac{1}{\sqrt{N}}\left(-\eta'_v\mathcal{L}'\sum_{s''}\frac{1}{\sqrt{LN}}\delta\boldsymbol{v}_{\ell,s''}\boldsymbol{h}_{\ell-1,s''}^\top\right)\bar{\boldsymbol{h}}_{\ell-1,s'}$$

$$= -\frac{1}{L}\eta'_v\mathcal{L}'\sum_{s'}\sum_{s''}\bar{a}_{\ell,ss'}\frac{\langle\boldsymbol{h}_{\ell-1,s''},\bar{\boldsymbol{h}}_{\ell-1,s'}\rangle}{N}\frac{1}{\sqrt{N}}\boldsymbol{W}_{\ell,o}\delta\boldsymbol{v}_{\ell,s''}$$

$$= -\frac{1}{L}\eta'_v\mathcal{L}'\sum_{s'}\sum_{s''}\bar{a}_{\ell,ss'}\frac{\langle\boldsymbol{h}_{\ell-1,s''},\bar{\boldsymbol{h}}_{\ell-1,s'}\rangle}{N}\frac{1}{\sqrt{N}}\boldsymbol{W}_{\ell,o}\left[\sum_{s'''}a_{\ell,s''',s''}\delta\boldsymbol{o}_{\ell,s'''}\right]$$

$$= -\frac{1}{L}\eta'_v\mathcal{L}'\sum_{s'}\sum_{s''}\sum_{s'''}a_{\ell,s'''s''}\bar{a}_{\ell,ss'}\frac{\langle\boldsymbol{h}_{\ell-1,s''},\bar{\boldsymbol{h}}_{\ell-1,s'}\rangle}{N}\left(\frac{1}{N}\boldsymbol{W}_{\ell,o}\boldsymbol{W}_{\ell,o}^\top\delta\boldsymbol{h}_{\ell,s'''}\right)$$

$$\sim -\frac{1}{L}\eta'_v\mathring{\mathcal{L}}'\sum_{s'}\sum_{s''}\sum_{s'''}a_{\ell,s'''s''}\bar{a}_{\ell,ss'}\mathbb{E}[Z^{\boldsymbol{h}_{\ell-1,s''}}Z^{\boldsymbol{h}_{\ell-1,s'}}]Z^{\delta\boldsymbol{h}_{\ell,s'''}}$$

Therefore, the residual stream characterization is given by

$$\underbrace{Z^{\bar{\boldsymbol{h}}_{\ell,s}} - Z^{\bar{\boldsymbol{h}}_{\ell-1,s}}}_{\text{residual stream}} = \underbrace{\frac{1}{\sqrt{L}}Z^{\widetilde{\boldsymbol{h}}_{\ell,s}^{(1)}}}_{\text{Gaussian}} - \underbrace{\frac{\eta'_o}{L}\mathring{\mathcal{L}}'\sum_{s'}\mathbb{E}[Z^{\boldsymbol{o}_{\ell,s'}}Z^{\bar{\boldsymbol{o}}_{\ell,s}}]Z^{\delta\boldsymbol{h}_{\ell,s'}}}_{\text{Learning}}$$

$$\underbrace{-\frac{\eta'_v}{L}\mathring{\mathcal{L}}'\sum_{s'}\sum_{s''}\sum_{s'''}a_{\ell,s'''s''}\bar{a}_{\ell,ss'}\mathbb{E}[Z^{\boldsymbol{h}_{\ell-1,s''}}Z^{\boldsymbol{h}_{\ell-1,s'}}]Z^{d\boldsymbol{h}_{\ell,s'''}}}_{\text{Interaction}}$$

### D.4. Second-Backward Pass

**Dynamics of internal representation**    After one-step gradient descent, we have

$$\delta \bar{\boldsymbol{o}}_{\ell,s} = \frac{1}{\sqrt{N}} \left( \boldsymbol{W}_{\ell,o} + \Delta \boldsymbol{W}_{\ell,o} \right)^\top \delta \bar{\boldsymbol{h}}_{\ell,s}$$

$$= \frac{1}{\sqrt{N}} \boldsymbol{W}_{\ell,o}^\top \delta \boldsymbol{h}_{\ell,s} + \frac{1}{\sqrt{N}} \left( -\eta_o' \mathcal{L}' \sum_{s'} \frac{1}{\sqrt{LN}} \delta \boldsymbol{h}_{\ell,s'} \boldsymbol{o}_{\ell,s'}^\top \right)^\top \delta \boldsymbol{h}_{\ell,s}$$

$$= \frac{1}{\sqrt{N}} \boldsymbol{W}_{\ell,o}^\top \delta \bar{\boldsymbol{h}}_{\ell,s} - \frac{1}{\sqrt{L}} \eta_o' \mathcal{L}' \sum_{s'} \frac{\langle \delta \boldsymbol{h}_{\ell,s'}, \delta \bar{\boldsymbol{h}}_{\ell,s} \rangle}{N} \boldsymbol{o}_{\ell,s}$$

We also need to identify the interaction term from

$$\frac{1}{\sqrt{N}} \boldsymbol{W}_{\ell,o}^\top \delta \bar{\boldsymbol{h}}_{\ell,s}$$

$$\sim \frac{1}{\sqrt{N}} \boldsymbol{W}_{\ell,o}^\top \left( \frac{1}{\sqrt{LN}} \Delta \boldsymbol{W}_{\ell+1,q}^\top \delta \bar{\boldsymbol{q}}_{\ell+1,s} + \frac{1}{\sqrt{LN}} \Delta \boldsymbol{W}_{\ell+1,k}^\top \delta \bar{\boldsymbol{k}}_{\ell+1,s} + \frac{1}{\sqrt{LN}} \Delta \boldsymbol{W}_{\ell+1,v}^\top \delta \bar{\boldsymbol{v}}_{\ell+1,s} \right)$$

Note that

$$\frac{1}{\sqrt{N}} \boldsymbol{W}_{\ell,o}^\top \left( \frac{1}{\sqrt{LN}} \Delta \boldsymbol{W}_{\ell+1,q}^\top \delta \bar{\boldsymbol{q}}_{\ell+1,s} \right)$$

$$= -\frac{1}{L} \eta_q' \mathcal{L}' \sum_{s'} \frac{\langle \delta \boldsymbol{q}_{\ell+1,s'}, \delta \bar{\boldsymbol{q}}_{\ell+1,s} \rangle}{N} \frac{1}{\sqrt{N}} \boldsymbol{W}_{\ell,o}^\top \boldsymbol{h}_{\ell,s'}$$

$$= -\frac{1}{L} \eta_q' \mathcal{L}' \sum_{s'} \frac{\langle \delta \boldsymbol{q}_{\ell+1,s'}, \delta \bar{\boldsymbol{q}}_{\ell+1,s} \rangle}{N} \frac{1}{\sqrt{N}} \boldsymbol{W}_{\ell,o}^\top \boldsymbol{h}_{\ell,s'}$$

$$\sim -L^{-\frac{3}{2}} \eta_q' \mathcal{L}' \sum_{s'} \frac{\langle \delta \boldsymbol{q}_{\ell+1,s'}, \delta \bar{\boldsymbol{q}}_{\ell+1,s} \rangle}{N} \left( \frac{1}{N} \boldsymbol{W}_{\ell,o}^\top \boldsymbol{W}_{\ell,o} \boldsymbol{o}_{\ell,s'} \right)$$

Combining the interactions yields

$$Z^{\delta \bar{\boldsymbol{o}}_{\ell,s}} = Z^{\delta \tilde{\boldsymbol{o}}_{\ell,s}} - \frac{\eta_o'}{\sqrt{L}} \mathring{\mathcal{L}}' \sum_{s} \mathbb{E}[Z^{d\boldsymbol{h}_{\ell,s}} Z^{d\bar{\boldsymbol{h}}_{\ell,s}}] Z^{\boldsymbol{z}_{\ell,s}}$$

$$- \sum_{s'} \mathring{\mathcal{L}}' \left( \frac{\eta_q'}{L^{\frac{3}{2}}} \mathbb{E}[Z^{\delta \boldsymbol{q}_{\ell+1,s'}} Z^{\delta \bar{\boldsymbol{q}}_{\ell+1,s}}] + \frac{\eta_k'}{L^{\frac{3}{2}}} \mathbb{E}[Z^{\delta \boldsymbol{k}_{\ell+1,s'}} Z^{\delta \bar{\boldsymbol{k}}_{\ell+1,s}}] + \frac{\eta_v'}{L^{\frac{3}{2}}} \mathbb{E}[Z^{\delta \boldsymbol{v}_{\ell+1,s'}} Z^{\delta \bar{\boldsymbol{v}}_{\ell+1,s}}] \right) Z^{\boldsymbol{o}_{\ell,s'}}$$

Then we have

$$Z^{\delta \bar{\boldsymbol{v}}_{\ell,s'}} = \sum_{s} \mathring{a}_{\ell,ss'} Z^{\delta \bar{\boldsymbol{o}}_{\ell,s}}$$

$$Z^{\delta \bar{\boldsymbol{q}}_{\ell,s}} = \sum_{s'} \mathbb{E}[Z^{\delta \bar{\boldsymbol{o}}_{\ell,s}} (Z^{\bar{\boldsymbol{v}}_{\ell,s'}} - Z^{\bar{\boldsymbol{o}}_{\ell,s}})] \mathring{a}_{\ell,ss'} Z^{\bar{\boldsymbol{k}}_{\ell,s'}}$$

$$Z^{\delta \bar{\boldsymbol{k}}_{\ell,s}} = \sum_{s'} \mathbb{E}[Z^{\delta \bar{\boldsymbol{o}}_{\ell,s'}} (Z^{\bar{\boldsymbol{v}}_{\ell,s}} - Z^{\bar{\boldsymbol{o}}_{\ell,s'}})] \mathring{a}_{\ell,s's} Z^{\bar{\boldsymbol{q}}_{\ell,s'}}$$

**Dynamics of residual stream.**    Observe that

$$\frac{1}{\sqrt{LN}} (\boldsymbol{W}_{\ell,v} + \Delta \boldsymbol{W}_{\ell,v})^\top \delta \bar{\boldsymbol{v}}_{\ell,s} = \frac{1}{\sqrt{LN}} \boldsymbol{W}_{\ell,v}^\top \delta \bar{\boldsymbol{v}}_{\ell,s} + \frac{1}{\sqrt{LN}} \Delta \boldsymbol{W}_{\ell,v}^\top \delta \bar{\boldsymbol{v}}_{\ell,s}$$

$$= \frac{1}{\sqrt{LN}} \boldsymbol{W}_{\ell,v}^\top \delta \bar{\boldsymbol{v}}_{\ell,s} + \frac{1}{\sqrt{LN}} \left( -\eta_v' \mathcal{L}' \sum_{s'} \frac{1}{\sqrt{LN}} \delta \boldsymbol{v}_{\ell,s'} \boldsymbol{h}_{\ell-1,s'}^\top \right)^\top \delta \bar{\boldsymbol{v}}_{\ell,s}$$

$$= \frac{1}{\sqrt{LN}} \boldsymbol{W}_{\ell,v}^\top \delta \bar{\boldsymbol{v}}_{\ell,s} - \frac{1}{L} \eta_v' \mathcal{L}' \sum_{s'} \frac{\langle \delta \boldsymbol{v}_{\ell,s'}, \delta \bar{\boldsymbol{v}}_{\ell,s} \rangle}{N} \boldsymbol{h}_{\ell-1,s'}$$

Tracking the interaction term

$$\frac{1}{\sqrt{LN}}\boldsymbol{W}_{\ell,\boldsymbol{v}}^{\top}\delta\bar{\boldsymbol{v}}_{\ell,s} = \frac{1}{\sqrt{LN}}\boldsymbol{W}_{\ell,\boldsymbol{v}}^{\top}\sum_{s'}\bar{a}_{\ell,s's}\delta\bar{\boldsymbol{o}}_{\ell,s'}$$

$$\sim \frac{1}{\sqrt{LN}}\boldsymbol{W}_{\ell,\boldsymbol{v}}^{\top}\sum_{s'}\bar{a}_{\ell,s's}\left(\frac{1}{\sqrt{N}}\Delta\boldsymbol{W}_{\ell,o}^{\top}\delta\bar{\boldsymbol{h}}_{\ell,s'}\right)$$

$$= \frac{1}{\sqrt{LN}}\boldsymbol{W}_{\ell,\boldsymbol{v}}^{\top}\sum_{s'}\bar{a}_{\ell,s's}\frac{1}{\sqrt{N}}\left(-\eta_o'\mathcal{L}'\sum_{s''}\frac{1}{\sqrt{LN}}\delta\boldsymbol{h}_{\ell,s''}\boldsymbol{o}_{\ell,s''}^{\top}\right)^{\top}\delta\bar{\boldsymbol{h}}_{\ell,s'}$$

$$= -\frac{1}{L}\eta_o'\mathcal{L}'\sum_{s'}\bar{a}_{\ell,s's}\sum_{s''}\frac{\langle\delta\boldsymbol{h}_{\ell,s''},\delta\bar{\boldsymbol{h}}_{\ell,s'}\rangle}{N}\left(\frac{1}{\sqrt{N}}\boldsymbol{W}_{\ell,\boldsymbol{v}}^{\top}\boldsymbol{z}_{\ell,s''}\right)$$

$$\sim -\frac{1}{L}\eta_o'\mathcal{L}'\sum_{s'}\bar{a}_{\ell,s's}\sum_{s''}\frac{\langle\delta\boldsymbol{h}_{\ell,s''},\delta\bar{\boldsymbol{h}}_{\ell,s'}\rangle}{N}\sum_{s'''}a_{\ell,s''s'''}\left(\frac{1}{N}\boldsymbol{W}_{\ell,\boldsymbol{v}}^{\top}\boldsymbol{W}_{\ell,v}\boldsymbol{h}_{\ell-1,s'''}\right)$$

Hence the characterization of the $\frac{1}{\sqrt{LN}}\bar{\boldsymbol{W}}_{\ell,\boldsymbol{v}}^{\top}\delta\bar{\boldsymbol{v}}_{\ell,s}$ is given by

$$\frac{1}{\sqrt{L}}\underbrace{Z^{\boldsymbol{W}_{\ell,v}^{\top}\delta\bar{\boldsymbol{v}}_{\ell,s}}}_{\text{Gaussian}} - \frac{\eta_v'}{L}\mathring{\mathcal{L}}'\underbrace{\sum_{s'}\mathbb{E}[Z^{\delta\boldsymbol{v}_{\ell,s'}}Z^{\delta\bar{\boldsymbol{v}}_{\ell,s}}]Z^{\boldsymbol{h}_{\ell-1,s'}}}_{\text{Learning}}$$

$$- \frac{\eta_o'}{L}\mathring{\mathcal{L}}'\underbrace{\sum_{s',s'',s'''}\bar{a}_{\ell,s's}a_{\ell,s''s'''}\mathbb{E}[Z^{\delta\boldsymbol{h}_{\ell,s''}}Z^{\delta\bar{\boldsymbol{h}}_{\ell,s'}}]Z^{\boldsymbol{h}_{\ell-1,s'''}}}_{\text{Interaction}}$$

Similarly, observe that

$$\frac{1}{\sqrt{LN}}\left(\boldsymbol{W}_{\ell,q}+\Delta\boldsymbol{W}_{\ell,q}\right)^{\top}\delta\bar{\boldsymbol{q}}_{\ell,s} = \frac{1}{\sqrt{LN}}\boldsymbol{W}_{\ell,q}^{\top}\delta\bar{\boldsymbol{q}}_{\ell,s} + \frac{1}{\sqrt{LN}}\left(-\eta_q'\mathcal{L}'\sum_{s'}\frac{1}{\sqrt{LN}}\delta\boldsymbol{q}_{\ell,s'}\boldsymbol{h}_{\ell-1,s'}^{\top}\right)^{\top}\delta\bar{\boldsymbol{q}}_{\ell,s}$$

$$= \frac{1}{\sqrt{LN}}\boldsymbol{W}_{\ell,q}^{\top}\delta\bar{\boldsymbol{q}}_{\ell,s} - \frac{1}{L}\eta_q'\mathcal{L}'\sum_{s'}\frac{\langle\delta\boldsymbol{q}_{\ell,s'},\delta\bar{\boldsymbol{q}}_{\ell,s}\rangle}{N}\boldsymbol{h}_{\ell-1,s'}$$

There is interaction term that occurred from

$$\frac{1}{\sqrt{LN}}\boldsymbol{W}_{\ell,q}^{\top}\delta\bar{\boldsymbol{q}}_{\ell,s}$$

$$= \frac{1}{\sqrt{LN}}\boldsymbol{W}_{\ell,q}^{\top}\left[\sum_{s'}\frac{\langle\delta\bar{\boldsymbol{o}}_{\ell,s},\bar{\boldsymbol{v}}_{\ell,s'}-\bar{\boldsymbol{o}}_{\ell,s}\rangle}{N}a_{\ell,ss'}\bar{\boldsymbol{k}}_{\ell,s'}\right]$$

$$\sim \frac{1}{\sqrt{L}}\sum_{s'}\frac{\langle\delta\bar{\boldsymbol{o}}_{\ell,s},\bar{\boldsymbol{v}}_{\ell,s'}-\bar{\boldsymbol{o}}_{\ell,s}\rangle}{N}a_{\ell,ss'}\left[\frac{1}{\sqrt{N}}\boldsymbol{W}_{\ell,q}^{\top}\left(-\frac{\eta_k'}{\sqrt{L}}\mathcal{L}'\sum_{s''}\frac{\langle\boldsymbol{h}_{\ell-1,s''},\bar{\boldsymbol{h}}_{\ell-1,s'}\rangle}{N}\delta\boldsymbol{k}_{\ell,s''}\right)\right]$$

$$\sim -\frac{1}{L}\eta_k'\mathcal{L}'\sum_{s'}\frac{\langle\delta\bar{\boldsymbol{o}}_{\ell,s},\bar{\boldsymbol{v}}_{\ell,s'}-\bar{\boldsymbol{o}}_{\ell,s}\rangle}{N}\bar{a}_{\ell,ss'}\sum_{s''}\frac{\langle\boldsymbol{h}_{\ell-1,s''},\bar{\boldsymbol{h}}_{\ell-1,s'}\rangle}{N}\left[\frac{1}{\sqrt{N}}\boldsymbol{W}_{\ell,q}^{\top}\delta\boldsymbol{k}_{\ell,s''}\right]$$

$$\sim -\frac{1}{L}\eta_k'\mathcal{L}'\sum_{s'}\frac{\langle\delta\bar{\boldsymbol{o}}_{\ell,s},\bar{\boldsymbol{v}}_{\ell,s'}-\bar{\boldsymbol{o}}_{\ell,s}\rangle}{N}\bar{a}_{\ell,ss'}\sum_{s''}\frac{\langle\boldsymbol{h}_{\ell-1,s''},\bar{\boldsymbol{h}}_{\ell-1,s'}\rangle}{N}$$

$$\left[\sum_{s'''}\frac{\langle\delta\boldsymbol{o}_{\ell,s''},\boldsymbol{v}_{\ell,s'''}-\boldsymbol{o}_{\ell,s'''}\rangle}{N}a_{\ell,s'''s''}\frac{1}{N}\boldsymbol{W}_{\ell,q}^{\top}\boldsymbol{W}_{\ell,q}\boldsymbol{h}_{\ell-1,s'''}\right]$$

Hence, we obtain the mean-field characterization of $\frac{1}{\sqrt{LN}}\bar{W}_{\ell,q}^\top \delta \bar{q}_{\ell,s}$ as

$$\frac{1}{\sqrt{L}}\underbrace{Z^{W_{\ell,q}^\top \delta \bar{q}_{\ell,s}}}_{\text{Gaussian}} - \frac{\eta_q'}{L}\mathring{\mathcal{L}}'\underbrace{\sum_{s'}\mathbb{E}[Z^{\delta q_{\ell,s'}}Z^{\delta \bar{q}_{\ell,s}}]Z^{h_{\ell-1,s'}}}_{\text{Learning}}$$

$$-\frac{\eta_k'}{L}\mathring{\mathcal{L}}'\underbrace{\sum_{s',s'',s'''}\mathbb{E}[Z^{\delta \bar{o}_{\ell,s}}(Z^{\bar{v}_{\ell,s'}} - Z^{\bar{o}_{\ell,s}})]\bar{a}_{\ell,ss'}\mathbb{E}[Z^{h_{\ell-1,s''}}Z^{h_{\ell-1,s'}}]\mathbb{E}[Z^{\delta o_{\ell,s''}}(Z^{v_{\ell,s''}} - Z^{o_{\ell,s'''}})]a_{\ell,s'''s''}Z^{h_{\ell-1,s'''}}}_{\text{interaction}}$$

Therefore, we obtain the characterization of residual stream:

$$\underbrace{Z^{\delta \bar{h}_{\ell-1,s}} - Z^{\delta \bar{h}_{\ell,s}}}_{\text{Residual stream}} = \frac{1}{\sqrt{L}}\underbrace{Z^{\delta \widetilde{h}_{\ell-1,s}}}_{\text{Gaussian}}$$

$$- \mathring{\mathcal{L}}'\sum_{s'}\left(\frac{\eta_q'}{L}\mathbb{E}[Z^{\delta q_{\ell,s'}}Z^{\delta \bar{q}_{\ell,s}}] + \frac{\eta_k'}{L}\mathbb{E}[Z^{\delta k_{\ell,s'}}Z^{\delta \bar{k}_{\ell,s}}] + \frac{\eta_v'}{L}\mathbb{E}[Z^{\delta v_{\ell,s'}}Z^{\delta \bar{v}_{\ell,s}}]\right)Z^{h_{\ell-1,s'}}$$

$$- \sum_{s',s'',s'''}(\frac{\eta_k'}{L}\cdots + \frac{\eta_q'}{L}\cdots + \frac{\eta_o'}{L}\cdots)Z^{h_{\ell-1,s'''}}$$

where $Z^{\delta \widetilde{h}_{\ell-1,s}} \sim \mathcal{N}(0, \mathbb{E}[(Z^{\delta q_{\ell,s}})^2 + (Z^{\delta k_{\ell,s}})^2 + (Z^{\delta v_{\ell,s}})^2])$.

### D.5. Convergence rate of first forward and first backward

We will highlight the main difference compared with the analysis in the two-layer case and in (Yao et al., 2025, Appendix C and G).

For the first forward, using discrete-time Gronwall's inequality, one can obtain the moment bound

$$\mathbb{E}|h_{\ell,s,i}|^2 \le C$$

uniform in $\ell \in [L]$, token $s \in [S]$, coordinate $i \in [N]$, and $N, L \ge 1$. Using a weak law of large numbers type analysis, one has

$$\mathbb{E}[(b_{\ell,ss'})^2 \mid h_{\ell-1,s}, h_{\ell-1,s'}] \le \frac{\|h_{\ell-1,s}\|}{\sqrt{N}}\frac{\|h_{\ell-1,s'}\|}{\sqrt{N}}\frac{C}{N}.$$

Combining these two estimates, we have

$$\mathbb{E}(b_{\ell,ss'})^2 \le \frac{C}{N}.$$

Noting that $a_{\ell,ss'} = e^{b_{ss'}}/\sum_{s''}e^{b_{ss''}}$ is Lipschitz in each $b_{ss''}$, we have

$$\mathbb{E}(a_{\ell,ss'} - \frac{1}{S})^2 \le C\mathbb{E}\sum_{s''}(b_{ss''} - 0)^2 \le \frac{C}{N}.$$

With this rate obtained, all the rest analysis is analogous to the two-layer case and one can get

$$\max_{\ell=0,1,\ldots,L}\max_{i=1,2,\ldots,N}\mathbb{E}(h_{\ell,s,i} - h_{\ell/L,s,i})^2 \le \frac{C}{N} + \frac{C}{L},$$

where, omitting the coordinate index $i$, $h_{t,s}$ is the solution of the stochastic differential equation

$$dh_{t,s} = \sqrt{\mathbb{E}\left[\frac{1}{S}\sum_{s'=1}^S h_{t,s'}\right]^2}\, dw_t$$

driven by a common standard Brownian motion $\{w_t\}$. Here note that $h_{t,s}$ may not have the same distribution as $h_{t,s'}$ if they have different initial conditions. To be more precise, the full limiting dynamics is

$$q_{t,s} = w_{t,s}^q,$$
$$k_{t,s} = w_{t,s}^k,$$
$$v_{t,s} = w_{t,s}^v,$$
$$b_{t,ss'} = 0,$$
$$a_{t,ss'} = \frac{1}{S},$$
$$o_{t,s} = \frac{1}{S} \sum_{s'=1}^{S} v_{t,s'},$$
$$dh_{t,s} = dw_t^o,$$

where $\{w_{t,s}^q\}_{s=1}^S$, $\{w_{t,s}^k\}_{s=1}^S$, $\{w_{t,s}^v\}_{s=1}^S$ are independent Gaussian random variables with mean $0$ and covariances

$$\mathbb{E}[w_{t,s}^q, w_{t,s'}^q] = \mathbb{E}[w_{t,s}^k, w_{t,s'}^k] = \mathbb{E}[w_{t,s}^v, w_{t,s'}^v] = \mathbb{E}[h_{t,s} h_{t,s'}],$$

and $\{w_t^o\}$ is an independent Brownian motion with mean $0$ and quadratic variation

$$d\langle w^o \rangle_t = \mathbb{E}\left[o_{t,s}\right]^2 dt = \mathbb{E}\left[\frac{1}{S} \sum_{s'=1}^{S} h_{t,s'}\right]^2 dt.$$

The rate of convergence of each variable is still

$$\max_{\ell=0,1,...,L} \max_{i=1,2,...,N} \mathbb{E}(\boldsymbol{\theta}_{\ell,s,i} - \theta_{\ell/L,s,i})^2 \leq \frac{C}{N} + \frac{C}{L}, \quad \theta \in \{q,k,v,o,h\}$$

with

$$\max_{\ell=0,1,...,L} \max_{i=1,2,...,N} \mathbb{E}(a_{\ell,ss'} - a_{\ell/L,ss'})^2 \leq \frac{C}{N}, \quad \max_{\ell=0,1,...,L} \max_{i=1,2,...,N} \mathbb{E}(b_{\ell,ss'} - b_{\ell/L,ss'})^2 \leq \frac{C}{N}.$$

Similarly, for the first backward, assuming that independent Gaussian matrices are used instead of $\boldsymbol{W}^T$, one can get

$$\max_{\ell=0,1,...,L} \max_{i=1,2,...,N} \mathbb{E}(\delta \boldsymbol{h}_{\ell,s,i} - \delta h_{\ell/L,s,i})^2 \leq \frac{C}{N} + \frac{C}{L},$$

where, omitting the coordinate index $i$, $\delta h_{t,s}$ is the solution of the backward stochastic differential equation

$$d(\delta h_{t,s}) = d\tilde{w}_{t,s},$$

and $\{\tilde{w}_{t,s}\}_{s=1}^S$ are Brownian motions with mean $0$ and cross-variations

$$d\langle \tilde{w}_s, \tilde{w}_{s'} \rangle_t = \mathbb{E}\left[\left(\frac{1}{S} \sum_{s''=1}^{S} \mathbb{E}[\delta o_{t,s}(v_{t,s''} - o_{t,s})]h_{t,s''}\right)\left(\frac{1}{S} \sum_{s''=1}^{S} \mathbb{E}[\delta o_{t,s'}(v_{t,s''} - o_{t,s'})]h_{t,s''}\right)\right] dt$$
$$+ \mathbb{E}\left[\left(\frac{1}{S} \sum_{s''=1}^{S} \mathbb{E}[\delta o_{t,s''}(v_{t,s} - o_{t,s''})]h_{t,s''}\right)\left(\frac{1}{S} \sum_{s''=1}^{S} \mathbb{E}[\delta o_{t,s''}(v_{t,s'} - o_{t,s''})]h_{t,s''}\right)\right] dt$$
$$+ \mathbb{E}\left[\frac{1}{S} \sum_{s''=1}^{S} \delta h_{t,s''}\right]^2 dt.$$

To be more precise, the full limiting dynamics is

$$\delta o_{t,s} = \tilde{w}^o_{t,s},$$

$$\delta v_{t,s} = \frac{1}{S} \sum_{s'=1}^{S} \delta o_{t,s'},$$

$$\delta q_{t,s} = \frac{1}{S} \sum_{s'=1}^{S} \mathbb{E}[\delta o_{t,s}(v_{t,s'} - o_{t,s})]k_{t,s'},$$

$$\delta k_{t,s} = \frac{1}{S} \sum_{s'=1}^{S} \mathbb{E}[\delta o_{t,s'}(v_{t,s} - o_{t,s'})]q_{t,s'},$$

$$d(\delta h_{t,s}) = d\tilde{w}^q_{t,s} + d\tilde{w}^k_{t,s} + d\tilde{w}^v_t,$$

where $\{\tilde{w}^o_{t,s}\}_{s=1}^S$ are independent Gaussian random variables with mean 0 and covariance

$$\mathbb{E}[\tilde{w}^o_{t,s}, \tilde{w}^o_{t,s'}] = \mathbb{E}[\delta h_{t,s} \delta h_{t,s'}],$$

and $\{\tilde{w}^q_{t,s}\}_{s=1}^S$, $\{\tilde{w}^k_{t,s}\}_{s=1}^S$, and $\{\tilde{w}^v_t\}$ are independent Brownian motions with mean 0 and cross-variations

$$d\langle \tilde{w}^q_s, \tilde{w}^q_{s'} \rangle_t = \mathbb{E}\left[\delta q_{t,s} \delta q_{t,s'}\right] dt,$$

$$d\langle \tilde{w}^k_s, \tilde{w}^k_{s'} \rangle_t = \mathbb{E}\left[\delta k_{t,s} \delta k_{t,s'}\right] dt,$$

$$d\langle \tilde{w}^v \rangle_t = \mathbb{E}\left[\delta v_{t,s}\right]^2 dt = \mathbb{E}\left[\frac{1}{S} \sum_{s'=1}^{S} \delta h_{t,s'}\right]^2 dt.$$

The rate of convergence of each variable is still

$$\max_{\ell=0,1,\ldots,L} \max_{i=1,2,\ldots,N} \mathbb{E}(\boldsymbol{\theta}_{\ell,s,i} - \theta_{\ell/L,s,i})^2 \leq \frac{C}{N} + \frac{C}{L}, \quad \theta \in \{\delta q, \delta k, \delta v, \delta o, \delta h\}.$$

### D.6. Convergence of second forward and second backward

The analysis of second forward and second backward is very similar to that in the two-layer case. So we directly write down the limiting dynamics as $L \to \infty$ and provide a sketch of the proof.

Letting $L \to \infty$, the limiting dynamics of second forward is

$$q^{(1)}_{t,s} = w^{q,(1)}_{t,s}, \tag{36}$$

$$k^{(1)}_{t,s} = k^{q,(1)}_{t,s}, \tag{37}$$

$$v^{(1)}_{t,s} = v^{q,(1)}_{t,s}, \tag{38}$$

$$b^{(1)}_{t,ss'} = 0, \tag{39}$$

$$a^{(1)}_{t,ss'} = \frac{1}{S}, \tag{40}$$

$$o^{(1)}_{t,s} = \frac{1}{S} \sum_{s'=1}^{S} v^{(1)}_{t,s'}, \tag{41}$$

$$dh^{(1)}_{t,s} = dw^{o,(1)}_t - \eta'_o \mathring{\mathcal{L}}' \sum_{s'} \mathbb{E}[o_{t,s'} o^{(1)}_{t,s}] \delta h_{t,s'} - \eta'_v \mathring{\mathcal{L}}' \mathbb{E}\left[\frac{1}{S} \sum_{s'=1}^{S} h_{t,s'}\right]^2 \sum_{s'=1}^{S} \delta h_{t,s'}, \tag{42}$$

where $\{(w^{o,(0)}_t, w^{o,(1)}_t)\}$ is an independent Brownian motion with mean 0 and cross-variation

$$d\langle w^{o,(j)}, w^{o,(j')} \rangle_t = \mathbb{E}[o^{(j)}_{t,s} o^{(j')}_{t,s}] dt,$$

and $\{(w_{t,s}^{q,(0)}, w_{t,s}^{q,(1)}), (w_{t,s}^{k,(0)}, w_{t,s}^{k,(1)}), (w_{t,s}^{v,(0)}, w_{t,s}^{v,(1)})\}$ are independent Gaussian random vectors with mean 0 and covariances

$$\mathrm{Cov}(w_{t,s}^{q,(j)}, w_{t,s'}^{q,(j')}) = \mathrm{Cov}(w_{t,s}^{k,(j)}, w_{t,s'}^{k,(j')}) = \mathrm{Cov}(w_{t,s}^{v,(j)}, w_{t,s'}^{v,(j')}) = \mathbb{E}[h_{t,s}^{(j)} h_{t,s'}^{(j')}].$$

The limiting dynamics of second backward is

$$\delta o_{t,s}^{(1)} = \tilde{w}_{t,s}^{o,(1)}, \tag{43}$$

$$\delta v_{t,s}^{(1)} = \frac{1}{S} \sum_{s'=1}^{S} \delta o_{t,s'}^{(1)}, \tag{44}$$

$$\delta q_{t,s}^{(1)} = \frac{1}{S} \sum_{s'=1}^{S} \mathbb{E}[\delta o_{t,s}^{(1)}(v_{t,s'}^{(1)} - o_{t,s}^{(1)})] k_{t,s'}^{(1)}, \tag{45}$$

$$\delta k_{t,s}^{(1)} = \frac{1}{S} \sum_{s'=1}^{S} \mathbb{E}[\delta o_{t,s'}^{(1)}(v_{t,s}^{(1)} - o_{t,s'}^{(1)})] q_{t,s'}^{(1)}, \tag{46}$$

$$d(\delta h_{t,s}^{(1)}) = d\tilde{w}_{t,s}^{q,(1)} + d\tilde{w}_{t,s}^{k,(1)} + d\tilde{w}_{t}^{v,(1)} \tag{47}$$

$$- \mathring{\mathcal{L}}' \sum_{s'=1}^{S} \left( \eta_q' \mathbb{E}[\delta q_{t,s'} \delta q_{t,s}^{(1)}] + \eta_k' \mathbb{E}[\delta k_{t,s'} \delta k_{t,s}^{(1)}] + \eta_v' \mathbb{E}[\delta v_{t,s'} \delta v_{t,s}^{(1)}] \right) h_{t,s'} \, dt \tag{48}$$

$$- \eta_o' \mathring{\mathcal{L}}' \mathbb{E} \left[ \left( \frac{1}{S} \sum_{s'=1}^{S} \delta h_{t,s'} \right) \left( \frac{1}{S} \sum_{s'=1}^{S} \delta h_{t,s'}^{(1)} \right) \right] \sum_{s'=1}^{S} h_{t,s'} \, dt \tag{49}$$

$$- \eta_k' \mathring{\mathcal{L}}' \frac{1}{S^2} \sum_{s',s'',s'''=1}^{S} \mathbb{E}[\delta o_{t,s}^{(1)}(v_{t,s'}^{(1)} - o_{t,s}^{(1)})] \mathbb{E}[h_{t,s'} h_{t,s''}] \mathbb{E}[\delta o_{t,s''}(v_{t,s''} - o_{t,s'''})] h_{t,s'''} \, dt \tag{50}$$

$$- \eta_q' \mathring{\mathcal{L}}' \frac{1}{S^2} \sum_{s',s'',s'''=1}^{S} \mathbb{E}[\delta o_{t,s'}^{(1)}(v_{t,s}^{(1)} - o_{t,s'}^{(1)})] \mathbb{E}[h_{t,s'} h_{t,s''}] \mathbb{E}[\delta o_{t,s''}(v_{t,s''} - o_{t,s'''})] h_{t,s'''} \, dt, \tag{51}$$

where $\{(\tilde{w}_{t,s}^{o,(0)}, \tilde{w}_{t,s}^{o,(1)})\}_{s=1}^{S}$ are independent Gaussian random variables with mean 0 and covariance

$$\mathbb{E}[\tilde{w}_{t,s}^{o,(j)}, \tilde{w}_{t,s'}^{o,(j')}] = \mathbb{E}[\delta h_{t,s}^{(j)} \delta h_{t,s'}^{(j')}],$$

and $\{(\tilde{w}_{t,s}^{q,(0)}, \tilde{w}_{t,s}^{q,(1)})\}_{s=1}^{S}$, $\{(\tilde{w}_{t,s}^{k,(0)}, \tilde{w}_{t,s}^{k,(1)})\}_{s=1}^{S}$, and $\{(\tilde{w}_{t}^{v,(0)}, \tilde{w}_{t}^{v,(1)})\}$ are independent Brownian motions with mean 0 and cross-variations

$$d\langle \tilde{w}_{s}^{q,(j)}, \tilde{w}_{s'}^{q,(j')} \rangle_t = \mathbb{E} \left[ \delta q_{t,s}^{(j)} \delta q_{t,s'}^{(j')} \right] dt,$$

$$d\langle \tilde{w}_{s}^{k,(j)}, \tilde{w}_{s'}^{k,(j')} \rangle_t = \mathbb{E} \left[ \delta k_{t,s}^{(j)} \delta k_{t,s'}^{(j')} \right] dt,$$

$$d\langle \tilde{w}^{v,(j)}, \tilde{w}^{v,(j')} \rangle_t = \mathbb{E} \left[ \delta v_{t,s}^{(j)} \delta v_{t,s}^{(j')} \right] dt.$$

Again, we note that here the evolution of $\delta h_{t,s}^{(0)}$ and $\delta h_{t,s}^{(1)}$ is interpreted backward from $t = 1$ to $t = 0$.

The following theorem gives the convergence rate as $L \to \infty$.

**Theorem D.1.** *Assume that*

- *$\mathring{\mathcal{L}}'$ is Lipschitz continuous;*

- *there exists a solution to the system Eq. (36)–Eq. (51) such that the $2 \times 2$ matrices $(\mathbb{E}[o_{t,s}^{(j)} o_{t,s}^{(j')}])$ and $(\mathbb{E}\left[\delta v_{t,s}^{(j)} \delta v_{t,s}^{(j')}\right])$ and $2S \times 2S$ matrices $(\mathbb{E}\left[\delta q_{t,s}^{(j)} \delta q_{t,s'}^{(j')}\right])$ and $(\mathbb{E}\left[\delta k_{t,s}^{(j)} \delta k_{t,s'}^{(j')}\right])$ are uniformly strictly positive definite, that is, the smallest eigenvalue is always great than some $\varepsilon > 0$.*

*Then for all $L \geq 1$, for each of the variable $\theta = x, h, \delta x, \delta h$, we have*

$$\sup_{\ell=0,1,\ldots,L} \mathbb{E}\left[|\boldsymbol{\theta}_{\ell,s}^{(0),L} - \theta_{\ell/L,s}^{(0)}|^2 + |\theta_{\ell,s}^{(1),L} - \theta_{\ell/L,s}^{(1)}|^2\right] \leq C/L, \quad \theta \in \{q, k, v, o, h, \delta q, \delta k, \delta v, \delta o, \delta h\}$$

*where $\theta_{\ell,s}^{(j),L}$ denotes the dynamics after taking $n \to \infty$.*

For the proof of this theorem, similar to the analysis of the two-layer setup and that of (Yao et al., 2025, Proposition G.6), a key observation is that the higher order interaction terms are vanishing as $L \to \infty$.

## E. Decoupled Training

Now, consider we decouple the backward pass from the forward propagation:

$$\delta\boldsymbol{h}_{L,s} = \delta_{s,1}\boldsymbol{v}$$

$$\delta\boldsymbol{o}_{\ell,s} = \frac{1}{\sqrt{N}}\widetilde{\boldsymbol{W}}_{\ell,o}^{\top}\delta\boldsymbol{h}_{\ell,s}$$

$$\delta\boldsymbol{v}_{\ell,s} = \sum_{s'} a_{s',s}\delta\boldsymbol{o}_{\ell,s'}$$

$$\delta\boldsymbol{q}_{\ell,s} = \sum_{s'} \frac{\langle\delta\boldsymbol{o}_{\ell,s}, \boldsymbol{v}_{\ell,s'} - \boldsymbol{o}_{\ell,s}\rangle}{N} a_{\ell,ss'}\boldsymbol{k}_{\ell,s'},$$

$$\delta\boldsymbol{k}_{\ell,s} = \sum_{s'} \frac{\langle\delta\boldsymbol{o}_{\ell,s'}, \boldsymbol{v}_{\ell,s} - \boldsymbol{o}_{\ell,s'}\rangle}{N} a_{\ell,s's}\boldsymbol{q}_{\ell,s'}$$

$$\delta\boldsymbol{h}_{\ell-1,s} = \delta\boldsymbol{h}_{\ell,s} + \frac{1}{\sqrt{LN}}\widetilde{\boldsymbol{W}}_{\ell,q}^{\top}\delta\boldsymbol{q}_{\ell,s} + \frac{1}{\sqrt{LN}}\widetilde{\boldsymbol{W}}_{\ell,k}^{\top}\delta\boldsymbol{k}_{\ell,s} + \frac{1}{\sqrt{LN}}\widetilde{\boldsymbol{W}}_{\ell,v}^{\top}\delta\boldsymbol{v}_{\ell,s}$$

Then we can separately update the backward weights:

$$\widetilde{\boldsymbol{W}}_{\ell,q}^{+} = \widetilde{\boldsymbol{W}}_{\ell,q} - \widetilde{\eta}_q'\mathcal{L}' \sum_s \frac{1}{\sqrt{LN}}\delta\boldsymbol{q}_{\ell,s}\boldsymbol{h}_{\ell-1,s}^{\top}$$

$$\widetilde{\boldsymbol{W}}_{\ell,k}^{+} = \widetilde{\boldsymbol{W}}_{\ell,k} - \widetilde{\eta}_k'\mathcal{L}' \sum_s \frac{1}{\sqrt{LN}}\delta\boldsymbol{k}_{\ell,s}\boldsymbol{h}_{\ell-1,s}^{\top}$$

$$\widetilde{\boldsymbol{W}}_{\ell,v}^{+} = \widetilde{\boldsymbol{W}}_{\ell,v} - \widetilde{\eta}_v'\mathcal{L}' \sum_s \frac{1}{\sqrt{LN}}\delta\boldsymbol{v}_{\ell,s}\boldsymbol{h}_{\ell-1,s}^{\top}$$

$$\widetilde{\boldsymbol{W}}_{\ell,o}^{+} = \widetilde{\boldsymbol{W}}_{\ell,o} - \widetilde{\eta}_o'\mathcal{L}' \sum_s \frac{1}{\sqrt{LN}}\delta\boldsymbol{h}_{\ell,s}\boldsymbol{z}_{\ell,s}^{\top}$$

### E.1. Asymmetric learning rates

In this subsection, we consider the case where the backward weights use the same weights as forward weights but *asymmetric* learning rates, *i.e.*, $\eta_x \neq \widetilde{\eta}_x$. As a result, we still have interaction terms, and the dynamics after one-step update become:

$$Z^{\boldsymbol{q}_{\ell,s}(1)} = Z^{\widetilde{\boldsymbol{q}}_{\ell,s}(1)} - \frac{\eta_q'}{\sqrt{L}}\mathcal{G}(Z^{\delta\boldsymbol{q}_{\ell}(0)}) - \frac{\eta_o'}{L^{\frac{3}{2}}}\mathcal{I}(Z^{\delta\boldsymbol{q}_{\ell}(0)})$$

$$Z^{\boldsymbol{k}_{\ell,s}(1)} = Z^{\widetilde{\boldsymbol{k}}_{\ell,s}(1)} - \frac{\eta_k'}{\sqrt{L}}\mathcal{G}(Z^{\delta\boldsymbol{k}_{\ell}(0)}) - \frac{\eta_o'}{L^{\frac{3}{2}}}\mathcal{I}(Z^{\delta\boldsymbol{k}_{\ell}(0)})$$

$$Z^{\boldsymbol{v}_{\ell,s}(1)} = Z^{\widetilde{\boldsymbol{v}}_{\ell,s}(1)} - \frac{\eta_v'}{\sqrt{L}}\mathcal{G}(Z^{\delta\boldsymbol{v}_{\ell}(0)}) - \frac{\eta_o'}{L^{\frac{3}{2}}}\mathcal{I}(Z^{\delta\boldsymbol{v}_{\ell}})$$

$$Z^{\boldsymbol{h}_{\ell,s}(1)} - Z^{\boldsymbol{h}_{\ell-1,s}(1)} = \frac{1}{\sqrt{L}}Z^{\widetilde{\boldsymbol{h}}_{\ell,s}^{(1)}} - \frac{\eta_o'}{L}\mathcal{G}(Z^{\delta\boldsymbol{h}_{\ell}}) - \frac{\eta_v'}{L}\mathcal{I}(Z^{\delta\boldsymbol{h}_{\ell}})$$

and second backward pass

$$Z^{\delta\bar{\boldsymbol{o}}_{\ell,s}} = Z^{\delta\tilde{\boldsymbol{o}}_{\ell,s}} - \frac{\widetilde{\eta}'_o}{\sqrt{L}}\mathcal{G}(Z^{\boldsymbol{o}_\ell}) - \frac{\widetilde{\eta}'_{qkv}}{L^{\frac{3}{2}}}\mathcal{I}(Z^{\boldsymbol{o}_\ell})$$

$$Z^{\delta\bar{\boldsymbol{h}}_{\ell-1,s}} - Z^{\delta\bar{\boldsymbol{h}}_{\ell,s}} = \frac{1}{\sqrt{L}}Z^{\delta\tilde{\boldsymbol{h}}_{\ell-1,s}} - \frac{\widetilde{\eta}'_{qkv}}{L}\mathcal{G}(Z^{\boldsymbol{h}^{(0)}_{\ell-1}}) - \frac{\widetilde{\eta}'_{kqo}}{L}\mathcal{I}(Z^{\boldsymbol{h}_{\ell-1}})$$

where we denote the learning term and interaction terms as follows to simplify the expression:

$$\mathcal{G}(Z^{\delta\boldsymbol{q}^{(0)}_\ell}) := \mathring{\mathcal{L}}' \sum_{s'} \mathbb{E}[Z^{\boldsymbol{h}_{\ell-1,s'}} Z^{\bar{\boldsymbol{h}}_{\ell-1,s}}] Z^{\delta\boldsymbol{q}^{(0)}_{\ell,s'}}$$

$$\mathcal{I}(Z^{\delta\boldsymbol{q}_\ell(0)}) := \mathring{\mathcal{L}}' \sum_{s'} \mathbb{E}[Z^{\boldsymbol{o}_{\ell-1,s'}} Z^{\bar{\boldsymbol{o}}_{\ell-1,s}}] Z^{\delta\boldsymbol{q}_{\ell,s'}}$$

From these dynamics, we can see that, even with an asymmetric learning rate $\eta'_v$, increasing $\eta'_v$ still results in the interaction term in the forward residual stream, $\partial_t \boldsymbol{h}$, being explored. However, because of the asymmetric learning rate, we can increase the learning rates $\eta'_q$ and $\eta'_k$ to $\sqrt{L}$ to counteract the learning collapse in $\boldsymbol{q}_\ell$ and $\boldsymbol{k}_\ell$ without triggering any exploring quantities.

### E.2. Decoupled backward weights

The exploration of interaction terms prevents us from using a larger learning rate, even when we choose asymmetric learning rates. Hence, a natural way is to use decoupled backward weights to vanish those sensitive interaction terms. We can see that the interaction terms stemming from $\boldsymbol{q}_\ell$ and $\boldsymbol{k}_\ell$ are higher orders and vanish in the large-depth limit. Hence, it is not necessary to use decoupled weights. On the other hand, by decoupling $\boldsymbol{W}_{\ell,v}$ and $\boldsymbol{W}_{\ell,o}$, the interaction terms steams from $\boldsymbol{v}_\ell$ and residual stream $\partial_t \boldsymbol{h}$ disappears. The resulting dynamics become

$$Z^{\boldsymbol{q}_{\ell,s}(1)} = Z^{\tilde{\boldsymbol{q}}_{\ell,s}(1)} - \frac{\eta'_q}{\sqrt{L}}\mathcal{G}(Z^{\delta\boldsymbol{q}_\ell(0)}) - \frac{\eta'_o}{L^{\frac{3}{2}}}\mathcal{I}(Z^{\delta\boldsymbol{q}_\ell(0)})$$

$$Z^{\boldsymbol{k}_{\ell,s}(1)} = Z^{\tilde{\boldsymbol{k}}_{\ell,s}(1)} - \frac{\eta'_k}{\sqrt{L}}\mathcal{G}(Z^{\delta\boldsymbol{k}_\ell(0)}) - \frac{\eta'_o}{L^{\frac{3}{2}}}\mathcal{I}(Z^{\delta\boldsymbol{k}_\ell(0)})$$

$$Z^{\boldsymbol{v}_{\ell,s}(1)} = Z^{\tilde{\boldsymbol{v}}_{\ell,s}(1)} - \frac{\eta'_v}{\sqrt{L}}\mathcal{G}(Z^{\delta\boldsymbol{v}_\ell(0)})$$

$$Z^{\boldsymbol{h}_{\ell,s}(1)} - Z^{\boldsymbol{h}_{\ell-1,s}(1)} = \frac{1}{\sqrt{L}}Z^{\tilde{\boldsymbol{h}}^{(1)}_{\ell,s}} - \frac{\eta'_o}{L}\mathcal{G}(Z^{\delta\boldsymbol{h}_\ell})$$

and

$$Z^{\delta\bar{\boldsymbol{o}}_{\ell,s}} = Z^{\delta\tilde{\boldsymbol{o}}_{\ell,s}} - \frac{\widetilde{\eta}'_o}{\sqrt{L}}\mathcal{G}(Z^{\boldsymbol{o}_\ell})$$

$$Z^{\delta\bar{\boldsymbol{h}}_{\ell-1,s}} - Z^{\delta\bar{\boldsymbol{h}}_{\ell,s}} = \frac{1}{\sqrt{L}}Z^{\delta\tilde{\boldsymbol{h}}_{\ell-1,s}} - \frac{\widetilde{\eta}'_{qkv}}{L}\mathcal{G}(Z^{\boldsymbol{h}^{(0)}_{\ell-1}})$$

Therefore, we obtain the schedule of learning rates as follows

$$\eta'_q = \eta'_k = \eta'_v = \widetilde{\eta}'_o = \eta_c\sqrt{L}, \qquad \eta'_o = \widetilde{\eta}'_q = \widetilde{\eta}'_k = \widetilde{\eta}_v = \eta_c.$$

# F. Interaction between FFN and Attention

Recall that the output from the attention is given by

$$h_\ell^{\text{attn}} = h_{\ell-1} + \frac{1}{\sqrt{LN}} W_{\ell,o} z_\ell$$

$$h_\ell = h_\ell^{\text{attn}} + \frac{1}{\sqrt{LN}} W_\ell \phi \left( \frac{1}{\sqrt{N}} U_\ell h_\ell^{\text{attn}} \right)$$

Observe that

$$x_\ell = \frac{1}{\sqrt{N}} U_\ell h^{\text{attn}}$$

$$= \frac{1}{\sqrt{N}} U_\ell \left( h_{\ell-1} + \frac{1}{\sqrt{LN}} W_{\ell,o} z_\ell \right)$$

In the large-width limit, we have their mean-field characterization as follows

$$Z^{x_\ell} = Z^{U h^{\text{attn}}} \sim \mathcal{N} \left( 0, \mathbb{E} \left[ Z^{h_{\ell-1}} + \frac{1}{\sqrt{L}} Z^{W_{\ell,o} z_\ell} \right]^2 \right)$$

As depth becomes larger, the quantity $\mathbb{E} \left[ Z^{h_{\ell-1}} + \frac{1}{\sqrt{L}} Z^{W_{\ell,o} z_\ell} \right]^2$ converges to $\mathbb{E} \left[ Z^{h_{\ell-1}} \right]^2$. This implies that

$$Z^{x_\ell} = Z^{U_\ell h_\ell^{\text{attn}}} \to Z^{U_\ell h_{\ell-1}}, \quad \text{as } L \to \infty.$$

Hence, in the large-depth limit, the mean-field characterization becomes

$$Z^{h_\ell} = Z^{h_{\ell-1}} + \frac{1}{\sqrt{L}} Z^{W_{\ell,o} z_\ell} + \frac{1}{\sqrt{L}} Z^{W_\ell \phi(Z^{U_\ell h_{\ell-1}})}.$$

That is, in the large-depth limit, the feedforward network and the attention module no longer act as a composite transformation but instead contribute **additively**.

$$\frac{1}{\sqrt{N}} \| x_\ell(h_{\ell-1}) - x_\ell(h_\ell^{att}) \| = \frac{1}{\sqrt{N}} \left\| \frac{1}{\sqrt{N}} U_\ell h_{\ell-1} - \frac{1}{\sqrt{N}} U_\ell h_\ell^{att} \right\| \sim \frac{1}{\sqrt{L}}$$

# G. On the Normalization used in Deep Transformers

## G.1. QK norm

The QK norm (Henry et al., 2020) has been widely used in today's pretrained LLMs. It is mathematically defined as follows

$$b_{s,s'} = \hat{q}^\top \hat{k} \odot g,$$

where $g$ is a trainable and $\hat{q} = q/\|q\|$. In fact, this is just propositional to standard attention under $\mu$ parameterization: by omitting the temperature parameter $\tau$, we have

$$\hat{q}^\top \hat{k} = \frac{q^\top k}{\|q\| \|k\|}$$

As training deep Transformer is a valid TP, and $q$ and $k$ are all valid TP variables, as $N \to \infty$, we have

$$\frac{1}{N} q^\top k = \frac{1}{N} \sum_{i=1}^{N} q_i k_i \longrightarrow \mathbb{E}[Z^q Z^k]$$

$$\frac{1}{N} \|q\|^2 = \frac{1}{N} \sum_{i=1}^{N} q_i^2 \longrightarrow \mathbb{E}[(Z^q)^2]$$

Hence, using the continuity of the square root function, we have

$$\hat{q}^\top \hat{k} \longrightarrow \frac{\mathbb{E}[Z^q Z^k]}{\sqrt{\mathbb{E}[(Z^q)^2]}\sqrt{\mathbb{E}[(Z^q)^2]}}$$

Therefore, the QK-normed attention logits are the correlation of $Z^q$ and $Z^k$. Hence, QK-normed attention logits essentially use the $\mu$P scaling rule.

### G.2. LayerNorm

LayerNorm (Ba et al., 2016) is broadly used in deep Transformer training. Given a vector $x \in \mathbb{R}^N$, their defintion is givne by

$$\text{LN}(x)_i = \frac{x_i - \mu}{\sigma}$$

where

$$\mu = \frac{1}{N}\sum_{i=1}^N x_i, \qquad \sigma^2 = \frac{1}{N}\sum_{i=1}^N (x_i - \mu)$$

Assume $x$ is a valid TP variable in a network training. Then, we have, as $N \to \infty$,

$$\mu \to \mathring{\mu} := \mathbb{E}(Z^x), \qquad \sigma^2 \to \mathring{\sigma} := \mathbb{E}[(Z^x - \mathbb{E}(Z^x))^2].$$

As a result, we have

$$\text{LN}(x)_i \to \frac{Z^x - \mathring{\mu}}{\mathring{\sigma}^2}.$$

### G.3. RMSNorm

RMSnorm (Zhang & Sennrich, 2019) has been broadly used in foundational models. Here we demonstrate that it is also a special case using $\mu$P. The RMSNorm is defined as follows

$$\bar{a}_i = \text{RMSNorm}(a)_i = \frac{a_i}{\sqrt{\frac{1}{N}\sum_{i=1}^N a_i^2}}$$

Recall that the Master theorem from TP implies that

$$\frac{1}{N}\sum_{i=1}^N a_i^2 \to \mathbb{E}[(Z^a)^2].$$

Hence, the coordinates of $\bar{a}$ converges to the the variance-normalized version

$$\bar{a}_i \to \frac{Z^a}{\sqrt{\mathbb{E}[(Z^a)^2]}}.$$

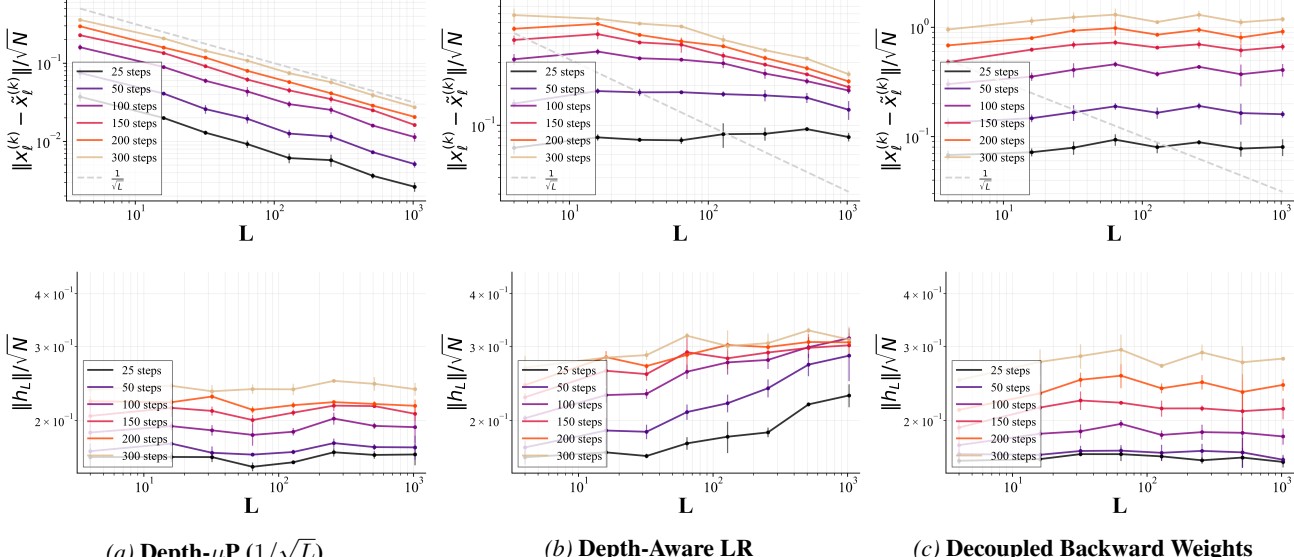

*(a)* **Depth-$\mu$P ($1/\sqrt{L}$)**    *(b)* **Depth-Aware LR**    *(c)* **Decoupled Backward Weights**

*Figure 6.* **Restoring internal learning at large depth. Top row:** Internal representation magnitude versus depth. Under standard depth-$\mu$P scaling ($1/\sqrt{L}$), internal features collapse. Both depth-aware learning rates and decoupled backward weights restore internal learning. **Bottom row:** Output magnitude versus depth. Depth-aware learning rates induce output blow-up at large depth due to amplified forward–backward interaction terms, whereas decoupled backward weights maintain stability by eliminating such interactions.

## H. Additional Experimental Results

### H.1. Two-Layer ResNets

**Restoring internal learning at large depth.** As shown in Section 2, the depth scaling $1/\sqrt{L}$ induces an *internal learning collapse* in the large-depth limit: learning within attention and MLP sublayers vanishes, while residual-stream learning remains active and stable. This silent collapse is illustrated in Figure 1. Section 4 further analyzes two principled mechanisms to counteract this effect. The first uses *depth-aware learning rates* to amplify updates to suppressed internal representations, at the risk of activating unstable forward–backward interaction channels. The second employs *decoupled backward weights*, which remove forward–backward weight reuse and hence eliminate interaction terms while preserving the same gradient updates. Figure 6 evaluates these mechanisms empirically. Both restore internal learning at large depth; however, depth-aware learning rates lead to systematic output amplification as depth increases, indicating interaction-driven instability, whereas decoupled backward weights recover internal learning while maintaining stable outputs, consistent with our theoretical predictions.

**Hyperparameter transfer across depth.** We evaluate depth-wise hyperparameter (HP) transfer on two-layer ResNets trained on CIFAR-10 with SGD for 20 epochs under three parameterizations: Depth-$\mu$P, depth-aware learning rates, and decoupled backward weights. Results are shown in Figure 7. Under Depth-$\mu$P, the optimal learning rate shifts systematically toward larger values as depth increases, indicating poor HP transfer. Both depth-aware and decoupled methods substantially improve alignment of optimal learning rates across depth, with the decoupled variant exhibiting the strongest HP transfer. However, decoupled backward weights show degraded training and test performance due to gradient misalignment, though performance remains competitive. Overall, depth-aware learning rates achieve the best test performance while maintaining stable HP transfer across depth.

### H.2. ViT

**Internal representation dynamics in Vision Transformers.** We further examine internal representation dynamics in Vision Transformers by tracking the magnitude of queries, keys, values, and attention logits across depth and training steps. Figure 8 shows that under standard Depth-$\mu$P scaling, all internal representations—including $q$, $k$, $v$, and attention logits—systematically collapse as depth increases, consistent with depth-induced internal learning suppression. In contrast, both depth-aware learning rates and decoupled backward weights successfully restore nontrivial learning dynamics across

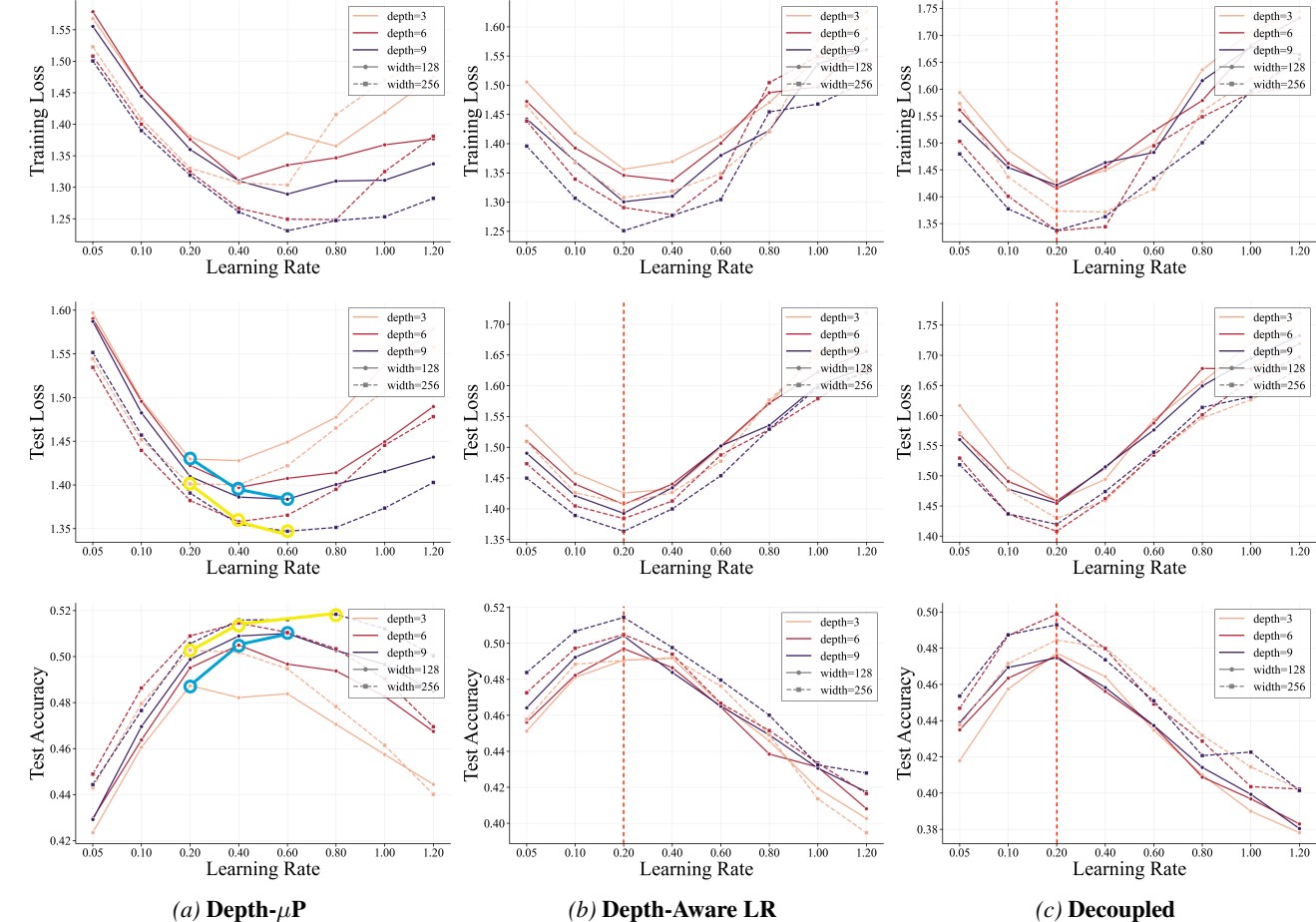

*Figure 7.* **Hyperparameter transfer across depth.** Columns correspond to Depth-$\mu$P, Depth-Aware LR, and Decoupled backward weights. Rows report training loss (top), test loss (middle), and accuracy (bottom). Hyperparameters are tuned at a reference depth and reused across depths without retuning.

all internal channels. These results confirm that the observed collapse is not specific to ResNets and extends to Transformer architectures, and that restoring internal learning requires explicitly counteracting depth-induced suppression mechanisms.

**Output magnitude dynamics in Vision Transformers.** We further analyze the output magnitude dynamics corresponding to Figure 8. Across all parameterizations, output magnitudes remain bounded at large depth, indicating that restoring internal feature learning does not necessarily induce global instability. Notably, under depth-aware learning rates, we observe a transient increase in output magnitude at early training steps as depth increases, consistent with amplified interaction effects. However, as training proceeds, this growth seems to saturate, and the output magnitudes converge, suggesting that interaction-driven amplification can be a short-horizon phenomenon rather than a persistent instability. In contrast, both Depth-$\mu$P and decoupled backward weights exhibit stable output magnitudes throughout training.

**Training-time emergence of attention structure.** We next compare how training time influences attention learning under standard Depth-$\mu$P scaling and asymmetric learning rates. Figure 9 reports histograms of attention score distributions at training steps 50, 100, and 300, stratified by depth. Under Depth-$\mu$P scaling, attention scores remain nearly uniform across all layers at early and intermediate stages, indicating suppressed attention learning. In contrast, asymmetric learning rates induce meaningful attention structure as early as step 50, not only in shallow layers but also in deeper layers. By step 100, lower layers under Depth-$\mu$P begin to exhibit weak structure, while deeper layers remain uniform; only after extended training (step 300) does structured attention gradually emerge at greater depth. These results demonstrate that asymmetric learning rates substantially accelerate attention learning across depth by counteracting depth-induced suppression. To further inspect the learned attention structures at a finer granularity, we visualize layer-wise attention heatmaps at different training

stages and different layers. For completeness, we include the corresponding visualizations for asymmetric learning rates alongside Depth-$\mu$P in Figures 10 and 11

**Hyperparameter transfer in ViT.** We further evaluate depth-wise HP transfer in ViT by measuring training and test loss across depth under Depth-$\mu$P, depth-aware learning rates, and decoupled backward weights. Figure 12 reports results. Under Depth-$\mu$P, the optimal learning rate increases with depth, leading to degraded HP transfer. Both depth-aware and decoupled methods substantially improve HP alignment across depth. Depth-aware learning rates achieve the best overall test performance, while decoupled backward weights exhibit slightly higher loss due to gradient misalignment, consistent with observations in ResNets.

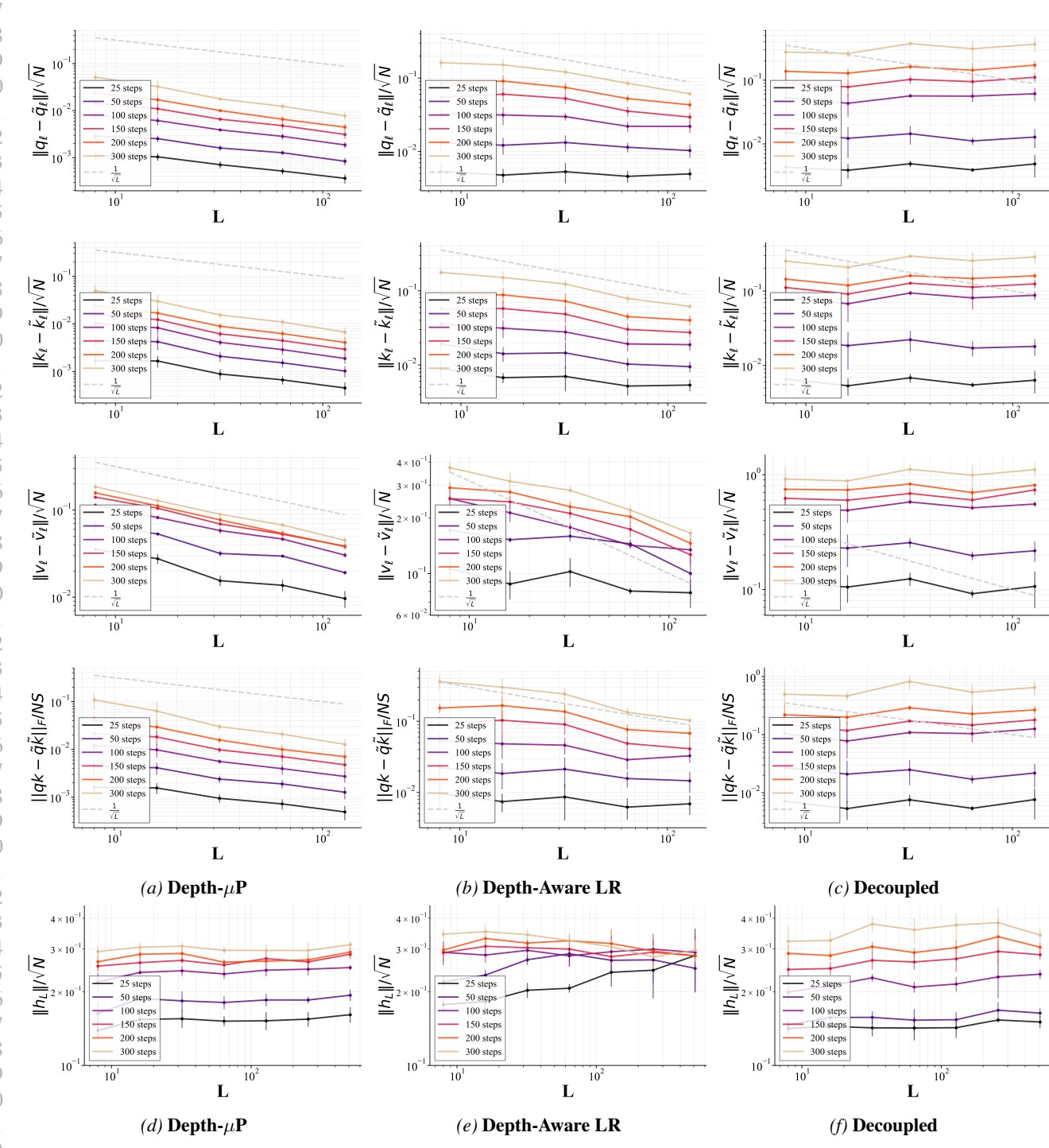

*Figure 8.* **Internal representation dynamics in Vision Transformers.** Columns correspond to Depth-$\mu$P, Depth-Aware LR, and Decoupled backward weights. Rows show the magnitude of queries ($q$), keys ($k$), values ($v$), and attention logits across depth and training. Under Depth-$\mu$P scaling, all internal representations collapse as depth increases. Both depth-aware learning rates and decoupled backward weights restore nontrivial internal dynamics, consistent with our theory.

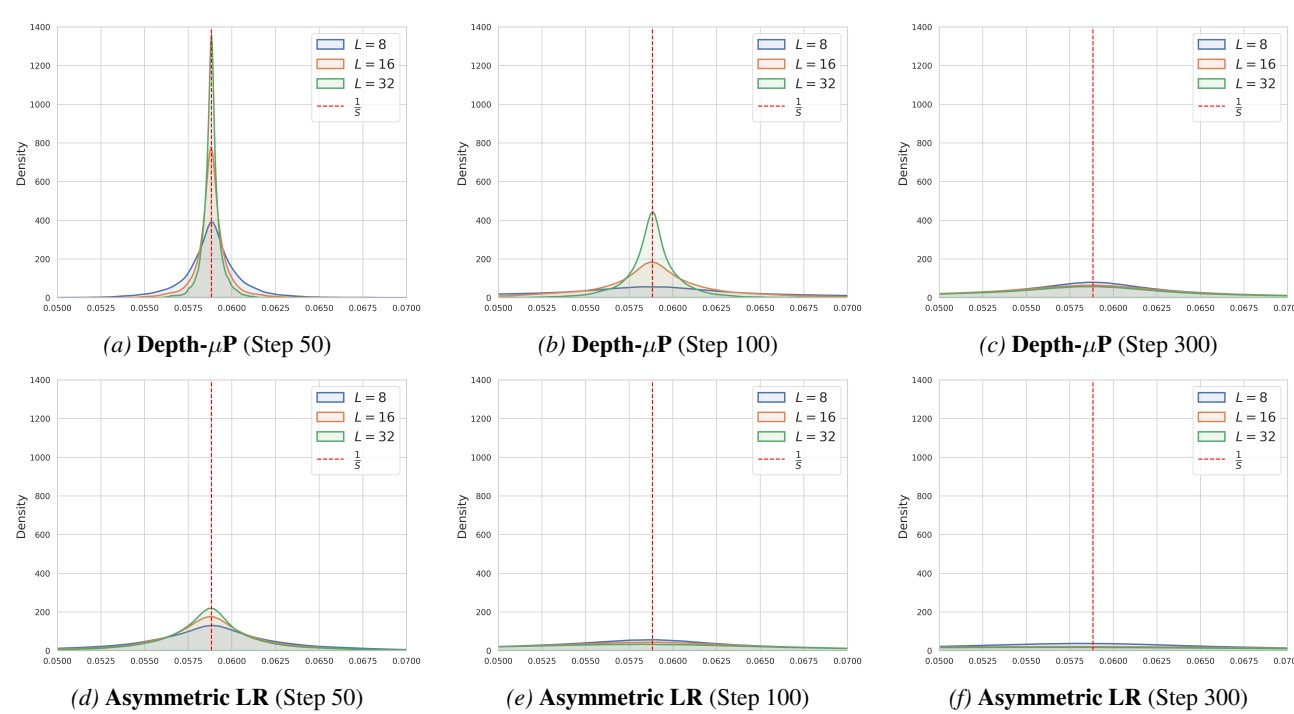

*Figure 9.* **Training-time evolution of attention score distributions.** Attention-score histograms for Depth-$\mu$P (top row) and asymmetric learning rates (bottom row) at steps 50, 100, and 300 (left to right). Under Depth-$\mu$P scaling, attention scores remain nearly uniform across training, indicating suppressed attention learning. Asymmetric learning rates induce structured, non-uniform attention substantially earlier and across greater depth, accelerating attention learning.

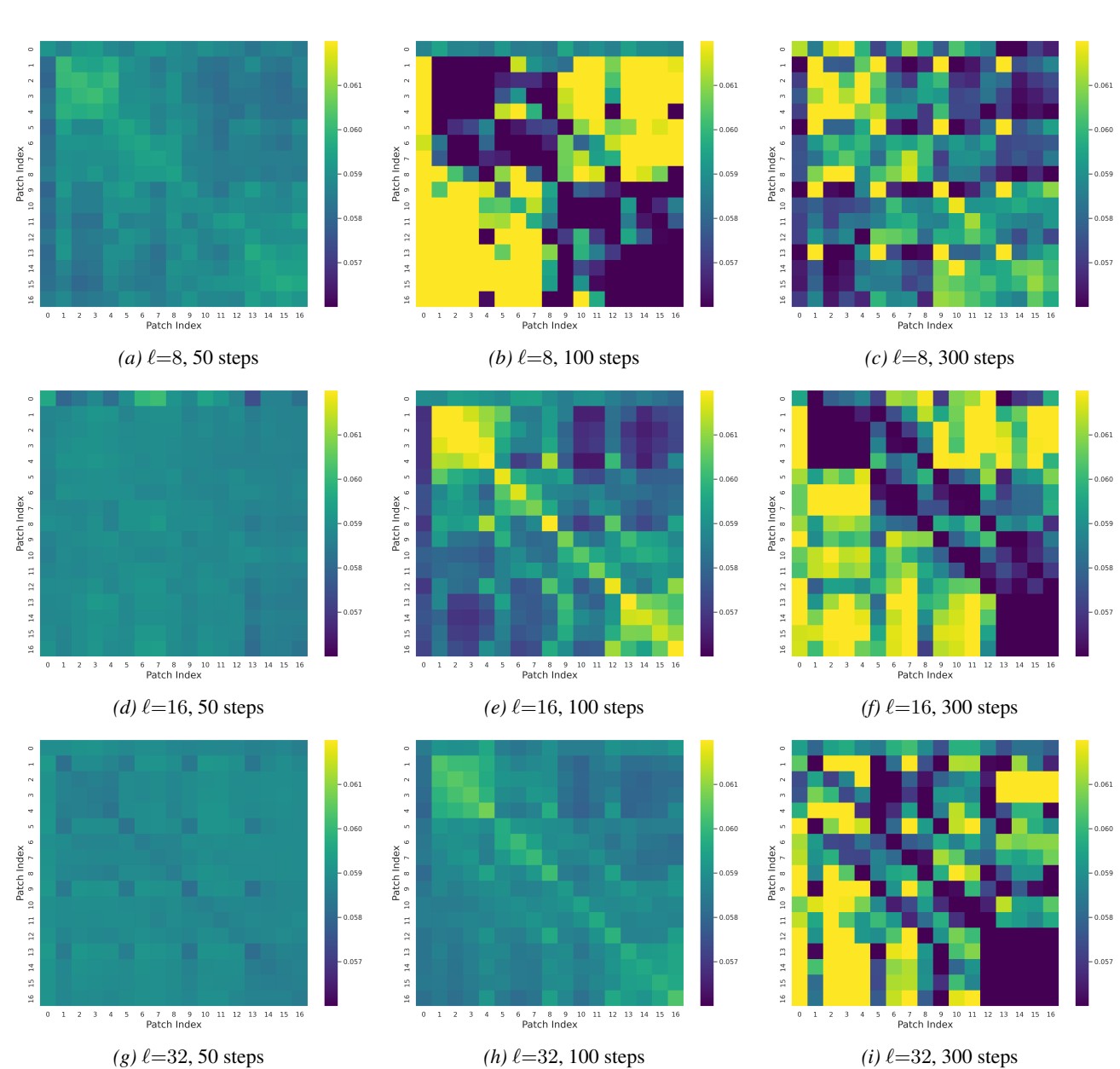

*Figure 10.* **Layer-wise homogenization of attention (depth-$\mu$P).**

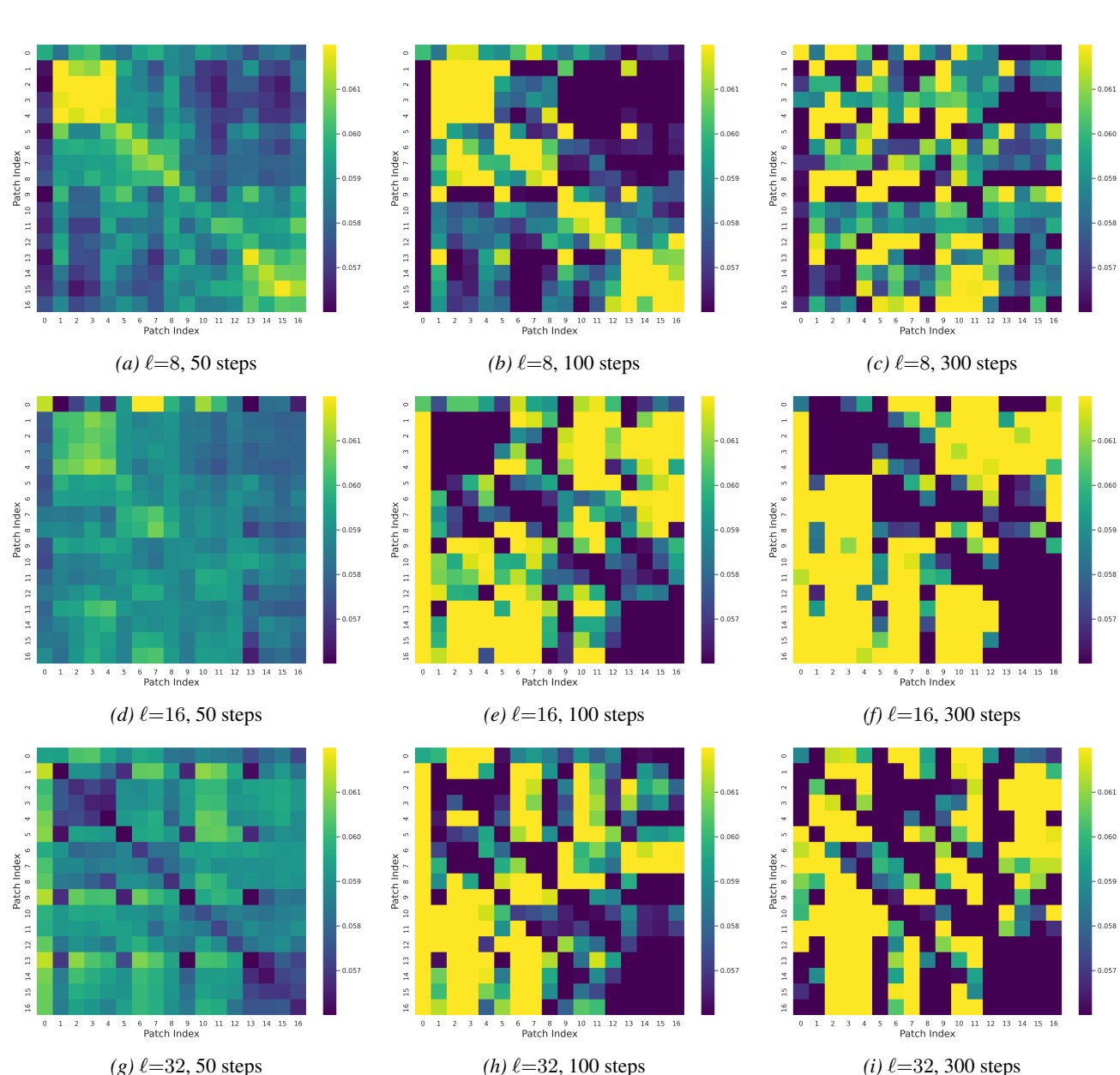

*(a) $\ell$=8, 50 steps*                *(b) $\ell$=8, 100 steps*                *(c) $\ell$=8, 300 steps*

*(d) $\ell$=16, 50 steps*                *(e) $\ell$=16, 100 steps*                *(f) $\ell$=16, 300 steps*

*(g) $\ell$=32, 50 steps*                *(h) $\ell$=32, 100 steps*                *(i) $\ell$=32, 300 steps*

*Figure 11.* **Layer-wise homogenization of attention (asymmetric learning rates).**

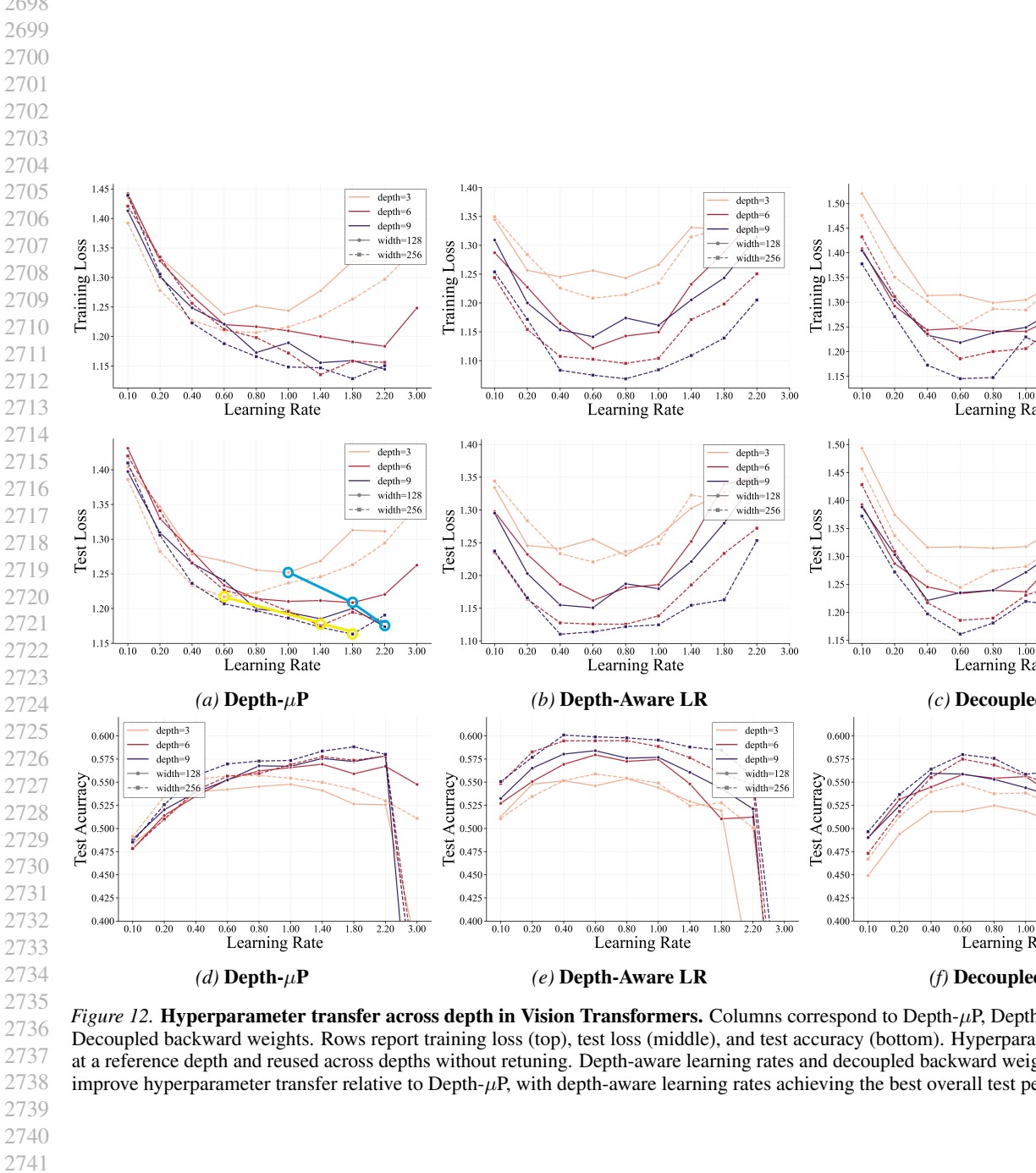

*Figure 12.* **Hyperparameter transfer across depth in Vision Transformers.** Columns correspond to Depth-$\mu$P, Depth-Aware LR, and Decoupled backward weights. Rows report training loss (top), test loss (middle), and test accuracy (bottom). Hyperparameters are tuned at a reference depth and reused across depths without retuning. Depth-aware learning rates and decoupled backward weights substantially improve hyperparameter transfer relative to Depth-$\mu$P, with depth-aware learning rates achieving the best overall test performance.

## I. Related Work

**Empirical scaling motivates a depth-aware feature-learning theory.** Transformers achieve remarkable success across diverse domains, including natural language processing (Vaswani et al., 2017) and vision (Dosovitskiy et al., 2021). A key driver of this success is empirical scaling laws (Kaplan et al., 2020; Hoffmann et al., 2022), which show that performance consistently improves as model size, data, and compute increase. However, training large-scale Transformers remains challenging, especially when scaling depth rather than width. Indeed, many practical training instabilities arise primarily from increasing depth (Liu et al., 2020). A broad empirical literature documents depth-related pathologies in Transformers, including attention entropy collapse (Zhai et al., 2023), rank collapse (Dong et al., 2021), attention sinks (Xiao et al., 2023), and near-uniform attention patterns (Hyeon-Woo et al., 2023). To mitigate these issues, numerous works propose normalization-based techniques, such as LayerNorm (Ba et al., 2016), RMSNorm (Zhang & Sennrich, 2019), and QK-normalization (Henry et al., 2020). While these methods improve training stability in practice, we still lack a theory explaining how internal Transformer representations evolve with depth

**Infinite-width kernel limits versus feature learning.** Early theoretical studies of Transformers focus on infinite-width or infinite-head limits at initialization. For example, (Hron et al., 2020) shows that multi-head attention converges to a Gaussian process as the number of heads tends to infinity, while (Sakai et al., 2025) demonstrates that, for single-head attention, the limiting distribution exhibits non-Gaussian structure due to hierarchical dependencies. Tensor Programs (Yang, 2020) provide a unified framework for analyzing forward and backward computations in neural networks and enable the derivation of the Neural Tangent Kernel (NTK) of Transformers in the infinite-width limit. However, NNGP and NTK regimes correspond to *lazy* or kernel training (Chizat et al., 2019), in which features remain close to their random initialization and do not undergo substantial representation learning. As a result, these theories do not capture the feature-learning behavior in deep Transformers during training.

**Feature-learning regimes, depth scaling, and large-depth limits.** To move beyond kernel regimes, $\mu$P parameterizations (Yang & Hu, 2021), rooted in mean-field analysis (Mei et al., 2018), enable nontrivial feature learning at infinite width and support width-wise hyperparameter transfer (Yang et al., 2021). These ideas extend to depth through depth-$\mu$P schemes (Yang et al., 2024; Bordelon et al., 2024b), which stabilize deep training and enable depth-wise transfer for simple residual blocks. However, depth-$\mu$P does not yield a rigorous feature-learning dynamics theory in the infinite-depth limit and fails to transfer hyperparameters across blocks with multiple internal layers, despite maintaining numerical stability. Closest to our setting, (Bordelon et al., 2024a) studies Transformer feature-learning dynamics at large depth using dynamical mean-field theory (DMFT), comparing $1/\sqrt{L}$ and $1/L$ residual scalings. They show that under $1/\sqrt{L}$ scaling, attention and MLP weights become effectively frozen at large depth, consistent with our observations. However, because this analysis does not resolve the *explicit depth orders* of forward–backward interaction terms induced by weight reuse, it advocates switching to $1/L$ scaling, which can lead to redundant learning dynamics (Yang et al., 2024). In contrast, our work derives precise depth orders for Gaussian, learning, and interaction components in Transformer feature dynamics, enabling targeted interventions—such as asymmetric learning rates and decoupled backward weights—to restore internal feature learning under practical $1/\sqrt{L}$ scaling while preserving training stability.

