# OpenReview forum: "A Theory of Feature Learning Dynamics in Transformers: How Depth Reshapes Learning"
_ICML.cc/2026/Conference — Submitted to ICML 2026_

### Official Review · Reviewer_Ao1D · 2026-03-02

**Soundness:** 2
**Presentation:** 2
**Significance:** 3
**Originality:** 3
**Overall Recommendation:** 3
**Confidence:** 2

**Summary:**

This paper uses Neural Feature Dynamics (NFD) framework to characterize feature and gradient evolution in Transformers. It shows that shallow transformers actively learn both residual-stream dynamics and internal representations while deep transformers only focus on residual-stream dynamics, leading to nearly uniform attention scores. Based on the analysis, the authors show some interventions which can restore internal representations and make  residual stable, and they also validate these interventions in experiments.

**Compliance With Llm Reviewing Policy:**

Affirmed.

**Final Justification:**

I will keep my score

**Key Questions For Authors:**

See above

**Limitations:**

-

**Strengths And Weaknesses:**

The key idea of this paper is insightful. Providing a rigorous theoretical analysis of deep Transformers trained with gradient descent is extremely challenging due to the highly non-convex and nonlinear nature of the optimization landscape. To make progress, the authors focus on a scaling regime where both depth and width grow large, enabling a tractable theoretical characterization. If I understand correctly, the features or gradients in this framework can be viewed as some points generalized from a distribution which is determined by $L$ (Maybe also $N$?). The authors find that feature dynamics can be decomposed into several terms, with each term has a specific order of $L$. I am not an expert on Tensor Programming (TP), so the following comments should be interpreted as suggestions from a non-specialist. I hope these suggestions may help improve the clarity and overall quality of the paper.

1. In the contribution part (around Line 87), it would be helpful if the authors could explicitly reference the specific propositions that support each claimed finding. For example, in the first contribution, are Propositions 2.2 and 2.4 the key technical results underlying the depth-induced learning phase transition?

2. From the equations in Propositions 2.2 and 2.4 alone, it is not immediately clear why “learning concentrates in the residual stream while updates to internal features become asymptotically suppressed” (Line 105). In particular, readers may get confused on what do the equations in Propositions 2.2 and 2.4 want to express, what is the connection between the equations and the statement that ''learning concentrates in the residual stream''. It seems that the residual-stream has a smaller order with respect to $L$. Similar questions on Section 3, I can not see the connections between each proposition. Are these propositions equally important? I only find Section 3.3 and 3.4 may be important on the topic ``Learning Collapse in Deep Transformer'', and I am not sure what is the role of Section 3.1 and 3.2.

3. Below Proposition 2.1, the authors state that ``any smooth test function $\psi$'', but in the reference paper that the authors provided, there exist some additional conditions to make proposition 2.1 hold such as $\psi$ is Lipschitz. Could the authors clarify whether these conditions are satisfied in the present setting? I recommend the authors to verify all the regularity conditions in this setting.

4. Are the training steps fixed here? If the training steps are fixed, then I understand that the order of each term can be expressed via $N$ and $L$. What if the training steps also tend to infinity? In practice, it seems that training steps can be related with $N$ and $L$, could the authors add some discussions on this point?

5. What is the relationship between TP and mean field theory + neural tangent kernel?  I am not sure whether this paper operate closer to the NTK regime. The manuscript would benefit from a clearer discussion of this distinction.

6. The paper is also very notation heavy, which makes it difficult to follow. I could not recommend acceptance, and I believe the authors should make additional efforts to improve the readability of the manuscript.

---

> ### Author Rebuttal · Authors · 2026-03-28
>
> We thank the reviewer for the thoughtful feedback and for the positive assessment of the main contributions. We understand the reviewer’s concerns regarding clarity and presentation. We address the key questions as follows.
>
> **Q1-Q2: Why learning concentrates in the residual stream? Role of each proposition?**
> The key point is the **difference in depth accumulation** between residual and internal features.
>
> Consider a simple linear two-layer ResNet:
>
> $$h_{\ell} = h_{\ell} + L^{-1/2} W_{\ell} x_{\ell},\quad x_{\ell} = U_{\ell}h_{\ell-1}.$$
>
> Due to depth scaling $L^{-1/2}$, gradients scale as:
>
> $$\nabla_{W_{\ell}} \mathcal{L}= L^{-\frac{1}{2}} \nabla_{h_{\ell}} \mathcal{L} \ x_{\ell}^{\top},\quad \nabla_{U_{\ell}} \mathcal{L}= L^{-\frac{1}{2}} \nabla_{x_{\ell}} \mathcal{L} \ h_{\ell-1}^{\top}.$$
>
> **Residual stream (Prop. 2.2)**
> - Due to skip connection, updates **accumulate** across $L$ layers:
> $\bar h_L= \bar h_0 + \sum_{\ell=1}^{L} L^{-1/2}(W_{\ell} + \Delta W_{\ell}) \bar x_{\ell}$
> - Each layer contributes a learning term: $L^{-1/2} \Delta W_{\ell} \bar x_{\ell} \sim \eta_W L^{-1}$
> - Aggregating over $L$ layers: $L \cdot \mathcal{O}(L^{-1}) = \mathcal{O}(1)$, so residual learning is stable and active (Prop. 2.3).
>
> **Internal feature (Prop. 2.4)**
> - The internal features $x_{\ell}$ **lack skip connections**, so no accumulation.
> - Their local update in a layer is
> $$
> \bar x_{\ell} = (U_{\ell} + \Delta U_{\ell}) \bar h_{\ell-1},\quad \Delta U_{\ell} h_{\ell-1}\sim \eta_U L^{-1/2}.
> $$
> - As $L^{-1/2}\to 0$, learning **vanishes asymptotically** (Prop. 2.5)
>
> **Extension to Transformers (Section 3)**
> The same depth-induced learning collapse applies into attention layers:
> - Residual stream $h_{\ell}$ remains active (Prop. 3.1)
> - Internal features $q, k, v$ lack skip connections, so no accumulation but have scaled gradients and learning terms scale as $\mathcal{O}(\eta_{qkv}L^{-1/2})\to 0$ (Prop. 3.2-3.3).
> - This leads to uniform attention (Prop. 3.4). and **parallelization** of MLP and attention blocks (Prop. 3.5.)
>
> We will revise the manuscript to make this structure explicit and add intuitive explanations for all key propositions.
>
> **Q3: Smooth test function $\psi$**: We thank the reviewer for pointing this out. The TP Master Theorem requires standard regularity conditions (e.g., pseudo-Lipschitz test functions), which are satisfied under our assumptions on activation functions and loss. We will revise the manuscript to reference the required assumptions (e.g., Assumption F.4 in [TP4](https://arxiv.org/pdf/2011.14522)).
>
> **Q4: Training steps and scaling**: Yes, our current analyses focus on **fixed** training steps while network width $N$ and depth $L$ diverge.
>
> We agree with the reviewer that extending to the joint scaling of network width/depth and training steps is an important direction for our future work.
>
> Based on our analysis,
> - the internal learning induced by gradient updates over $k$ training steps accumulate as $\sim k$.
> - but gradients scale as $L^{-1/2}$
> - so the effective learning scales as $k/\sqrt{L}$.
>
> Thus, if training step $k$ tends to infinity, we have $k/\sqrt{L}\to\infty$ and so internal features still learn (but slowly). This is consistent with delayed learning observed in Fig2.
>
> **Q5: Relationship to NTK and mean-field regimes**:
> Our work operates in the **feature learning regime**, not the NTK (lazy) regime.
> - In NTK, features remain close to initialization as $N\to \infty$
> - In mean-field, features evolve nontrivially but typically with **fixed depth** $L=\Theta(1)$.
>
> In contrast, we study feature learning in the large-depth regime (i.e., $L\to\infty$). In particular, we show that
> -  the residual-stream $h_{\ell}$ evolves nontrivially (feature learning),
> -  the internal features learning of $x_{\ell}$ (and $q/k/v$) exhibit a asymptotic learning suppression due to depth scaling.
> - Importantly, $x_{\ell}$ is **not** frozen to its initialization; instead, it evolves indirectly via $h_{\ell}$, i.e., $\bar x_{\ell} \sim U\bar h_{\ell-1}$.
>
> **Q6: Improving readability and presentation**: We appreciate this feedback and will improve the clarity by:
> - explicitly linking propositions to contributions
> - adding intuitive explanations for formal statements
> - moving some technical details to the appendix and simplifying notation where possible
>
> **Final remark**: We thank the reviewer again for the constructive feedback. We hope these clarifications, together with the planned revisions, will improve the accessibility of the paper and clarify our contributions. We are happy to address any further questions.

---

> > ### Author Rebuttal · Reviewer_Ao1D · 2026-04-01
> >
> > The notation needs to improve. The equation $$h_{\ell} = h_{\ell} + L^{-1/2} W_{\ell} x_{\ell}?$$ appears a typo.
> >
> >
> >
> > If the notation is wrong, let's understand it by $h_{\ell} = h_{\ell-1} + L^{-1/2} W_{\ell} x_{\ell}$, then I am still confused on the definition of  $\bar{h}_L$? Please point out the definition in your paper. I can find $\bar{h}$ in line 660-684, but before that I can not see any definition of  $\bar{h}$. Moreover, why $\bar{h}_L$ can be written as the formula only related with $\bar{x}_l$ and $\bar{h}_0$. If $\bar{h}_L$ is related with ${h}_L$, it should be a complex function depends on both $x_l$ and $h_l$.
> >
> > $\bar{h}_L$ is just one example of the unclear definition. Please define each term clearly, use \begin{definition}, and give the proof clearly. It is difficult to locate which proof corresponds to which proposition/theorem. Please add explicit cross-references (e.g., “see proof in Appendix X”) to improve readability. Please use the proof environment such as \begin{proof}[Proof of Proposition~XXX] to improve readability.
> >
> > I will keep my score.

---

> > > ### Author Response · Authors · 2026-04-01
> > >
> > > Thank you for pointing out the confusion.
> > >
> > > During the rebuttal, due to the 5000-character limit and the request for intuition, we use $\bar x_{\ell}$ and $\bar h_{\ell}$ to denote features after one SGD step with a new input $\bar x$, i.e., $\bar h_{\ell} = h_{\ell}^{(1)}$ in the original paper.
> > >
> > > The expansion is:
> > >
> > > $$\bar h_{\ell} = \bar h_{\ell-1} + L^{-1/2}(W_{\ell} + \Delta W_{\ell}) \bar x_{\ell} = \bar h_0 + L^{-1/2}\sum_{k \leq \ell} (W_{k} + \Delta W_{k}) \bar x_{k}.$$
> > >
> > > We want to clarify that $\bar h_L$ depends not only on $\bar x_{\ell}$ and $\bar h_{\ell}$ but also on $x_{\ell}$ and $h_{\ell}$ (and $\partial_{x_{\ell}} \mathcal{L}$ and $\partial_{h_{\ell}} \mathcal{L}$) via $\Delta W_{\ell}$.
> > >
> > > We agree that the presentation can be improved. In the revision, we will provide more clear definition of the symbols and add explicit references to the corresponding proofs for each proposition and theorem.

---

### Official Review · Reviewer_pA4X · 2026-03-08

**Soundness:** 3
**Presentation:** 3
**Significance:** 3
**Originality:** 2
**Overall Recommendation:** 3
**Confidence:** 3

**Summary:**

The authors investigate the effect of depth on training dynamics in transformers. In their theoretical framework, they derive expressions for the activations and their gradients after one gradient step in the mean-field limit, identifying three contributing terms and their depth scaling. In the infinite depth limit, they identify a silent learning collapse that is present in residual networks and transformers, effectively impeding learning of internal representations without showing as overall training instabilities due to continued learning in the residual stream. Based on their theoretical insights, they suggest interventions to mitigate this silent learning collapse that improve depth hyperparameter transfer.

**Compliance With Llm Reviewing Policy:**

Affirmed.

**Final Justification:**

Following the reasoning outlined in the initial review, I maintain my score.

**Key Questions For Authors:**

- How is the decoupling of the learning rates for activations and their gradients as well as the decoupling of the backward weights done in practice? How is the network parameterized in terms of weights and how are these updated?
- The authors claim a phase transition from shallow to deep models based on the learning collapse. Complete collapse, however, only occurs at infinite depth to our understanding; for deep but finite networks learning still occurs, albeit very slowly. While this slowing down certainly has implications in practice, it is unclear how the different phases are defined in finite size networks and whether it can indeed be characterized as a proper phase transition.
- Sec. 4.3 refers to Fig. 5 for empirical validation of the mentioned trade-off. From the caption and shown results it is unclear to us, however, where we see this trade-off. Further, it is unclear whether both interventions from Sec. 4.2 and 4.3 are required for the improved hyperparameter transfer in the right panels in Fig. 5 and their individual contributions.
- From the figure captions and references in the main text, the experiment details (model, data set, training method, etc.) are highly unclear and thus the interpretation of the figures themselves. Could the authors elaborate on what settings are used?

**Limitations:**

The current version of the manuscript discusses limitations only with respect to their suggested depth-scaling interventions (trade-off when decoupling learning rates and weights wrt to performance). Limitations regarding their theoretical framework, however, are not discussed. We believe that in particular the points regarding the double-limit of infinite width and depth as well as streaming SGD needs carefull discussion.

**Strengths And Weaknesses:**

**Strenghts**
- The authors utilize the Tensor Programs framework to study depth effects in training residual networks and transformers. The discussed silent learning collapse has implications for training such models in practice and thus poses a relevant field of study.
- Based on their theoretical framework, they suggest two novel interventions that mitigate this effect and improve hyperparameter transfer wrt to depth.
- The manuscript is overall well written and easy to follow.

**Weaknesses**
- The authors do not discuss limitations of their theoretical framework due to the double limit of infinite width and depth as well as using streaming SGD; models used in practice are of finite size and typically trained with optimization methods such as Adam. While we understand that mathematical tractability often requires such limits and simplifications, we believe it is important to discuss their possible impllications.
- The paper would benefit from a clearer differentiation from the work by Bordelon et al, 2024a; their work already describes the silent learning collapse, which is discussed in the submitted manuscript, from a theoretical viewpoint. While the interventions suggested in this manuscript are certainly novel, a large proportion of the main text discussed the silent learning collapse, albeit using a different theoretical approach. To gauge the novelty of their theoretical findings, it would be helpful to better understand the connection and differences to this prior work.
- The analysis and empirical validation of the suggested interventions to mitigate the silent learning collapse could be improved (see Key Questions).
- On a methodological level, a section on related works is missing from the main text. It can be found at the end of the appendix, but there is no reference to it in the main text. While we understand that there is limited space in the main text, we believe it to be crucial to allow readers to judge novelty and impact of the presented work.

**Additional remarks**
- In Fig. 5, the yellow and blue lines in the left panel on test loss presumably indicate the shift of optimal learning rate. It would be helpful to indicate their meaning in the caption.
- The current running title seems to still originate from the ICML template and should be changed to the manuscript's title.
- The current version of the manuscript is missing the required Impact Statement.
- The current Appendix A seems to still originate from the ICML template and should be removed.

---

> ### Author Rebuttal · Authors · 2026-03-29
>
> We thank the reviewer for the thoughtful feedback and the positive assessment of our theoretical analysis, the relevance of silent learning collapse, and the our exploration of the potential interventions. Below we address the key concerns.
>
> **W1: Limitations of the current theoretical framework.**
> We agree that the limitations were not sufficiently discussed and will add them in the revision. In particular:
> - For analytical tractability, we currently focus on SGD; extending to advanced optimizers (e.g., Adam, Muon) is important future work.
> - We also do not analyze the effect of trainable normalization layers, which may interact with depth scaling.
> - Our analysis uses sequential limits ($N\to\infty$, then $L\to\infty$) with training steps $k=\Theta_{N,L}(1)$. Extending to joint scaling limits of $N, L, k$ is an important future work direction.
>
> **W2: Relation to Bordelon et al. (2024a).**
> We thank the reviewer for this suggestion. Bordelon et al. (2024a) also identify silence learning collapse in deep Transformers under $L^{-1/2}$ scaling and show that switching to $L^{-1}$ scaling can restore learning.
>
> Our work differences in several key aspects:
> - *Methodology*: They use dynamical mean-field theory (DMFT), while we use the TP framework with stochastic analysis, providing a complementary and mathematically explicit characterization.
> - *Quantitative decomposition*: We not only identify collapse but also explicitly quantify the depth-scaling orders of Gaussian, learning, and interaction terms for each representation.
> - *Targeted interventions*: This quantification enables selective restoration of learning (e.g., in $q,k$) without destabilizing other components (Fig. 4).
> - *New structural result*: We show that attention and MLP blocks become asymptotically parallel at large depth. This suggests that, in practice, parallel implementations could be more appropriate in deep networks and can yield efficiency gains (e.g., ~15% training speed up reported in [prior study](https://arxiv.org/pdf/2204.02311)) that serial variants do not full exploit.
>
> **Q1: Practical implementation of decoupled weights and LRs.**
> We implement the interventions in PyTorch using a **custom autograd function**, where each residual block has separate forward and backward weights (e.g., `W_fwd` and `W_bwd`). The forward pass uses `W_fwd` to define `forward()`, while the backward pass explicitly defines gradient flow using `W_bwd` via a custom `backward()`.
>
> This enables
> - *decoupled initialization*: initialize `W_fwd` and `W_bwd` independently
> - *coupled weights*: set backward weights equal to forward weights (e.g., `W_bwd.copy_(W_fwd)`)
> - *asymmetric learning rates*: assign forward and backward weights to different optimizer `param_groups`.
>
> **Q2: Phase transition vs finite-depth behavior**.
> We thank reviewer for this insightful question. We agree that complete collapse only occurs as $L\to \infty$. Our intent was to highlight a structural change in network training from shallow to deep transformers; thus, it is more precise to describe this as a **learning change** rather than a strict phase transition.
>
> In finite-depth Transformers, internal feature learning scales as $\mathcal{O}(k/\sqrt{L})$, where $k$ is the number of training steps. Thus, the effective training time is $k/\sqrt{L}$ rather than $k$, implying that learning is not totally disappeared but significantly delayed in deeper networks.
>
> We will clarify this terminology and emphasize the finite-depth interpretation in the revision.
>
> **Q3: Clarification of Fig. 5 and intervention effects**
> In Fig. 5, as noted by the reviewer, the yellow and blue lines in the left panel indicate the shift of the optimal learning rate under vanilla depth-MuP.
>
> In the depth-MuP setting, **both** decoupled weights and asymmetric learning rates are necessary for stable and active learning and consistent HP transfer:
> - *Decoupled weights*: increasing LRs under tied weights easily leads to *exploding interaction components* (e.g., $\eta_U=\Theta(L)$ causes instability in $\delta x_{\ell}$, see Prop. 2.4 and Fig1)
> - *Asymmetric LRs*: Even with decoupled weights, naively increasing LRs to restore internal feature learning in the forward pass can destabilize the backward pass (e.g., $\eta_U=\Theta(L)$ leads to exploding updates in $\delta h_{\ell}$, see Prop. 2.2).
>
> Thus, decoupling weights and asymmetric LRs both are needed to restore active learning without destabilizing other components. We will clarify these roles and improve the figure captions in the revision.
>
> **Final remarks.**
> We will also add a brief related work section in the main text (with reference to the comprehensive appendix), clarify experimental details, and address formatting issues in the revision. We hope these clarifications, together with the planned revisions, will address the reviewer's concerns. We are happy to address any further questions.

---

> > ### Author Rebuttal · Reviewer_pA4X · 2026-04-02
> >
> > Given the raised concerns by the area chair and the impression that the current work is rather incremental, I will keep my score.

---

> > > ### Author Response · Authors · 2026-04-02
> > >
> > > Thank you for the feedback and continued discussion.
> > >
> > > We would appreciate some clarification regarding the *“concerns raised by the area chair,”* as we were not able to see specific comments corresponding to this point (possibly due to the deleted review?). Understanding these concerns would help us address them more directly, either in the current discussion or in a future revision.
> > >
> > > Regarding novelty, we acknowledge that the silent learning collapse phenomenon has been discussed in Bordelon et al. (2024a). However, we would like to clarify that our contribution goes beyond identifying the phenomenon. In particular, we provide a **quantitative characterization** of the collapse (e.g., depth scaling of different components), analyze its **structural implications** (e.g., asymptotic parallelization), and explore targeted **interventions** grounded in this quantitative analysis, which would not be feasible without such characterization.
> > >
> > > If there are specific aspects where the distinction or novelty remains unclear, we would greatly appreciate further guidance so we can address them more precisely.
> > >
> > > Thank you again for your time and consideration.

---

### Official Review · Reviewer_ey6j · 2026-03-13

**Soundness:** 2
**Presentation:** 1
**Significance:** 3
**Originality:** 2
**Overall Recommendation:** 3
**Confidence:** 3

**Summary:**

The paper studies how feature learning behaves in very deep Transformers under depth-µP scaling. Its main claim is that, although training can remain numerically stable as depth grows, the model undergoes a silent learning collapse: the residual stream continues to learn, but internal attention features such as queries, keys, and values become increasingly suppressed. To analyze this, the authors introduce Neural Feature Dynamics (NFD), which decomposes the evolution of each representation into three parts: a Gaussian term from initialization, a learning term from gradient updates, and an interaction term coming from forward–backward weight reuse. They first explain this mechanism on a simpler two-layer ResNet and then extend it to a ViT-style Transformer, arguing that deep Transformers drift toward near-uniform attention and effectively more parallel, rather than serial, block behavior.

Building on that analysis, the paper studies how to recover internal learning without breaking depth stability. A naive fix would be to raise learning rates for the suppressed internal channels, but the theory predicts that this can amplify harmful interaction terms. The authors therefore propose two more structured remedies: asymmetric learning rates, which can reactivate query/key learning while keeping the residual stream stable, and decoupled backward weights, which remove some interaction channels altogether and more fully restore internal feature learning. Their experiments, mainly on small ViTs and ResNets trained on CIFAR-10, show that standard depth-µP leads to collapse of internal representations with depth, while the proposed remedies recover nontrivial attention dynamics and improve hyperparameter transfer across depth, though decoupling can hurt performance somewhat because of gradient misalignment.

**Compliance With Llm Reviewing Policy:**

Affirmed.

**Final Justification:**

I am keeping my score. To the best of my understanding, the rebuttal does not fully resolve the main technical concerns raised in discussion.

The central issue is the treatment of tied forward/backward weights in the Transformer analysis. The authors explain that the tied-weight effect is represented by the interaction term and characterized through its depth scaling order. However, the key question is whether the reused-weight dependence is actually handled correctly in the derivation of the Transformer limit. In the feature-learning regime, the gradient independence surrogate is known to be delicate, and prior TP/NFD work suggests that tied-weight and independent-backward treatments can lead to different limiting equations unless the relevant conditioning terms are shown to vanish. In the Transformer setting, with additional couplings through q/k/v branches, attention scores, the softmax Jacobian, and token mixing, I do not think the rebuttal provides enough detail to show that this issue is fully resolved.

I also remain uncertain about the first-order Taylor approximation. The rebuttal refers to a fixed-depth remainder bound, but the concern was whether this approximation remains justified in the depth-scaled feature-learning regime studied here.

For these reasons, I keep my current score. That said, given the AC's more detailed technical comments and stronger expertise on this topic, I remain open to revising my assessment depending on the outcome of that discussion.

**Key Questions For Authors:**

Appendix B.3, line 750, you write
$$
\phi \left(\hat x_\ell + \frac{1}{\sqrt N}\Delta U_\ell \bar h_{\ell-1}\right)
\sim
\phi(\hat x_\ell)
+
\phi'(\hat x_\ell)\odot \frac{1}{\sqrt N}\Delta U_\ell \bar h_{\ell-1}.
$$
Could you clarify why this first-order Taylor approximation is valid in the feature-learning regime you study? More specifically, what is the precise asymptotic order of
$$
\frac{1}{\sqrt N}\Delta U_\ell \bar h_{\ell-1},
$$
and how do you control the second-order remainder term uniformly in depth? This seems particularly important because later parts of the paper consider depth-aware learning rates such as $\eta'_U=\Theta(\sqrt L)$ to counteract internal learning collapse, in which case the perturbation may no longer be small enough for the same first-order approximation to remain justified.

**Limitations:**

Yes.

**Strengths And Weaknesses:**

- Soundness:
1. The Transformer extension does not rigorously handle tied forward/backward weights.

In the warm-up two-layer ResNet, the relevant weight reuse is explicit already in the backward recursion, where

$$\delta h_{\ell-1}=\delta h_\ell+L^{-1/2}U_\ell^\top\delta x_\ell
\qquad\text{and}\qquad
\delta x_\ell=W_\ell^\top\delta h_\ell\odot\phi'(x_\ell)
.$$

The appendix then clearly states that the first backward is analyzed under a gradient-independence surrogate only temporarily, and that the next subsection will prove convergence without that assumption (p.18, lines 972-989).

By contrast, in the Transformer appendix the same problematic step reappears: the first backward is again derived only after "assuming that independent Gaussian matrices are used instead of $W^\top$'' (p.36, line 1959), but is not properly resolved. This is not a cosmetic simplification, since the paper itself repeatedly attributes the interaction terms to forward-backward weight reuse and coupling. However, unlike in the warm-up ResNet, the manuscript does not carry out the analogous tied-weight correction for Transformers; instead, the final theorem is followed only by a brief appeal to "similar'' arguments and omitted details. Since repairing this in Transformers would require controlling reused-weight correlations through $q/k/v$, token mixing, and the softmax Jacobian, this gap seems substantially harder than in the warm-up model and is not convincingly resolved as written.

2. The theory is compelling within the specific regime analyzed, but the scope of the rigorous results is somewhat narrower than the paper’s broader motivation. In particular, the analysis focuses on a ViT-style single-head Transformer trained with streaming SGD, and the main convergence results assume existence of the limiting NFD system together with uniform strict positive definiteness of the relevant covariance matrices. These assumptions are understandable from a technical perspective, but they leave open how directly the theory extends to more standard large-scale Transformer settings.

3. The empirical evidence is well aligned with the theory in the controlled regime considered, but the scope of that validation is still fairly narrow. The main ViT experiments are on CIFAR-10, with a single attention head, affine-free Pre-LayerNorm, and at most 20 epochs per run. The paper is transparent that extensions to trainable normalization layers, advanced optimizers, and long-horizon training dynamics are left for future work. This is a reasonable scoping decision, but it also means that the current evidence mainly supports the proposed mechanism in a simplified setting, which reduces the practical scope of the conclusions at this stage.

- Presentation:
1. The paper has a lot of content and proofs which inevitably leads to multiple important parts being moved to the appendix. In some sense this is unavoidable, but it does lead to a paper which is quite hard to read and proof-check.

2. The main idea is interesting, but the paper is harder to read than necessary. The intuition behind the failure mode and the fixes only becomes fully clear after going through a fairly dense technical development.

3. The paper would benefit from more intuitive visualizations early on, especially a simple Transformer-block schematic showing what depth-$\mu$P suppresses, how this leads to near-uniform attention / serial-to-parallel drift, and how asymmetric learning rates and decoupled backward weights modify that picture. Right now, the reader has to reconstruct much of this from later figures and proofs.

- Significance:
1. The problem the paper studies is important: understanding why depth-$\mu$P can remain numerically stable while still suppressing meaningful internal learning in deep Transformers is a relevant theoretical question. However, since the analysis and experiments focus on a relatively controlled setting, the significance of the work is currently stronger at the conceptual level than at the level of immediate practical impact.

- Originality:
1. The paper is not developing the whole framework from scratch. It explicitly introduces NFD via earlier work on two-layer ResNets, so the basic NFD lens appears inherited from prior work.

2. Prior work had also already studied large-depth Transformer feature-learning behavior and reported that under $1/\sqrt{L}$ scaling, attention and MLP weights can become effectively frozen. More broadly, closely related recent work such as CompleteP already studies depth-wise hyperparameter transfer and non-lazy learning in deep Transformers, which reduces the novelty of the present work at the level of the broader problem setting. See Nolan Dey et al., "Don't be lazy: CompleteP enables compute-efficient deep transformers".

3. The more specific originality here seems to be in extending the NFD analysis to Transformers, isolating the Transformer-specific forward-backward interaction structure, and deriving the particular remedies of asymmetric learning rates and decoupled backward weights. So I would view the contribution as meaningful but incremental rather than fully novel.

---

> ### Author Rebuttal · Authors · 2026-03-29
>
> We thank the reviewer for the thoughtful feedback. Below we address the main concerns.
>
> **S1: Tied forward/backward weights**: In the first forward/backward pass for both ResNet/Transformer, we adopt the GIA, which is standard and has been justified at **initialization** in [prior TP work](https://arxiv.org/abs/2006.14548) due to width divergence. Hence, we do not re-derive it.
>
> In the second forward/backward pass, the full tied-weight analysis is conducted under sequential limits (width $N\to\infty$, then $L\to \infty$) for both ResNet/Transformer. In particular, the width limit follows from the [TP master theorem](https://arxiv.org/abs/2011.14522). It provides a systematic way to obtain the mean-field limit by applying TP rules to the Transformer computation graph. Hence, we do not re-prove these results, but instead apply the TP framework to characterize the Transformer dynamics, and highlight distinctive depth scaling order for different representation components. The depth convergence is conduced using standard stochastic analysis. While the Transformer involves additional components (e.g., token mixing and softmax), the proof roadmap parallels the two-layer ResNet, and we thus focus on presenting the key results while omitting duplicated technical details.
>
> **S2: Limitation of theoretical results**.
> We agree that limitations should be stated more clearly and will add a dedicated discussion. Our current analysis focuses on a controlled setting for analytical tractability, including stream SGD (rather than advanced Adam), sequential limits with fixed training steps (rather than joint scaling of width/depth/training steps), and simplified architectures (e.g, multi-head attention).
>
> However, we want to clarify that these simplifications are standard in theoretical studies of deep networks and are necessary to obtain tractable and interpretable results in a complex setting such as Transformers training dynamics. In fact, some prior works (e.g., Nolan Dey et al., "Don't be lazy: CompleteP enables compute-efficient deep transformers".) primarily focus on on heuristic and intuitive analysis (e.g., two-layer linear ResNet), without explicitly analyzing attention nor providing full theoretical proofs.
>
> Hence, we view these limitations as important directions for future work rather than fundamental weaknesses.
>
> **S3: Insufficient empirical studies on simplified setting**.
> We thank the reviewer for this concern and respectfully disagree that this constitutes a fundamental weakness.
>
> Our experiments are intentionally conducted in a simplified, controlled setting to validate the theoretical findings and provide insights, rather than to target large-scale training or SOTA performance. This aligns with the theoretical focus of the work.
>
> Due to computational constraints, we cannot run billion-parameter experiments on large datasets, which is common in the current AI community. Importantly, this does not affect our conclusions, as the experiments directly validate the predicted mechanisms (e.g., delayed internal learning and recovery under interventions).
>
> In this sense, the current experimental design is appropriate: it isolates the mechanisms studied in the theory, and the setup is transparent and aligned with the analytical assumptions.
>
> **Originality**: We agree that our work builds on prior frameworks (e.g., NFD and TP) and that related works have observed forms of learning suppression in deep Transformers under $L^{-1/2}$ scaling.
>
> However, our contribution goes beyond prior observations in several key aspects:
> - **Quantitative characterization**: We provide an explicit decomposition of feature dynamics and derive precise depth-scaling orders (Gaussian, learning, interaction) for each representation, enabling a mechanistic understanding of why and where collapse occurs, and guiding our interventions.
> - **Transformer-specific structural insight**: We show that Transformer blocks become asymptotically parallel (rather than serial) at large depth. To our knowledge, this is the first theoretical result formally characterizing this transition and linking it to feature learning dynamics.
> - **Principled interventions**: Our analysis leads to targeted remedies (e.g., asymmetric learning rates, decoupled backward weights). We also show why naive fixes (e.g., uniformly increasing learning rates) fail due to interaction-induced instability.
>
> Therefore, while extending existing frameworks, our work provides new quantitative understanding, new structural insights, and principled design implications beyond prior studies on depth scaling and HP transfer.
>
> **Q1: second-order remainder term.**
> Coordinate-wise, the second-order remainder term is $N^{-1}\to 0$ uniformly in fixed depth $L$ (see [TP4](https://arxiv.org/abs/2011.14522)).
>
> **Q2: $\eta_U'=\Theta(\sqrt{L})$.**
> We explicitly state that the naive $\eta_U'=\Theta(\sqrt{L})$ does not work (Section 4.1): it leads to exploding interaction terms (see Sec. 4.1 and Fig1).

---

> > ### Author Rebuttal · Reviewer_ey6j · 2026-04-03
> >
> > Regarding S1, my concern is not whether the overall proof roadmap is similar to the ResNet warm-up, but whether the specific tied-weight correction also carries over. That is the nontrivial part. Prior TP work already shows that the gradient independence assumption is only conditionally correct in weight-sharing settings, and TP4 makes clear that in the feature-learning regime the tied-weight case and the version where the backward pass uses a fresh independent Gaussian matrix can lead to different limiting equations. So saying that the Transformer proof is “similar” is not enough by itself; the step that needs to be shown is exactly the one where reused forward/backward weights are handled correctly.
> >
> > I also do not think the transfer from the warm-up ResNet to the Transformer is automatic, because the Transformer has several extra dependencies that are not present in the simple residual block. In particular, the backward signal now passes through the q/k/v branches, attention scores, the softmax Jacobian, and token mixing. These introduce additional couplings beyond the two-layer ResNet case, so it is not obvious that the same tied-weight correction goes through unchanged. At a minimum, I would expect an explicit argument that these Transformer-specific couplings do not produce additional terms on the limit.
> >
> > Relatedly, the recent NFD ResNet paper does show that the gradient independence assumption can become valid again at infinite depth, but there this happens because of a specific vanishing mechanism in the single-layer residual setting. That is a special theorem in that paper, not a generic fact that automatically transfers to more complicated architectures. So if the claim here is that the same idea applies to Transformers, I think the paper needs to show that the same vanishing mechanism survives the Transformer-specific couplings, rather than only saying that the argument is parallel to the ResNet case.
> >
> > Please provide some of those details.

---

> > > ### Author Response · Authors · 2026-04-03
> > >
> > > Thank you for the clarification. We believe there may be a misunderstanding regarding how tied-weight effects are handled in our paper.
> > >
> > > In our framework, the tied forward/backward weight correction is **not** omitted. It is explicitly captured by the interaction term $\mathcal{I}$ in the NFD decomposition, which arises precisely from forward–backward weight reuse.
> > >
> > > A key contribution of our work is to provide a **quantitative** characterization of these tied-weight correction terms by deriving their depth scaling orders. This scaling analysis is exactly what determines whether such terms vanish or survive in the infinite-depth limit, and it also guides our targeted interventions.
> > >
> > > **ResNet (Section 2).**
> > > -  (Proposition 2.4) For internal features $x_{\ell}$, the interaction (tied-weight) term scales as $ L^{-3/2}$, which vanishes as $L\to\infty$, due to the lack of skip connection from $x_{\ell-1}$.
> > > - (Proposition 2.2) For residual dynamics $h_{\ell}$, the interaction term scales as $L^{-1}$. Due to the skip connection, it *accumulates across depth*, i.e., $\sum_{\ell}\eta_U L^{-1}\sim \eta_U$, and therefore survives at leading order.
> > >
> > > This distinction arises from skip connections, which enable accumulation across layers for $h_{\ell}$ but not for $x_{\ell}$.
> > >
> > > **Transformer (Section 3).**
> > > The same analysis is carried out:
> > > - (Proposition 3.2) The internal representations q/k/v have interaction terms scaling as $L^{-3/2}$, and therefore vanish in the infinite-depth limit.
> > > - (Proposition 3.1) The residual stream has interaction terms of order $L^{-1}$, which again *accumulate across layers* and survive.
> > >
> > > Therefore, the tied-weight correction is not assumed away, nor replaced by an independent-backward surrogate. It is explicitly derived and quantified, and its contribution is determined by its depth scaling order, which dictates whether it vanishes or is carried through (i.e., survives) in the limit.
> > >
> > > This is precisely why we say the Transformer proof is “similar”: the argument follows the same principle, identify interaction terms and analyze their depth scaling, with all Transformer specific couplings (q/k/v, attention, softmax, token mixing) incorporated into this analysis.
> > >
> > > To avoid this misunderstanding, we will revise the paper to more explicitly state that:
> > > - the interaction term corresponds to the tied-weight correction, and
> > > - whether it survives in the infinite-depth limit is determined by its depth scaling order.

---

### Decision · Program_Chairs · 2026-04-30

**Decision:**

Reject

**Comment:**

The authors study the effect of depth on feature learning dynamics in Transformers. This is an important problem, in particular, where residual blocks can be multiple layers which end up playing different roles in feature learning. However, I am not too convinced by the authors that the work is sufficiently novel, given that the neural feature dynamics (NFD), which is claimed to be a novel object, is mostly a rehashing of the Tensor Programs (TP) and Dynamical Mean Field Theory (DMFT) approaches that were previously developed. I'm not sure if there is many new ideas introduced in this work that were not previously studied by authors using TP/DMFT approaches.

My greatest concern is actually about unresolved technical issues in this submission. In particular, I don't believe the gradient independence assumption is valid in the feature learning regime. Yang in Tensor Programs II proved the gradient independence assumption only in the kernel regime, which does not apply properly in the feature learning regime. As I have outlined in a more detailed comment, the conditional dependence of the weights is the fundamental reason why we get DMFT equations in the first place, and specifically the order of the feature learning strength directly arises from the conditionally dependent terms. There is also an issue of Taylor approximations which I believe can be resolved with a more careful analysis, but not correct as stated.

As a result, I don't believe the current manuscript is ready to be accepted at the moment due to a serious mathematical issue that need to fixed. I would recommend reject and hope the authors return to fix the errors again.